# Hierarchical interactions between nucleolar and heterochromatin condensates are mediated by a dual-affinity protein

Nucleoli are surrounded by pericentromeric heterochromatin (PCH), reflecting a conserved spatial association between the two largest biomolecular condensates in eukaryotic nuclei. Nucleoli are the sites of ribosome synthesis, whereas the repeat-rich PCH is essential for chromosome segregation, genome stability and transcriptional silencing, yet the mechanisms for their co-assembly are unclear. Here we use high-resolution live imaging during *Drosophila* embryogenesis and reveal that de novo establishment of PCH–nucleolar associations is highly dynamic, as PCH transitions from extending along the nuclear edge to surrounding the nucleolus. Elimination of the nucleolus by removing the ribosomal RNA genes disrupted this process causing increased PCH compaction, followed by its reorganization into a toroidal structure. Furthermore, in embryos lacking ribosomal RNA genes, nucleolar proteins were redistributed into new bodies or 'neocondensates', including enrichment in the PCH toroidal hole. Combining these in vivo observations with molecular dynamics simulations based on multiphase wetting theory revealed that nucleolar–PCH associations can be mediated by a hierarchy of interaction strengths between PCH, nucleoli and proteins with dual affinities for both compartments. We validate this model by identifying such a protein, a DEAD-box RNA helicase called Pitchoune, and show that modulation of its affinity for either nucleolar or PCH components alters nucleolar–PCH organization. Together, this study unveils a dynamic programme for establishing nucleolar–PCH associations during animal development and demonstrates how interaction hierarchies and dual-affinity molecular linkers co-organize compositionally distinct condensates.

The eukaryotic nucleus contains many membraneless compartments or biomolecular condensates that assemble via phase-separation mechanisms, such as nucleoli, nuclear speckles, Cajal bodies and heterochromatin[1,2]. These compartments form by multivalent interactions between a unique set of macromolecules concentrated in each condensate, including proteins with structured and/or intrinsically disordered domains, and nucleic acids[3,4]. Condensates containing the same molecules can nucleate and grow at separate locations and then coarsen by fusions into larger clusters, whereas those with distinct compositions remain immiscible[5]. Nevertheless, in the crowded nuclear environment or under specific conditions (for example, stress), distinct condensates can form conserved associations with each other to form higher-order assemblies[6]. Although many studies have examined the formation and function of individual condensates, how distinct interacting condensates form and impact each other in vivo is less clear. This study addresses this

e-mail: gkarpen@berkeley.edu

question with respect to the two largest nuclear condensates, nucleoli and heterochromatin.

The nucleolus is the site of ribosome synthesis with essential roles in cell-cycle progression, stress response and protein sequestration[7]. It is a multiphasic condensate that assembles at transcribing ribosomal RNA (rRNA) genes (rDNA)[8] to form three subcompartments with distinct compositions and material properties: the fibrillar centre (FC) for rRNA transcription, the dense fibrillar component (DFC) for rRNA processing and the granular component (GC) for ribosome assembly[8–12]. In most eukaryotic nuclei, the nucleolus is surrounded by pericentromeric heterochromatin (PCH), a chromatin compartment composed of megabases of pericentromeric repeats, including tandemly repeated satellite DNAs and transposable elements[13–15]. PCH is associated with transcriptional silencing and has essential roles in nuclear architecture, chromosome segregation and genome stability[16]. Under the microscope, PCH can be visualized as chromatin regions enriched for the AT-rich DNA dye 4,6-diamidino-2-phenylindole (DAPI), histone modifications di- and trimethylation of histone H3 (H3K9me2/3), and their cognate epigenetic reader heterochromatin protein 1 (HP1)[17]. HP1 is a multivalent protein with structured and disordered domains[18] that phase separates and partitions DNA and nucleosomes in vitro[19–21] and forms a liquid-like condensate nucleated by H3K9me2/3-enriched chromatin in vivo[22–24]. One explanation for why the nucleolus is adjacent to PCH is that tandem repeats of rDNA are embedded within heterochromatic repeats on a subset of chromosomes[25]. However, cytological and sequencing analyses have revealed that PCH sequences from most chromosomes (including those lacking rDNA) contact nucleoli[26–28]; thus, *cis* proximity to rDNA is not necessary for PCH to organize at the nucleolar edge. The mechanisms that position PCH around the nucleolus are poorly understood. Understanding this is important as their co-organization is a prominent conserved feature of nuclear architecture and its disruption in senescent cells suggests that PCH–nucleolar interactions are important for cellular health[29].

In this study we use live imaging and genetic tools in *Drosophila melanogaster* to uncover dynamic patterns of de novo assembly of PCH around the nucleolus. Removal of rDNA caused dramatic changes in PCH assembly dynamics and organization, and the formation of abnormal nuclear bodies ('neocondensates') in response to the dissociation of nucleolar proteins. These in vivo phenotypes led us to develop a minimal coarse-grained model, based on the theory of wetting[30], to explain the three-dimensional (3D) organization of nucleolar phases and PCH. We found that a hierarchy of interaction strengths between PCH and nucleolar components, including 'dual-affinity' protein(s) that interact with both PCH and nucleoli, can recapitulate the layered organization of nucleoli and PCH as well as the phenotypes caused by rDNA deletion. We validate this model by identifying Pitchoune (Pit), a DEAD-box RNA helicase protein, as a dual-affinity linker required for PCH–nucleolar associations. Our study shows that affinity hierarchies between interacting condensates establish their higher-order architecture and that altering these hierarchies can remodel compartments and redistribute their constituents to form new condensates or other aberrant structures.

## Results

### Dynamic de novo assembly of PCH organization around the nucleolus during *Drosophila* embryonic development

How is the PCH condensate organized relative to the multiphasic nucleolus in *Drosophila* cells? To determine their 3D organization, we performed live-cell imaging in late-stage *Drosophila* embryos (14–16 h) and S2R+ cell lines co-expressing fluorescently tagged markers for the nucleolar subcompartments and PCH. We used RNA-polymerase I subunit E (Polr1E)[31] to label the FC, Fibrillarin (Fib)[32] and Nopp140 (ref. 33) for the DFC, Modulo (Mod)[34] and Nucleostemin 1 (Ns1)[35] for the GC, and HP1a to visualize PCH (Fig. 1a and Extended Data Fig. 1a). Consistent with the bipartite nucleolar organization reported in

*Drosophila*[36], Polr1E (FC) and Fib (DFC) co-localize and are nested within Mod (GC) in late embryos and S2R+ cells (Fig. 1a and Extended Data Fig. 1a,b). Although separate FC and DFC compartments may exist below the resolution of our imaging, we visualized this layer using Fib and refer to it as FC/DFC. Unlike human cells where each GC compartment has multiple, distinguishable FCs and DFCs[37], *Drosophila* nucleoli form one FC/DFC per GC (Extended Data Fig. 1a–c). The bulk of the PCH marker HP1a is organized around the GC in each nucleus (Fig. 1a and Extended Data Fig. 1c,d). However, in this canonical 'surrounded' conformation, HP1a does not fully encapsulate the nucleolus but covers approximately 30% of the nucleolar surface in 3D (Fig. 1c and Supplementary Video 1). This HP1a–nucleolar surrounded conformation is also observed in different tissues and later developmental stages such as gut cells in late embryos, epidermal cells in first-instar larvae and eye discs in third-instar larvae (Extended Data Fig. 1e). Together, these data define the stable 3D organization of PCH relative to the nucleolar subcompartments in *Drosophila* cells.

To determine how PCH forms the surrounded conformation around the nucleolus, we performed high-resolution time-lapse imaging of green fluorescent protein (GFP)–HP1a and red fluorescent protein (RFP)–Fib in early *Drosophila* embryos. *Drosophila* embryos undergo 14 syncytial nuclear divisions before cellularization at the blastoderm stage, with chromatin features such as H3K9 methylation progressively established during these cycles[38,39]. Although PCH condensates first emerge in cycle 11 (ref. 22), nucleoli first emerge during cycle 13 (ref. 40); thus, cycle 13 is the earliest time both condensates appear in the same nucleus. Following entry into cycle 13 interphase, HP1a and Fib proteins are initially diffuse throughout the nucleus, each becoming enriched in multiple distinct foci within approximately 8 min (Fig. 1b(i) and Supplementary Video 2). Small PCH and nucleolar condensates remain separated throughout cycle 13, probably because growth is limited by the short interphase (approximately 15 min) before both dissolve in mitosis[22,40]. Cycle 14 begins similarly to cycle 13 in that HP1a and Fib foci emerge soon after mitotic exit and are initially separated. During the longer interphase (approximately 90 min) of cycle 14, PCH and nucleoli undergo growth and self-fusions[22,24,40]. However, instead of directly forming the canonical surrounded conformation, in cycle 14 PCH extends away from the nucleolus while being tethered by the rDNA at one end, hereafter referred to as the 'extended conformation' (Fig. 1b(ii) and Supplementary Video 3). The extended conformation is also observed in nuclei with two nucleoli, which appear when the two rDNA arrays in a nucleus are not paired (Extended Data Fig. 2a). Immunostaining early embryos for H3K9me2/3, Lamin and Fib reveals that the PCH abuts the nuclear lamina in the extended conformation (Extended Data Fig. 2b).

Continued live imaging of HP1a and Fib in post-blastoderm, asynchronous cell divisions revealed how the extended PCH configuration dynamically transitions into the surrounded form observed in later developmental stages. PCH rapidly forms the extended configuration by lining the nuclear edge through the rest of cycle 15 (Fig. 1b(iii) and Supplementary Video 4), transitions between the extended and surrounded configurations during cycle 16 (Fig. 1b(iv) and Supplementary Video 5) and stably wraps around the nucleolus approximately 15 min into cycle 17 interphase (Fig. 1b(v) and Supplementary Video 6). HP1a occupancy around the nucleolus in 3D increases from 10% to 30% between cycles 14 and 17 (Fig. 1c). The PCH reorganization observed in cycle 17 is mirrored in cultured S2R+ cells exiting mitosis, where HP1a transitions through an 'extended' intermediate before stably surrounding the nucleolus (Extended Data Fig. 2c).

Next, we performed DNA fluorescent in situ hybridization (FISH) for pericentromeric repeats and rDNA to determine how different PCH and nucleolar DNA sequences are reorganized during development. In the early embryo, PCH is connected to the nucleolus due to the physical proximity of rDNA repeats to pericentromeric repeats on the X and Y chromosomes in both females (XX) and males (XY;

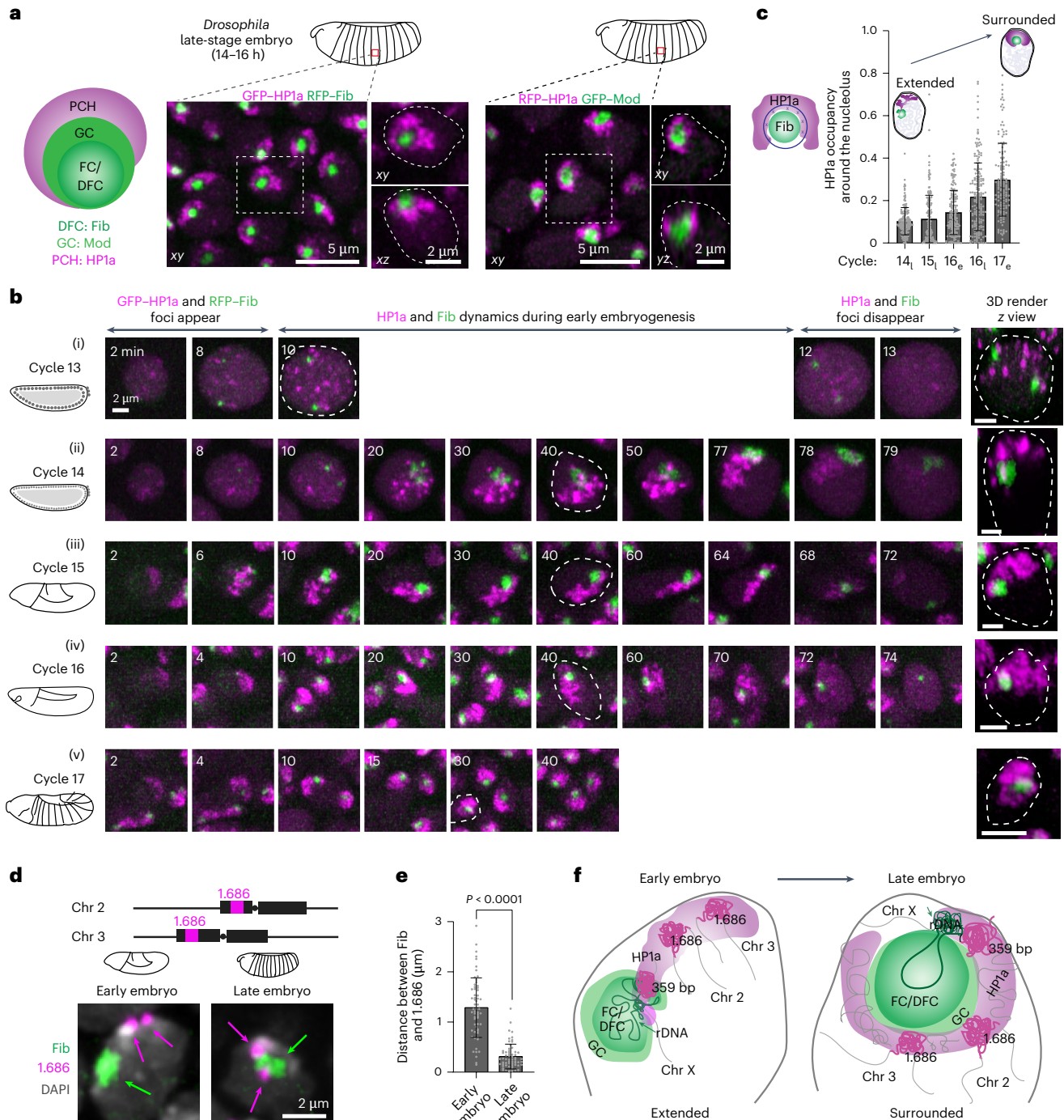

**Fig. 1 | Pericentromeric heterochromatin progressively surrounds the nucleolus during *Drosophila* embryonic development. a**, Schematic of PCH organization around the multiphasic nucleolus in *Drosophila* nuclei (left). Maximum intensity projections of live epidermal nuclei from late *Drosophila* embryos (approximately 14–16 h after egg laying; centre and right). The distribution of GFP–HP1a and RFP–Fib (centre) as well as RFP–HP1a and GFP–Mod is shown (right). Nuclei marked with a white box are shown enlarged in *xy* and *xz* projections, and the dashed lines indicate the nuclear boundaries. **b**, Maximum intensity projections (*xy* view) of individual live nuclei of GFP–HP1a and RFP–Fib in cycles 13–17 ((i)–(v), respectively) of *Drosophila* embryogenesis. The numbers on the top left corner of each image indicate the time (min) after mitotic exit. Three-dimensional renders of the nucleus marked with white dashed lines are presented (right). Time-lapse imaging at minute-scale resolution in Supplementary Videos 2–6. **c**, Levels of HP1a occupancy around the nucleolus during specific developmental cycles; subscript e, early cycle (15–30 min into the specified interphase); l, late cycle (50–70 min). The schematic next to the

graph illustrates the analysis approach: a shell was generated around Fib (dark blue circle) and HP1a occupancy was calculated as the fraction of the volume of this shell that intersects with HP1a segments (dark blue crosses). Data are the mean ± s.d. of $n$ = 162 ($14_l$), 179 ($15_l$), 174 ($16_e$), 189 ($16_l$) and 143 ($17_e$) Fib–HP1a pairs in individual nuclei analysed from five embryos imaged at the defined developmental stage. **d**, Combined immunolabelling and FISH of 1.686 repeats, Fib and DAPI in epidermal cell nuclei from fixed early (cycle 15) and late-stage *Drosophila* embryos. **e**, Distance between the centres of geometry of 1.686 loci and Fib in epidermal cell nuclei from early- (cycle 15) and late-stage *Drosophila* embryos. Data are the mean ± s.d. of $n$ =61 (early) and 62 (late) 1.686–Fib pairs pooled from three fixed embryos at each developmental stage. The *P* value was calculated using an unpaired two-tailed Student's *t*-test. **f**, Schematic summarizing the dynamic reorganization of nucleoli and PCH during *Drosophila* development, highlighting the analysed nucleolar and PCH phases, proteins and DNA repeats.

Extended Data Fig. 2d,e). In the late embryo, we observed an intense rDNA signal at the nucleolar periphery, adjacent to the 359 base pair (bp) sequence on the X chromosome (Extended Data Fig. 2f). This pattern aligns with nucleolar dominance, where the rDNA array on one X chromosome is silenced in *D. melanogaster*[41,42], and with the repositioning of silenced rDNA outside the nucleolus[43]. Consistent with HP1a reorganization observed in live embryos, the 1.686 repeat sequences, located on chromosomes lacking rDNA (chromosomes 2 and 3), are positioned away from the nucleolus in the early embryo, forming the extended conformation, but surround the nucleolus in the late embryo (Fig. 1d,e).

Together, these experiments reveal the reorganization of the two largest nuclear condensates during *Drosophila* embryonic development at high spatial and temporal resolution. We observe that PCH and nucleolar condensates undergo independent nucleation, growth and fusion, displaying cycle-specific differences, with PCH dynamically transitioning from the extended (predominantly nuclear periphery-associated) to the surrounded (nucleolar-associated) configurations (summarized in Fig. 1f).

### Removal of rDNA leads to increased PCH compaction, subsequent reorganization into a toroid-like structure and redistribution of nucleolar proteins

The specific patterns of PCH reorganization around the nucleolus during embryonic development prompted us to investigate how the nucleolus impacts PCH assembly dynamics. Therefore, we imaged RFP–HP1a and GFP–Fib in embryos completely lacking rDNA repeats (referred to hereafter as −rDNA) due to a rearranged X chromosome (C(1)DX/0) (Fig. 2a). The −rDNA embryos do not form functional nucleoli[40]; however, maternal deposition of ribosomes supports normal development through late embryogenesis, enabling us to image PCH dynamics during early nuclear cycles. In the absence of rDNA, Fib forms spherical structures (as previously reported by Falahati et al.[40]), which no longer associate with HP1a (Fig. 2a). This result suggests that Fib and HP1a proteins do not interact directly. Instead, when factors responsible for nucleolus formation (rDNA/rRNA) are absent, Fib self-associations[44] and/or secondary affinities with other molecules are sufficient to generate new dense phases (hereafter termed neocondensates).

In early −rDNA embryos, the characteristic extended HP1a conformation is replaced by a compacted structure localized to the nuclear apical edge (Fig. 2a). In late-stage embryos, when the surrounded conformation is typically observed, we unexpectedly found that PCH formed a toroidal (doughnut-like) structure in the −rDNA nuclei, with a central hole devoid of HP1a ('PCH void'; Fig. 2a). We visualized these transitions using time-lapse microscopy in cycle 15 −rDNA nuclei and observed that following PCH reassembly after mitosis HP1a collapses into a rounded structure, quantified by the significant decrease in aspect ratio (major axis/minor axis; Fig. 2b,c; +rDNA mean = 1.96, −rDNA mean = 1.40, $P < 0.0001$). Consistent with these data, the distance between pericentromeric DNA repeats 1.686 (on chromosomes 2 and 3) and AAGAG (satellite repeats on all chromosomes) decreases in −rDNA nuclei compared with the +rDNA controls (Fig. 2d,e; +rDNA mean = 0.74 μm, −rDNA mean = 0.32 μm, $P < 0.0001$), indicating increased PCH compaction when nucleoli are absent.

We next performed time-lapse microscopy in the larger amnioserosa cells[45] and visualized the emergence of the PCH void within the compacted PCH domain of −rDNA nuclei at higher spatiotemporal resolution (Fig. 2f and Supplementary Video 7). The PCH void lacked the DFC protein Fib, which formed a separate neocondensate (Extended Data Fig. 3a), and the GC protein Mod, which dispersed in the nuclear space (Extended Data Fig. 3b). The PCH void did not contain DNA, histones marked with H3K9me2 (Extended Data Fig. 3c) or any prominent accumulation of RNA (Extended Data Fig. 3d). A separate nuclear condensate, the histone locus body, appeared unaffected (Extended Data Fig. 3e). However, treatment with 488 NHS

ester (a pan-protein label)[46] revealed that the PCH void was enriched for proteins (Fig. 2g).

We conclude that PCH initially displays a more compacted morphology in embryos lacking rDNA and nucleoli. As development proceeds, the PCH morphs into a toroidal structure whose central core (the PCH void) lacks HP1a, the nucleolar proteins Fib and Mod, chromatin and RNA but is densely filled with protein(s) whose sphericity is consistent with the presence of a neocondensate. Together, these results show that nucleoli organize PCH in the 3D nuclear space by preventing PCH hypercompaction and that the absence of rDNA/rRNA results in nucleolar components forming different types of neocondensates or dispersing in the nucleoplasm.

### Modelling hierarchical interactions between the nucleolus and PCH recapitulates their in vivo organization and implicates dual-affinity proteins in their association

Given that both the nucleolus and PCH assemble via phase-separation mechanisms[10,22,24,47], we applied the theory of three-phase wetting[30] to understand how physical principles can govern their 3D organization. This theory predicts that the equilibrium configuration of three immiscible phases (for example, fully engulfing, partial engulfing and individual separated phases) is determined by the balance of their interfacial tensions, defined as the free energy cost of forming an interface between two phases (Fig. 3a). These interfacial tensions are in turn dictated by the relative interaction strengths between components of the different phases. The nucleolus is itself a multiphasic condensate whose subcompartments organize through a hierarchy of interfacial tensions; phases with higher interfacial tension relative to the nucleoplasm form the inner layers, whereas those with lower interfacial tension form outwards to minimize the total surface free energy[10].

To connect wetting theory with the in vivo spatial organization of the nucleolus and PCH in *Drosophila* cells, we developed a coarse-grained model that captures key molecular interactions between four minimal components (Fig. 3b). First, PCH was represented as a long self-attracting polymer. Second, Fib was modelled as a self-associating nucleolar protein representing the FC/DFC. Third, rDNA was included as a polymeric block within the chromatin fibre that experiences good solvent conditions and is flanked by two PCH blocks. Although we do not explicitly include rRNA, rDNA is used as its proxy for the recruitment of Fib condensates. Finally, we introduced protein X as a self-associating molecule with an affinity for PCH, to account for the spherical, protein-rich compartment formed within the PCH void in −rDNA nuclei (Fig. 2g). In the model, molecular interactions are defined by a pairwise potential with interaction strength $\varepsilon_{ij}$, where $i$ and $j$ represent rDNA (rD), Fib (F), PCH (H) or protein X (X). Interfacial tension $\gamma_{ij}$ between phases enriched in components $i$ and $j$ is proportional to their relative interaction strengths. For instance, the stronger the self-interaction of a component, the larger the interfacial tension between that component's phase and others[48].

To recapitulate the layered organization of PCH around the nucleolus, we reasoned that Fib self-associates most strongly, as it occupies the nucleolar core, while protein X self-interactions exceed those of PCH, consistent with PCH surrounding X. This hierarchy ($\varepsilon_{F-F} > \varepsilon_{X-X} > \varepsilon_{H-H}$) leads to $\gamma_{F-N} > \gamma_{X-N} > \gamma_{H-N}$, where $\gamma_{iN}$ is the interfacial tension between a phase enriched in component $i = F, X, H$ and the nucleoplasm (N). Higher interfacial tension with the nucleoplasm reduces surface exposure, yielding a layered structure with Fib as the innermost core. Next, for a stable association between PCH and the Fib-rich phase, we reasoned that protein X must also interact with Fib (or nucleolar components more generally). Therefore, we define protein X as a dual-affinity protein due to its interactions with both the PCH and nucleolar phases. These constraints led us to define a 4 × 4 interaction matrix of pairwise interaction strengths with the following interaction hierarchy, $\varepsilon_{rD-F} \geq \varepsilon_{F-F} > \varepsilon_{X-X} > \varepsilon_{F-X} > \varepsilon_{X-H} > \varepsilon_{H-H}$ (Fig. 3c), and simulations based on this

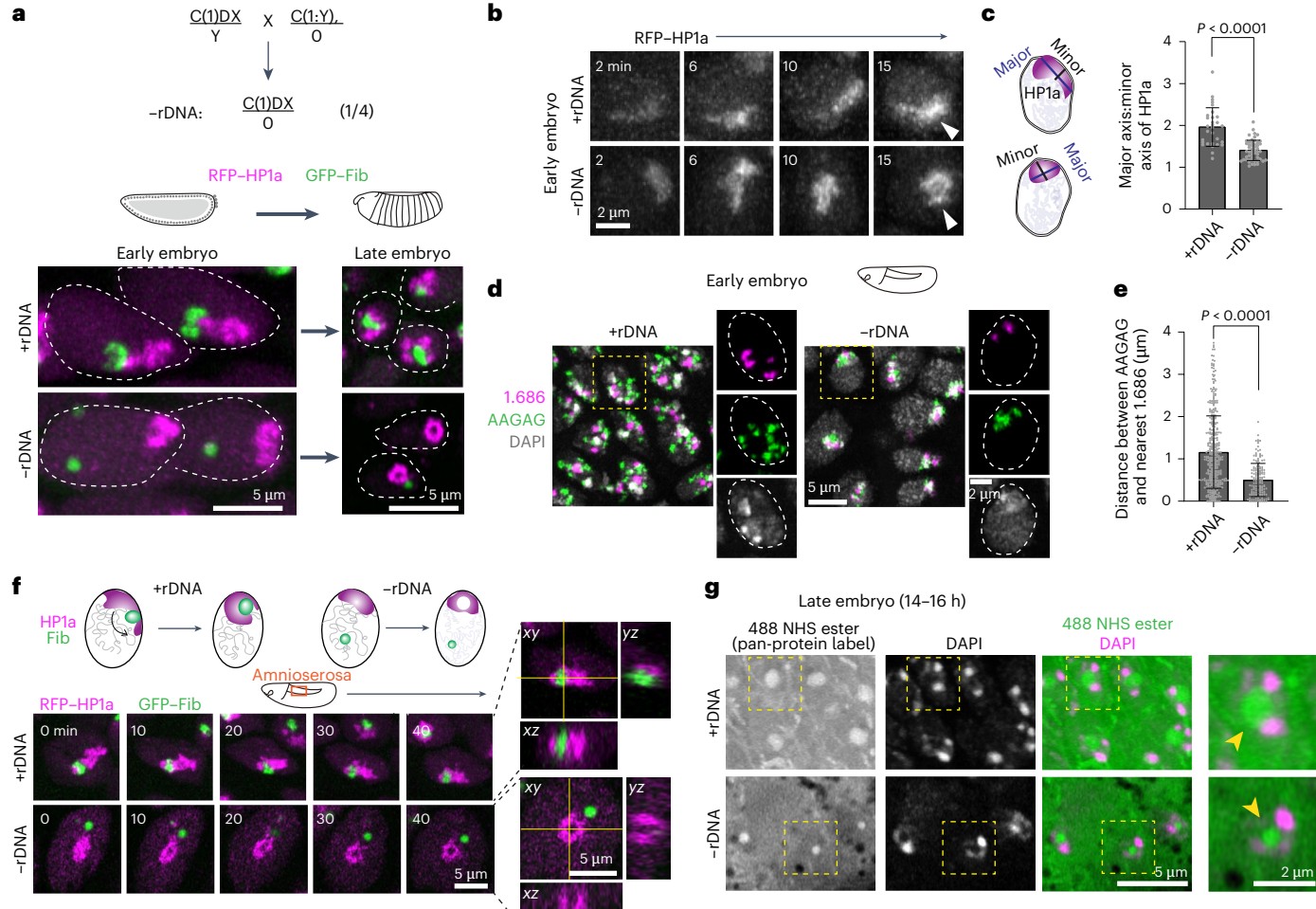

**Fig. 2 | PCH undergoes increased compaction and reorganizes into a toroid-like structure in nuclei lacking rDNA. a**, Genetic cross used to generate −rDNA embryos (top). Representative still images of live +rDNA and −rDNA nuclei in early (cycle 14) and late-stage (approximately 16 h) embryos expressing RFP–HP1a (magenta) and GFP–Fib (green; bottom). **b**, Maximum intensity projections from live imaging of nuclei from +rDNA and −rDNA embryos showing the reassembly of RFP–HP1a at the indicated time after the start of cycle 15. White arrowheads point to HP1a organization 15 min after the start of cycle 15 in +rDNA and −rDNA embryos. **c**, Aspect ratio (length of the major axis/length of the minor axis; illustrated on the left) of HP1a segments 15 min after the start of cycle 15 in +rDNA ($n = 30$ nuclei pooled from three embryos) and −rDNA ($n = 51$ nuclei pooled from five embryos) genotypes. **d**, Maximum intensity projections of FISH for satellite repeats 1.686 and AAGAG in DAPI-stained nuclei of +rDNA and −rDNA early embryos (fixed at developmental stage corresponding to cycle 15). Nuclei in the dashed yellow boxes have been enlarged (right). **a,d**, The dashed white lines indicate the nuclear boundaries. **e**, Distance between AAGAG and its nearest 1.686 locus in individual nuclei of +rDNA and −rDNA early embryos; $n = 400$ (+rDNA) and 167 (−rDNA) loci pairs pooled from nuclei in three embryos per genotype. **c,e**, Data are the mean ± s.d. **f**, Time-lapse still images (single slices) of +rDNA and −rDNA amnioserosa nuclei expressing RFP–HP1a and GFP–Fib. Time ($t$) = 0 min was set to capture the time window where PCH transitions from the extended to surrounded conformation in +rDNA nuclei and from the compacted to the toroidal structure in −rDNA nuclei. Orthogonal projections are shown along the yellow intersecting lines in amnioserosa nuclei at 40 min for the +rDNA and −rDNA genotypes. **g**, Late embryos with the +rDNA or −rDNA genotype co-stained with a pan-protein label (488 NHS ester) and DAPI (magenta) show that the PCH void in −rDNA embryos is enriched for proteins (yellow arrowhead). Nuclei in the yellow dashed box have been enlarged (right).

affinity hierarchy reproduced the surrounded conformation observed in +rDNA nuclei in vivo (Fig. 3c and Supplementary Video 8). A detailed rationale for the choice of individual parameters and the resulting simulation outcomes is provided in Methods and Extended Data Fig. 4. Deviating from this interaction hierarchy simulated outcomes that do not match the layered organization of the nucleolus and PCH. For example, if $\varepsilon_{X-X} > \varepsilon_{F-F}$, PCH does not surround the nucleolus (Extended Data Fig. 4d). Importantly, our choice of interaction parameters is not unique; as long as the interfacial tensions fulfil $\gamma_{F-N} > \gamma_{X-N} > \gamma_{H-N}$, we can obtain an engulfment of the nucleolus by PCH.

Next, we modelled the −rDNA nuclear phenotypes by setting the interaction $\varepsilon_{rD-F} = 0$, effectively eliminating the recruitment of nucleolar components such as Fib to rDNA (matrix; Fig. 3d). Reducing the attraction between Fib and protein X ($\varepsilon_{F-X}$) resulted in the equilibrium

organization of protein X within the PCH void and a spatially separate Fib condensate (Fig. 3d,e), as observed in −rDNA embryos in vivo (Fig. 2). This arrangement arises in any simulation where the protein X–PCH interaction equals or exceeds the X–Fib interactions ($\varepsilon_{X-H} \geq \varepsilon_{F-X}$; Extended Data Fig. 4d,e), suggesting that formation of the protein X-rich phase within PCH has the lowest total interfacial energy as compared with protein X forming a phase detached from PCH[30].

Together, our experimental and modelling results generate interaction hierarchies that recapitulate the multiphasic nucleolar–PCH organization under both +rDNA and −rDNA conditions. Furthermore, they predict that nucleolar–PCH interactions in +rDNA nuclei are mediated by a protein with dual affinities for both condensates, which accumulates in the PCH void in −rDNA nuclei, due to self-association combined with PCH affinity.

## The DEAD-box RNA helicase Pit is a GC protein that forms a neocondensate within PCH in −rDNA embryos

A potential candidate for a dual-affinity protein that can interact with both PCH and the nucleolus emerged from Falahati and Wieschaus[47], who examined the localization of various nucleolar proteins in −rDNA embryos at cycle 14. Although most proteins were found to either co-localize with the Fib neocondensate or disperse in the nucleus[17], the DEAD-box RNA helicase Pit caught our attention. In cycle 14 −rDNA nuclei, Pit-rich domains formed in a temperature-dependent manner, suggesting self-association, and localized to a region resembling the compacted PCH domain observed in our studies (Fig. 2a), hinting at potential PCH interactions. Pit, the *Drosophila* orthologue of DDX18, is required for larval development[49] and possesses an amino (N)-terminal intrinsically disordered region, a central helicase core and a disordered carboxy (C)-terminal domain (Fig. 4a). In addition, Pit belongs to the DDX family of proteins, other members of which phase separate and modulate condensate behaviour[50]. Together, these clues led us to hypothesize that Pit is a candidate dual-affinity protein that can form condensates and interact with both nucleolar components and PCH.

Live imaging of fluorescently tagged Pit in late embryos and S2R+ cells showed that it is enriched in the outermost layer of the nucleolus and co-localizes with Mod, confirming that Pit is a GC protein (Fig. 4a and Extended Data Fig. 5a) like its human orthologue[51]. When we transfected S2R+ cells with monomeric yellow fluorescent protein (mYFP)-tagged truncated forms of Pit corresponding to the N-terminal disordered domain (Pit[N]), the structured helicase core (Pit[Hel]) or the C-terminal disordered tail (Pit[C]), we found that Pit[N] is sufficient for nucleolar localization (Extended Data Fig. 5b), consistent with the presence of nucleolar localization motifs[52].

Next, to investigate whether Pit is enriched in the PCH void in −rDNA embryos, we performed live imaging in +rDNA and −rDNA embryos co-expressing Pit–GFP and RFP–HP1a transgenes. In cycle 14 +rDNA nuclei, Pit localized to the nucleolus, with HP1a in the extended configuration. In contrast, a faint Pit signal co-localized with the more compact HP1a domain in −rDNA nuclei (Fig. 4b). Strikingly, in cycle 17 epidermal nuclei and amnioserosa nuclei, HP1a surrounded Pit in +rDNA embryos, whereas in −rDNA embryos, high-intensity Pit puncta appeared within the HP1a/PCH void (Fig. 4b). Time-lapse imaging of amnioserosa nuclei in −rDNA embryos revealed that Pit initially mixed with HP1a but gradually separated from it, with the Pit intensity increasing approximately 2.5-fold over an hour (Fig. 4c,d and Supplementary Video 9). Furthermore, the circularity of Pit enrichment increased and approached one (in projections; Fig. 4e) and its area decreased over time (Fig. 4f). These results suggest that Pit initially mixes with HP1a in cycle 14 −rDNA embryos, and later de-mixes from HP1a and forms a protein neocondensate surrounded by PCH, likely minimizing the interfacial energy by avoiding the creation of a larger interface between Pit and the nucleoplasm.

### Pit interacts with HP1a via a conserved PxVxL motif

Pit localization patterns in +rDNA and −rDNA embryos, together with our modelling results, predict a weak interaction between HP1a and Pit. Specifically, Pit and HP1a seldom remain mixed and stronger interactions would be expected to result in constitutive mixing, and setting Pit–PCH interactions to be weaker than the other Pit interactions is required to recapitulate the surrounded conformation in coarse-grained simulations (Fig. 3c). We thus directly tested whether Pit interacts with HP1a using complementary approaches. First, we performed proximity ligation using APEX2-tagged Pit in vivo, which can detect protein interactions within a 20-nm radius[53]. We generated a stable *Drosophila* S2R+ cell line expressing APEX2–Pit, and confirmed its nucleolar localization and biotinylation activity after peroxide treatment for 1 min (Extended Data Fig. 6a). Western blot analysis confirmed the presence of HP1a among the biotinylated proteins (Fig. 5a), suggesting that Pit is located within close proximity to HP1a in vivo. Second, to determine whether

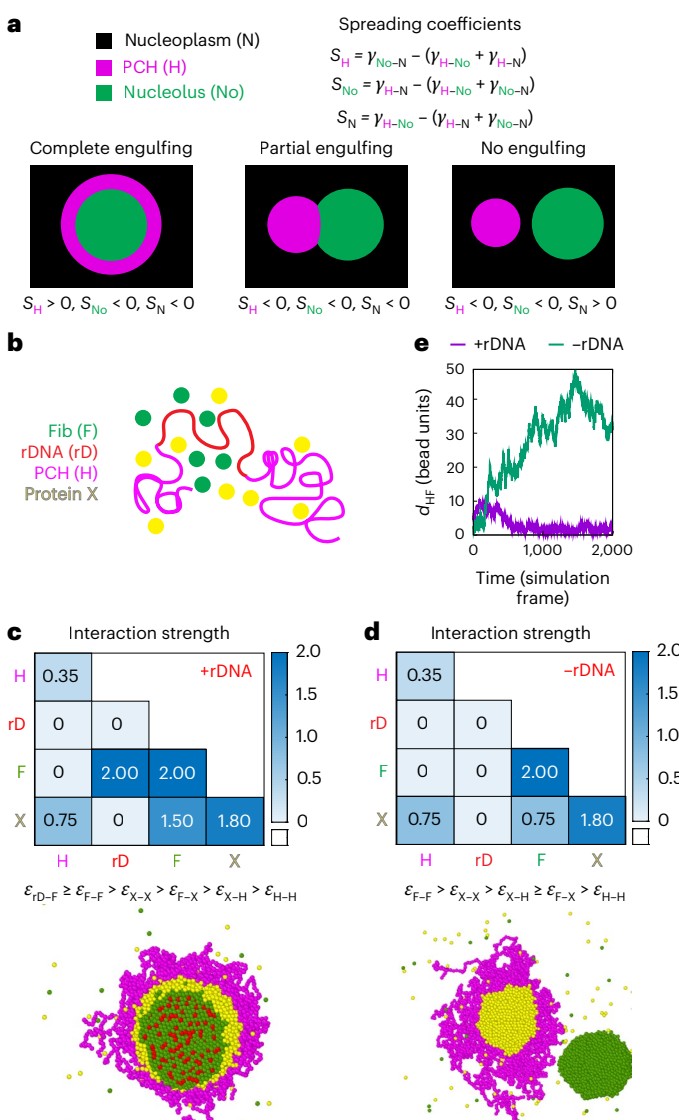

**Fig. 3 | A hierarchy of interaction strengths between PCH, nucleoli and protein(s) with dual affinities for both PCH and nucleoli recapitulates +rDNA and −rDNA in vivo organization. a**, Schematic of the spatial organization of three immiscible phases using nucleoplasm (N), PCH (H) and the nucleolus (No) as an example. The balance of the interfacial tensions ($\gamma_{ij}$) between each pair of phases ($\gamma_{H-N}$, $\gamma_{No-N}$ and $\gamma_{H-No}$) determines their equilibrium configuration. The spreading coefficient, $S_i$ (top right), is determined from these interfacial tensions. Phases with higher interfacial tension relative to the nucleoplasm form the inner layers, whereas those with lower interfacial tension form outwards to minimize the total surface free energy. PCH organizes around the nucleolus in vivo, implying $\gamma_{No-N} > \gamma_{H-N}$. Thus, the possible combinations of $S_i$ predict the configuration of the three phases, ranging from complete engulfing to partial engulfing to no engulfing. This framework motivated the development of our coarse-grained model for PCH–nucleolar organization. **b**, Coarse-grained modelling of PCH–nucleolar assembly with four minimal components: PCH (H) as a self-interacting polymer, rDNA (rD) as polymeric block embedded in PCH, Fib (F) representative of a self-associating nucleolar protein and a self-associating protein X with dual affinity to Fib and PCH representing those enriched in the PCH void in −rDNA nuclei. **c,d**, The matrices indicate the pairwise interaction strengths, $\epsilon_{ij}$ (blue gradient, units: $k_B T$), where $i$ and $j$ can represent F, rD, H or X. The indicated affinity hierarchies result in simulated outcomes that recapitulate the in vivo organization in +rDNA (**c**) and −rDNA (**d**). Snapshots of the corresponding simulation results are shown below each interaction matrix. **e**, Distance ($d_{HF}$) between the centres of mass of PCH and Fib clusters in +rDNA and −rDNA simulations. In the presence of rDNA, the distance between PCH and Fib is small, corresponding to the simulation snapshot in **c** (purple profile). In the absence of rDNA–Fib interactions, PCH and Fib are farther apart (no engulfing), corresponding to the simulation snapshot in **d** (green profile).

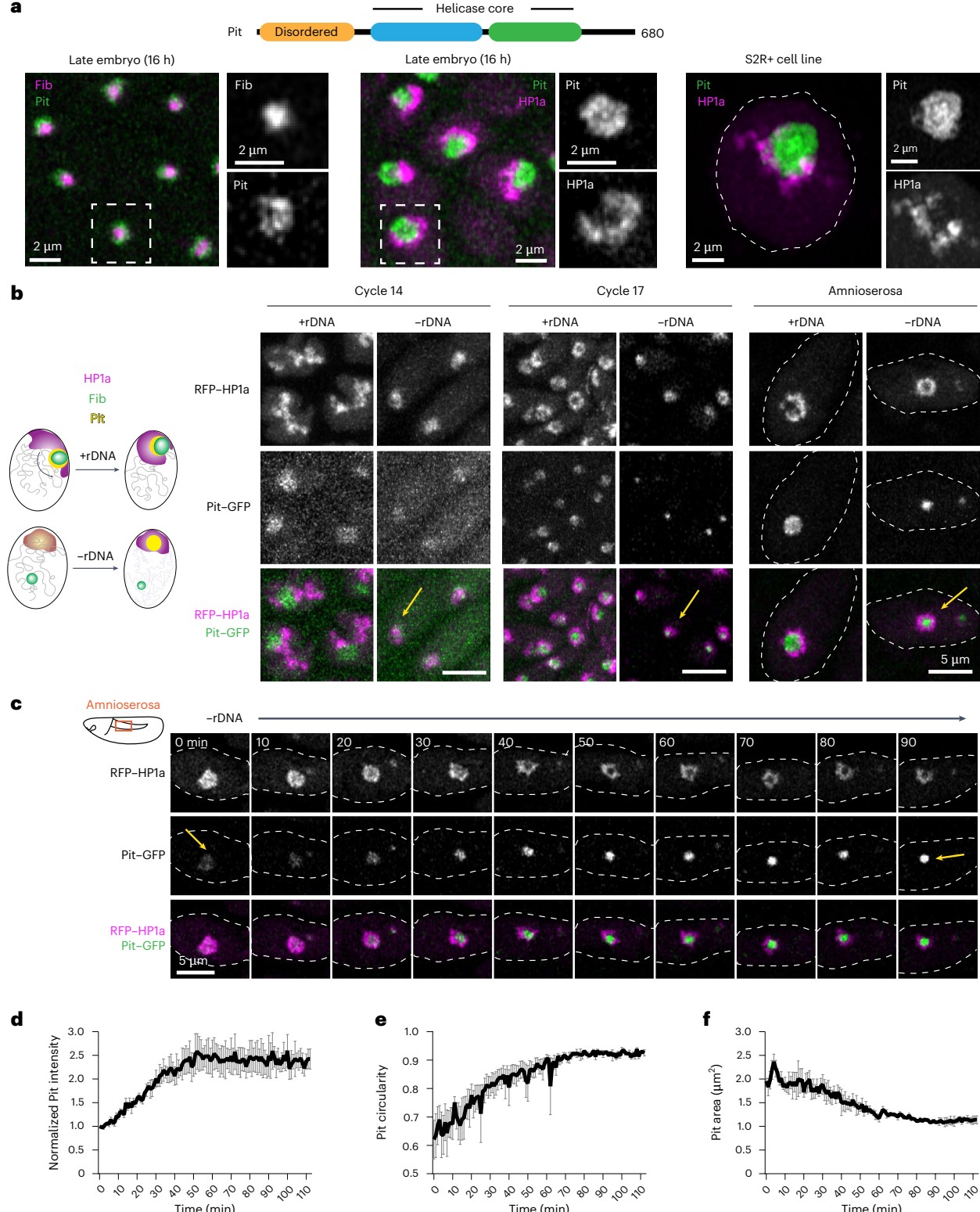

**Fig. 4 | The RNA helicase Pit is enriched in the PCH void in −rDNA embryos.**
**a**, Protein subdomains in Pit (top). Localization of RFP–Fib and Pit–GFP in a late embryo (bottom left), RFP–HP1a and Pit–GFP in a late embryo (bottom middle), and Scarlet-I–HP1a and Pit–mYFP transfected in S2R+ *Drosophila* cells (bottom right). Magnified views of the delineated regions in the main images are provided. **b**, Maximum intensity projections of cycle 14, cycle 17 and amnioserosa nuclei of +rDNA and −rDNA embryos expressing RFP–HP1a and Pit–GFP. Yellow arrows in −rDNA embryos in cycle 14 show the mixing of Pit and HP1a, whereas those in −rDNA in cycle 17 and amnioserosa show the formation of the Pit neocondensate in the PCH void. The schematic (left) summarizes the dynamics of HP1a, Fib and Pit during embryonic development in +rDNA and −rDNA nuclei. **c**, Time-lapse still images (single slices) of a −rDNA amnioserosa nucleus expressing RFP–HP1a and Pit–GFP; $t = 0$ min marks the start of a 90-min time window that captures the emergence of the Pit neocondensate within HP1a in amnioserosa nuclei. **d**, Mean intensity normalized to its value at $t = 0$ min. **e,f**, Circularity (**e**) and area of projections (**f**) of Pit–GFP neocondensates in amnioserosa nuclei of −rDNA embryos over time. At each time point, Pit–GFP segments correspond to one Pit neocondensate per amnioserosa nucleus, averaging 14 measured nuclei per embryo. **d**–**f**, Plotted values represent the average of mean measurements from $n = 3$ embryos over time. The error bars show the s.e.m.

this proximity reflects a physical interaction between Pit and HP1a, we performed co-immunoprecipitation experiments in the nuclear lysates of S2R+ cells. HP1a was enriched in samples immunoprecipitated with an antibody to Pit compared with an IgG control antibody, confirming that Pit interacts with HP1a in vivo (Fig. 5b). Note that HP1a was also prominently detected in the flow-through, consistent with the expectation that Pit–HP1a interactions are weak and/or involve only a subset of HP1a molecules in the nuclear lysate.

Third, we identified two tandem PxVxL HP1a-interacting motifs[54–56] in the C-terminal tail of Pit (PVVDLKVGA), where PVVDL is conserved across eukaryotes, and LKVGA is conserved among Drosophilids (Extended Data Fig. 6b). The Pit motif sequences suggest weak HP1a binding as they lack flanking basic residues associated with stronger binding and PCH co-localization[56]. Structural modelling using Alpha-Fold[57] predicted that the C-terminal amino acids of Pit bind to the interface generated by HP1a chromo-shadow-domain (CSD) dimerization, centred at the central V of PVVDL (Fig. 5c). Fourth, we purified histidine (His)-tagged recombinant Pit protein and determined whether it directly interacts with purified HP1a in an in vitro His-tagged pull-down binding assay (Extended Data Fig. 6c). In addition to wild-type (WT) Pit (Pit[WT]), we mutated the two central valines in the putative HP1a interaction motifs to glutamic acid (PVEDLKEGA, hereafter Pit[PxExL]). The central valine binds in a hydrophobic pocket at the interface of HP1a dimers and mutating the hydrophobic valine to the charged glutamic acid residue is known to disrupt HP1a interactions[54,56]. In addition, we predicted that if Pit binds at the interface of HP1a dimers, mutating HP1a to prevent its dimerization (HP1a[I191E])[58] should also block Pit–HP1a interactions. Pull-down binding assays that used Pit as bait and HP1a as prey revealed that Pit[WT] directly interacts with HP1a[WT]. However, this interaction was lost when Pit[WT] and HP1a[I191E] or Pit[PxExL] and HP1a[WT] were incubated together (Fig. 5d and Extended Data Fig. 6d). These in vitro and in vivo results demonstrate that Pit directly interacts with HP1a dimers via a conserved PxVxL motif.

### Pit and its HP1a interactions are required for PCH–nucleolar associations

Given the observed interaction between Pit and HP1a, we next investigated whether Pit is required to organize PCH around the nucleolus. Coarse-grained simulations predicted that a decrease in the concentration of Pit under +rDNA conditions (as defined in Fig. 3c) leads to PCH clustering away from Fib while maintaining a contact point given that rDNA is embedded between PCH (Fig. 6a). To test whether decreasing Pit concentrations impacted PCH–nucleolar associations in vivo, we performed RNA interference (RNAi)-mediated knockdown of Pit. Maternal depletion of Pit using a GAL4 driver resulted in impaired oogenesis and no egg laying, preventing the analysis of Pit knockdown in embryos (Extended Data Fig. 7a,b). To bypass this early lethality, we used eyeless-GAL4 to deplete Pit later in development in third-instar larval eye-antennal discs. Although this also caused severe developmental defects in the eye discs (Extended Data Fig. 7c), viable cells displayed a 50% reduction in the fraction of the nucleolar edge occupied by H3K9me2 after Pit knockdown (control RNAi mean = 0.296, Pit RNAi mean = 0.146, $P$ = 0.0011; Extended Data Fig. 7d,e). As a complementary approach, we also knocked down Pit transcripts in S2R+ cells (Extended Data Fig. 7f), which led to a decrease of about 50% in the HP1a occupancy at the nucleolar edge and reduced the percentage of nuclei displaying the surrounded configuration (mock RNAi, 87%; Pit RNAi, 27%; Fig. 6b,c and Extended Data Fig. 7g,h). Consistent with the simulations, PCH maintained a point of contact with the nucleolus even in the absence of Pit due to the PCH-embedded rDNA locus. Importantly, these phenotypes were rescued by the reintroduction of Pit[WT] after RNAi depletion (Fig. 6b,c and Extended Data Fig. 7h). We conclude that Pit is needed to establish the surrounded configuration of PCH and the nucleolus in larval tissues and S2R+ cells.

However, it is possible that the impact on nucleolar–PCH interactions is indirect, for example Pit depletion may result in defective GC organization or composition, perturbing other components that may be directly responsible. Therefore, to test whether Pit interactions with HP1a are directly required for PCH organization around the nucleolus, we attempted to rescue Pit RNAi phenotypes with Pit constructs defective only in HP1a binding (Pit[PxExL]). As a control, we also included a construct with a point mutation in the conserved DEVD motif (E342Q; Pit[DQVD]), which was predicted to inactivate its RNA helicase activity[59] while leaving the PxVxL motifs intact. Our rationale was that if HP1a binding is the driver of Pit-mediated nucleolar–PCH interactions, then the helicase-dead mutant should still be able to rescue Pit RNAi phenotypes. Defects in HP1a organization relative to the nucleolus caused by Pit knockdown in S2R+ cells were indeed rescued by transfection with Pit[WT] and Pit[DQVD] but not Pit[PxExL] (Fig. 6b,c and Extended Data Fig. 7h). We conclude that Pit is directly required for PCH–nucleolar associations and that this function relies on the interaction between Pit and HP1a dimers via its PxVxL motifs.

### Modulation of Pit interactions with nucleoli or HP1a alters its subnucleolar localization and PCH–nucleolar associations

In addition to decreased PCH organization around the nucleolus, reduced Pit–HP1a interactions (Pit[PxExL] mutant) also resulted in changes in the subnucleolar localization of Pit. Transfected, fluorescently tagged Pit[WT] was found to be enriched at the nucleolar periphery (GC) and centre (within FC/DFC) in S2R+ cells (Figs. 4a and 6a, and Extended Data Fig. 5a), visualized as three intensity peaks (line scans; Fig. 6d). In contrast, transfected Pit[PxExL] protein was distributed uniformly through the nucleolus and lacked peripheral enrichment. The helicase-dead mutant Pit[DQVD] displayed a diminished central peak with increased peripheral enrichment (Fig. 6b,d). We conclude that Pit's RNA helicase activity promotes its localization to the nucleolar centre where nascent rRNA is transcribed, whereas PxVxL-mediated HP1a binding promotes the enrichment of Pit at the nucleolar periphery.

To test whether Pit's distinct subnucleolar localization and PCH association patterns can emerge from a minimal set of molecular interactions, we refined our coarse-grained model (Fig. 3c) to incorporate contributions from the helicase domain and the PxVxL motifs. We introduced a direct interaction between rDNA and Pit ($\varepsilon_{rD-P}$) to represent the helicase-dependent enrichment in the nucleolar centre, using rDNA as a proxy for rRNA as previously done for Fib. Pit–HP1a binding was modelled by maintaining $\varepsilon_{H-P} = 0.75$, and the original interfacial tension hierarchy (Fig. 3c) between other components was preserved. When $\varepsilon_{rD-P} \geq 2.1$, simulations resulted in the layered organization of rDNA, Fib, Pit and HP1a, but with the additional appearance of Pit beads within the Fib cluster (Fig. 7a and Extended Data Fig. 8a,b). Plots of the radial distribution of Pit (measured from the Fib centre of mass) showed a central and peripheral peak, reflecting its experimentally observed localization patterns (Fig. 7b, blue profile). Perturbation of individual Pit interactions resulted in distinct reorganizations. For instance, removal of the interaction between Pit and PCH ($\varepsilon_{H-P} = 0$) resulted in the detachment of PCH from the nucleolar components and loss of the nucleolar periphery enrichment of Pit, resembling Pit[PxExL] (Fig. 7a,b, orange profile). In contrast, elimination of the rDNA–Pit interaction ($\varepsilon_{rD-P} = 0$) limited the Pit enrichment to only the nucleolar periphery, recapitulating the Pit[DQVD] distribution (Fig. 7a,b, green profile). These simulations show that the spatial distribution of Pit and PCH–nucleolar organization can emerge from a minimal set of competing domain-specific interactions with distinct partners. Disruption of one interaction allows others to dominate, resulting in the observed patterns of redistribution.

In our refined model, we assume that the Pit enrichment at the nucleolar centre arises from its helicase-derived interaction with nascent rRNA (modelled via rDNA as $\varepsilon_{rD-P} = 2.1$). Thus, an independent way to mimic $\varepsilon_{rD-P} = 0$ would be to stop rRNA transcription. To test this prediction, we blocked nascent rRNA synthesis using a low

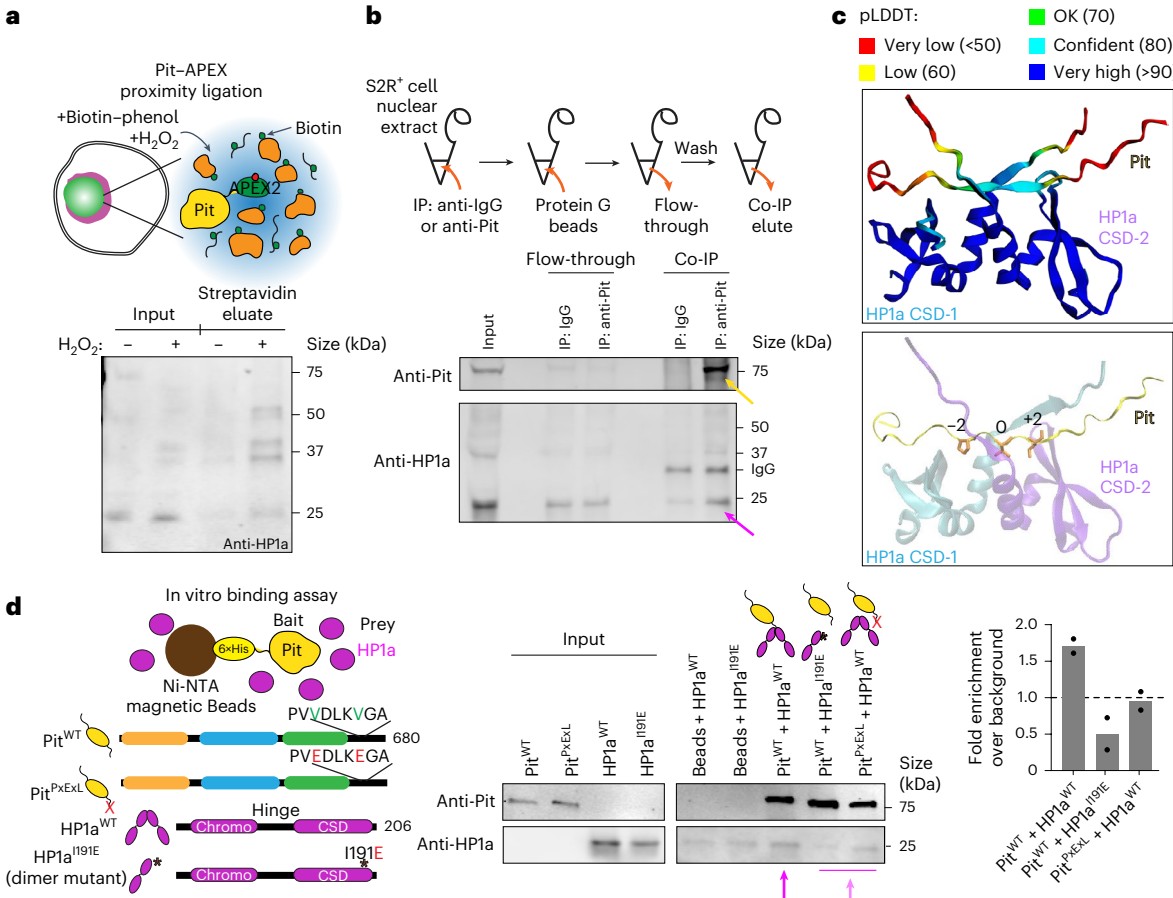

**Fig. 5 | Pit interacts with HP1a via a conserved HP1a interaction motif.**
**a**, Schematic of the proximity labelling experiment (top). In S2R+ cells expressing V5-APEX2–Pit, proteins close to Pit are biotinylated following the addition of biotin-phenol and $H_2O_2$, and then isolated using streptavidin pull-down. Input lysates and streptavidin eluates from S2R+ cells expressing V5-APEX2–Pit, with and without $H_2O_2$ treatment, were analysed by western blotting and probed with anti-HP1a showing HP1a as a biotinylated target of V5-APEX2–Pit (bottom).
**b**, Schematic of the co-immunoprecipitation (co-IP) experiment (top). Nuclear lysates from S2R+ cells were incubated with either anti-Pit or control IgG, followed by immunoprecipitation using Protein G magnetic beads. Western blot analysis of input, flow-through and co-IP samples probed with anti-HP1a and anti-Pit (bottom). HP1a (magenta arrow) co-immunoprecipitates with Pit (yellow arrow). **c**, AlphaFold-Multimer structural prediction of a 25-amino-acid peptide from the C-terminal disordered domain of Pit containing the putative PxVxL HP1a-interacting motif (PVVDLKVGA), with the CSDs of an HP1a dimer (UniProt

identifier: P05205). Heatmap indicating the predicted Local Distance Difference Test (pLDDT), a measure of confidence in the predicted structure (top). In the predicted structure, the Pit PxVxL motif binds to the HP1a–CSD dimer cleft, with the central valine residue of PVVDL positioned at zero (bottom). **d**, Schematic of in vitro binding assay where purified His-tagged $Pit^{WT}$ or $Pit^{PxExL}$ was incubated with purified $HP1a^{WT}$ or the $HP1a^{I191E}$ dimer mutant and captured using Ni-NTA magnetic beads, and then eluted for analysis (left). Western blots of input and elution fractions in the His-binding assay probed with antibodies to Pit and HP1a (centre). HP1a is enriched in the elution fraction when $Pit^{WT}$ and $HP1a^{WT}$ are incubated together (magenta arrow), indicating a direct interaction in vitro. This interaction is lost when $Pit^{WT}$ and $HP1a^{I191E}$ or $Pit^{PxExL}$ and $HP1a^{WT}$ are incubated together (faded magenta arrow), indicating that Pit binds the HP1a dimer cleft via its PxVxL motif. HP1a band intensities shown as fold enrichment over the background control (beads + HP1a only; right). Background signal was set to 1.0 (dashed line). Bars represent the mean of two independent experiments.

concentration (0.08 µg ml⁻¹) of actinomycin D (ActD), an RNA Pol-I inhibitor[60,61] (Extended Data Fig. 9a,b). Aligning with the model, Pit loses its central nucleolar localization and displays increased peripheral enrichment, accompanied by increased HP1a occupancy around the nucleolus (Fig. 7c–e). ActD treatment-induced Pit redistribution contrasts with the behaviour of two other nucleolar proteins that lack the dual PCH–nucleolar affinities identified for Pit. Fib remained localized to the FC/DFC of the nucleolus, probably due to its association with pre-existing rRNA, but its volume decreased by 50% (Extended Data Fig. 9c,d). Meanwhile, Mod (another GC protein) was no longer enriched in the nucleolus, showing a 20-fold decrease in its intensity in the nucleolus relative to the nucleoplasm (Extended Data Fig. 9e,f). Together, these experiments define the molecular basis of the affinities of Pit for nucleolar and PCH components, and demonstrate how a balance between its domain-specific interactions is required for its localization and PCH–nucleolar associations.

## Discussion

This study presents the following five main discoveries about the organization of PCH and nucleolar condensates in *Drosophila* embryos and cultured cells (summarized in Fig. 7f). First, during early embryonic development, PCH–nucleolar associations are highly dynamic, transitioning from extended (nuclear edge-associated) to surrounded (nucleolar periphery-associated) configurations. Second, the nucleolus is required for normal PCH condensate organization in the 3D nucleus and prevents PCH hypercompaction. Third, a model based on a hierarchy of interaction strengths between nucleolar and PCH components, coupled with the presence of a dual-affinity protein, recapitulates the spatial organization seen in nuclei with and without rDNA. Fourth, in vivo disruption or modulation of these interaction hierarchies leads to the dissociation of nucleolar phases, neocondensate formation and altered condensate organization. Fifth, Pit is a dual-affinity protein whose domain-specific interactions with both nucleolar and PCH

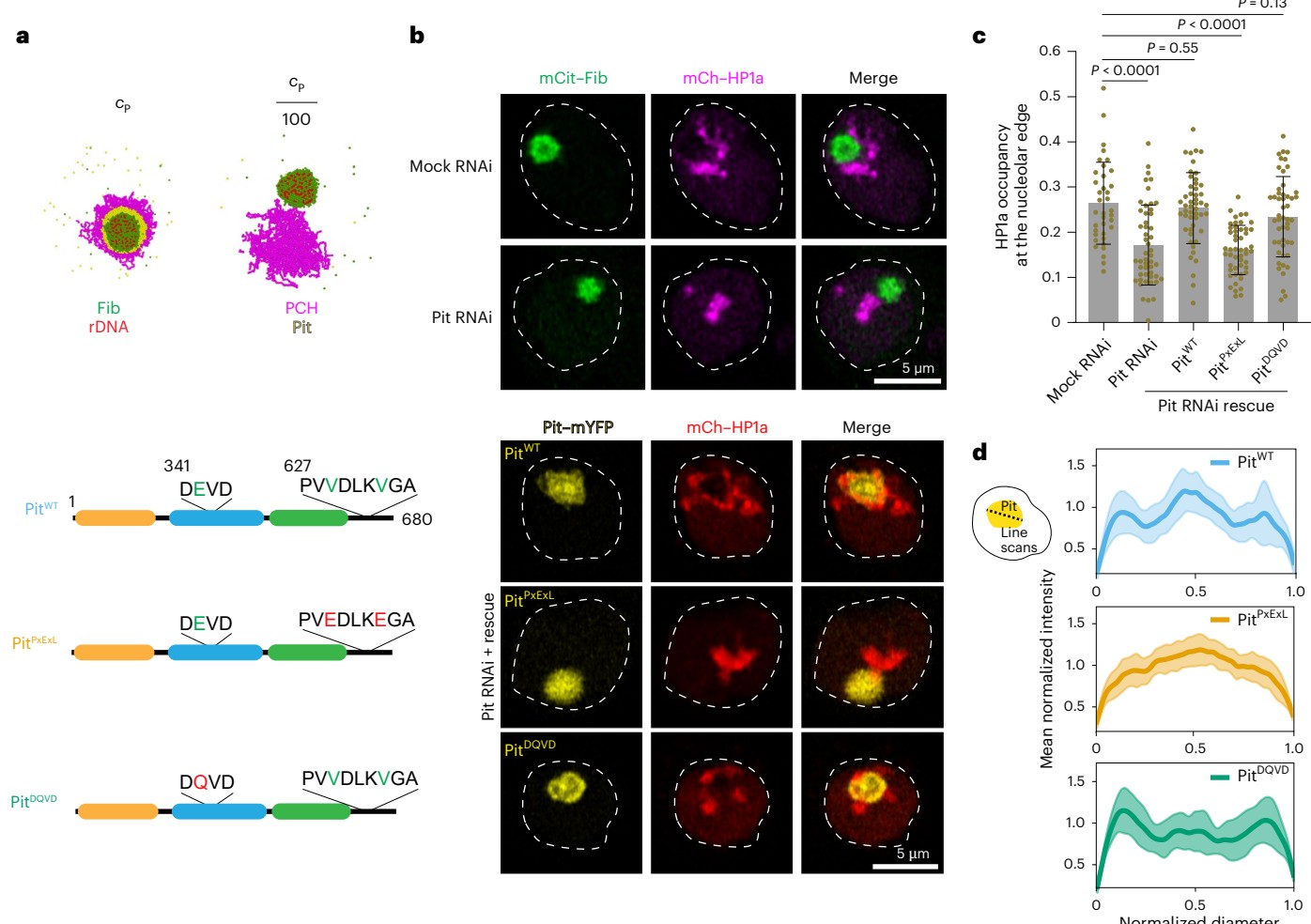

**Fig. 6 | Pit and its HP1a interactions are required for PCH–nucleolar associations. a**, Simulation end-point snapshots demonstrate how decreasing the concentration of Pit ($c_P$) predicts reduced association of PCH around the nucleolar condensate (Fib). **b**, Nuclei expressing mCitrine (mCit)–Fib and mCherry (mCh)–HP1a transiently transfected after Pit knockdown show decreased PCH–nucleolar associations (top). Schematic representation of Pit mutations: Pit$^{PxExL}$, central valine-to-glutamic acid mutations in the conserved HP1a interaction motifs; and Pit$^{DQVD}$, putative helicase-dead mutant (bottom left). Representative S2R+ cells transfected with mYFP-tagged Pit$^{WT}$, Pit$^{PxExL}$ and Pit$^{DQVD}$ following RNAi-mediated knockdown of endogenous Pit (bottom right). Rescue constructs are not targeted by Pit RNAi. The dashed lines indicate the

nuclear boundaries. **c**, Fraction of the nucleolar edge occupied by HP1a in control (mock) RNAi ($n = 37$ nucleoli, with one nucleolus per cell), Pit RNAi ($n = 50$) and Pit-knockdown nuclei rescued with full-length Pit (Pit$^{WT}$; $n = 50$), Pit$^{PxExL}$ ($n = 50$) or Pit$^{DQVD}$ ($n = 50$) pooled from two independent experiments. Data are the mean ± s.d. $P$ values were calculated using an unpaired two-tailed Student's $t$-test. **d**, Line scans show Pit distribution in the nucleolus in cells transfected with Pit$^{WT}$ ($n = 25$ nucleoli, with one nucleolus per cell; top), Pit$^{PxExL}$ ($n = 37$; middle) and Pit$^{DQVD}$ ($n = 40$; bottom) after endogenous Pit depletion. Intensities were normalized to the average value for each profile, and lengths were normalized to the diameter of the corresponding nucleolus. The mean is indicated by the solid line and the shaded region indicates the s.d.

components are required for its subnuclear localization and PCH organization around the nucleolus.

Sequence-based approaches have revealed that 3D nuclear organization transitions from a naive state to specific higher-order patterns during embryonic development[62]. However, these methods typically exclude the highly repetitive sequences that comprise most of the PCH and nucleoli, limiting our understanding of how PCH–nucleolar organization is established during animal development. Our study addresses this gap by visualizing PCH and nucleolar dynamics during *Drosophila* embryonic development in individual nuclei at minute-scale resolution. We find that PCH from different chromosomes undergo liquid-like fusions[22,24] to form a contiguous condensate that is connected to nucleoli at the rDNA locus. Initially, the fused PCH is in an intermediate extended configuration, which subsequently wraps around the nucleolus to form the surrounded state, explaining how PCH from chromosomes without rRNA genes are also positioned at the nucleolar edge[28] (Fig. 7f(i)). PCH in the extended configuration

is closely associated with the nuclear periphery. The dynamic interactions of the PCH domain with both the lamina and the nucleolus are consistent with sequence-based studies showing partial overlap between genomic loci defined as nucleolus-associated domains and lamina-associated domains[27,63]. Transitions in heterochromatin organization reminiscent of the extended intermediate have been observed during early embryonic development in *Caenorhabditis elegans*[64] and mice[65], suggesting similar dynamics between species. Although the precise mechanisms driving PCH repositioning during embryonic development remain unclear, our affinity hierarchy model predicts that an increase in Pit–HP1a interactions between cycles 14 and 17 (and potentially reduced lamina associations) may drive the reorganization of PCH around the nucleolus in +rDNA nuclei while temporally coinciding with the formation of a Pit-rich phase within PCH in −rDNA nuclei (Fig. 7f(ii)). Mesoscale changes causing an increase in interactions between Pit and HP1a could include post-translational modifications of Pit, increased Pit concentrations, reduced nuclear size during these

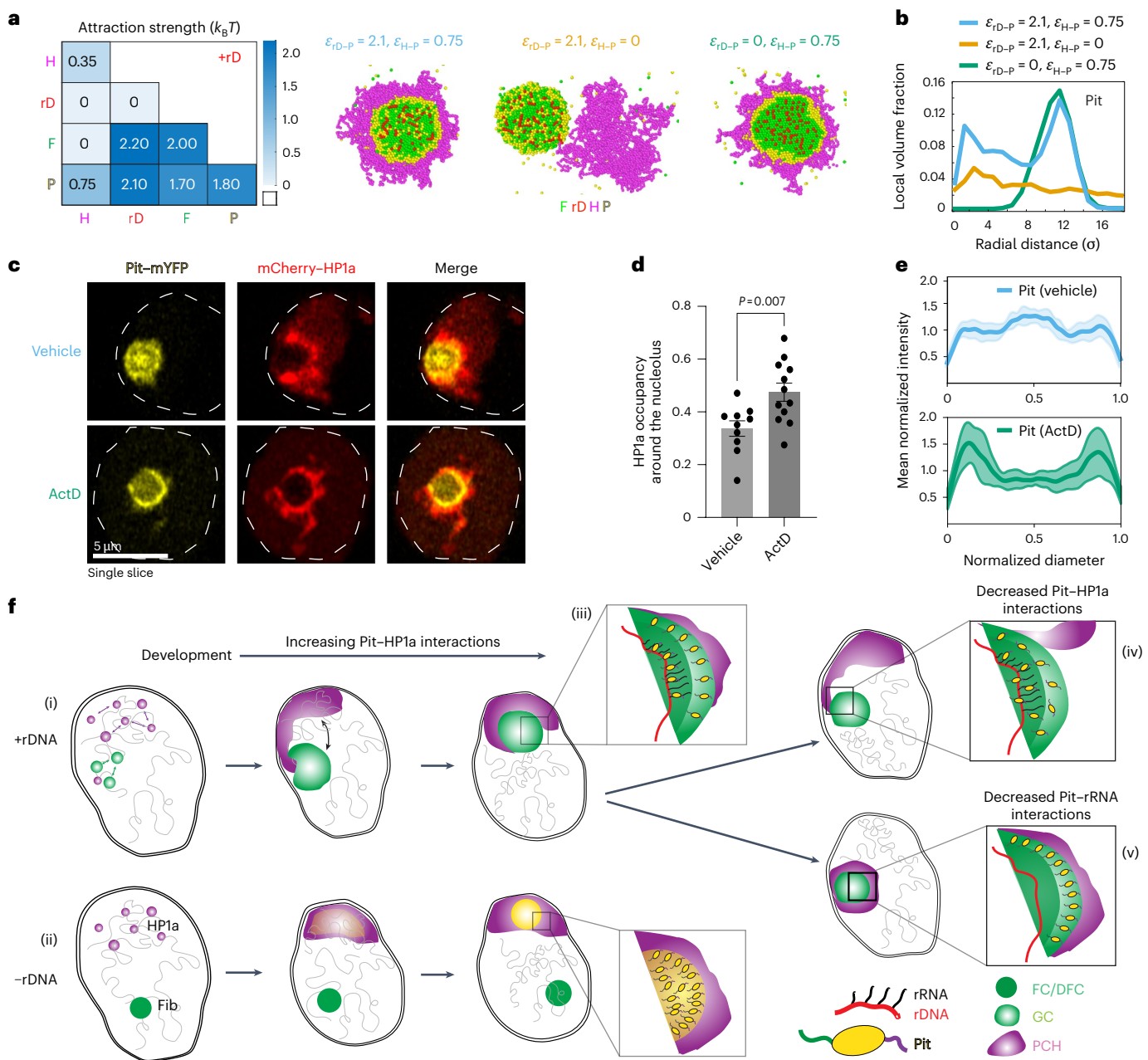

**Fig. 7 | Modulating Pit's interactions with the nucleolus or HP1a alters its subnucleolar localization and PCH–nucleolar associations. a**, Attraction matrix showing the pairwise interaction strengths (blue gradient) between Fib (F), rDNA (rD), PCH (H) and Pit (P) (left). In this refined model, an interaction between Pit and rDNA is included ($\varepsilon_{rD-P} = 2.1$), which results in the simulated outcome recapitulating the WT in vivo organization. Additional simulations at $\varepsilon_{H-P} = 0$ represent the condition where Pit–HP1a interactions are disrupted, whereas $\varepsilon_{rD-P} = 0$ simulates conditions where the Pit–rDNA association is lost. Snapshots of the three corresponding simulation outcomes are shown (right). **b**, Radial distribution of Pit calculated from the centre of mass of the Fib beads (nucleolar centre) under the three interaction conditions. The x axis shows the radial distance ($\sigma$, bead size), where zero corresponds to the nucleolar centre. **c**, Representative live-cell fluorescence images of S2R+ cells transfected with Pit–mYFP and mCherry–HP1a, and treated with vehicle (dimethylsulfoxide) or 0.08 μg ml⁻¹ ActD, imaged between 10 and 60 min of drug treatment. Pit is enriched at the edge of the nucleolus and HP1a localization around the nucleolus increases after ActD treatment. The dashed lines indicate the nuclear boundaries. **d**, Levels of HP1a occupancy around the nucleolus in S2R+ cells treated with vehicle (n = 10 nucleoli, with one nucleolus per cell) or 0.08 μg ml⁻¹ ActD (n = 12 nucleoli). Data are the mean ± s.d. The P value was calculated using an unpaired two-tailed Student's t-test. **e**, Line scans show Pit distribution in the nucleolus of S2R+ cells treated with vehicle (n = 18 nucleoli, with one nucleolus per cell) or 0.08 μg ml⁻¹ ActD (n = 19 nucleoli). Intensities were normalized to the average value for each profile and lengths were normalized to the diameter of the corresponding nucleolus. The mean is indicated by the solid line and the shaded region indicates the s.d. **f**, Model summarizing how the dual-affinity linker Pit mediates the spatial organization of PCH and nucleolar condensates in vivo. Modulating nucleation sites, Pit concentrations and interaction motifs alter the affinity hierarchies leading to specific rearrangements in condensate architecture and Pit distribution. Scenarios relating to (i)–(v) are described in Discussion.

developmental stages that brings PCH closer to Pit or changes in the molecular composition of PCH and/or nucleoli. Future studies will reveal whether one or more of these mechanisms mediate the dynamic transitions in PCH–nucleolar association during early development.

An unexpected finding from this study was that in the absence of rDNA, PCH becomes more compact and then reorganizes into a toroid-like structure with Pit occupying the toroidal hole. To explain this phenotype, we propose the interaction hierarchy model of

PCH–nucleolar organization, in which a gradient of interaction strengths between biomolecules creates a hierarchical organization and stable associations between immiscible PCH and nucleolar condensates. We draw from the differential tension model used to explain coexisting nucleolar subcompartments[10] and build on it by adding PCH as an additional phase layered around the nucleolus. We also introduce a dual-affinity linker protein that exhibits differential interaction strengths to nucleolar and PCH phases, that is, Pit's affinity for nucleoli is stronger than for PCH (Fig. 7f(iii)). Unlike the nested subcompartments within the nucleolus (which are immiscible but associate due to sequential rRNA synthesis, processing and ribosome assembly)[10–12,66], PCH–nucleolar associations are independent of rRNA transcription and rely instead on weak physical interactions between Pit and HP1a. Although engineered 'amphiphiles' have been shown to generate multiphasic condensates in vitro[67] and in vivo within stress granules[68], we demonstrate Pit as having a similar function in a natural context with PCH and nucleoli, and propose a generalizable role for other dual-affinity molecules in co-organizing immiscible condensates in cells.

The molecular dissection of Pit in this study revealed a domain-specific contribution for its PCH–nucleolar interactions. The N-terminal disordered domain is required for broad localization to the nucleolus, consistent with the presence of nucleolar localization motifs[52] and aspartic acid/glutamic acid tracts associated with nucleolar localization[12]. The Pit C-terminal PxVxL motif interacts with HP1a dimers and is required for PCH–nucleolar associations. An additional enrichment of Pit within the FC/DFC region is dependent on its helicase activity and nascent rRNA transcription, providing support for a model where distinct Pit domains mediate interactions with specific phases. This behaviour highlights a broader principle that the same molecule can display different distributions depending on the relative interaction strengths of motifs and the availability of binding partners. Under normal conditions, most Pit molecules are predicted to interact with nucleolar components with high affinity, compared to those that interact with HP1a at the PCH–nucleolar interface. It is possible that Pit adopts different dominant conformational states depending on its local environment that promote specific interactions, as suggested for two different phase-separating proteins[69]. Importantly, modulation of these interaction strengths alters the localization of Pit within nucleoli and higher-order condensate architecture. For example, loss of Pit–rRNA binding increases the availability of Pit to interact with HP1a, leading to its enrichment only at the nucleolar edge where Pit–HP1a interactions can occur. Reciprocally, when the Pit HP1a-interacting PxVxL motif is mutated, nucleolar–PCH proximity is lost and Pit loses its enrichment at the nucleolar edge, demonstrating that a balance of interactions dictates 3D condensate organization (Fig. 7f(iv),(v)). Pit belongs to a family of DEAD-box ATPases, many of which remodel condensates by tuning interaction networks[50], a function consistent with our observations for Pit. Although our data show that Pit and its HP1a interactions are required for PCH organization around the nucleolus, we do not exclude the role of other proteins in contributing to this organization. Depletion of the Nucleoplasmin homologue NLP, CTCF or Mod in *Drosophila* cultured cells[70] or NPM1 in mammalian cell lines[71] indeed causes the loss of heterochromatin clustering around the nucleolar periphery. Given the requirement for Pit and its HP1a interaction, it is probable that these and other proteins are required to form an intact GC, which in turn recruits Pit and potentially other partners that together contribute to the overall affinity of PCH for the nucleolar edge.

Cellular condensates are composed of a large network of interacting molecules and their compositions are determined by the multivalency and binding affinities of constituent molecules[5]. Our findings demonstrate how perturbation of a nucleation site or specific interaction domains can reorganize the entire condensate by redistributing phases or dissociating its components. These rearrangements are not random and reflect the interaction preferences of individual molecules. For example, deletion of rDNA leads to nucleolar loss and compaction of

PCH, resembling a collapsed polymer-like state, probably due to the loss of competing nucleolar surface associations and increased homotypic HP1a interactions. Despite the lack of rDNA/rRNA, nucleolar proteins such as Fib and Pit still form dense phase-separated assemblies or neocondensates but do not associate with each other, highlighting the role of rRNA synthesis in promoting interactions among nucleolar phases. Fib forms a separate spherical neocondensate when deprived of its processing substrate rRNA, driven by strong self-association[40,44]. Pit also forms a spherical domain in the absence of rDNA due to strong self-association but, in contrast, this neocondensate resides within the compacted PCH due to its weak HP1a interaction (Fig. 7f(ii)). However, when rDNA is removed, Mod is broadly dispersed throughout the nuclear volume, probably due to a weak self-association and affinity for other nuclear structures. Given the large number of multicomponent condensates in cells, it is crucial to assess whether new condensates or other potentially impactful redistributions of molecular components arise when perturbing interactions (for example, by mutating a protein-binding domain). These neomorphic structures may also be developmentally regulated, for example, we note the similarity between the PCH voids observed in −rDNA *Drosophila* embryos and nucleolus precursor bodies observed in mammalian oocytes and early embryos[72] before the initiation of rRNA synthesis. Finally, understanding these outcomes will be important during stress responses, ageing and cellular senescence, when new condensates often form or the composition of existing condensates changes[73].

## Online content

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

**Srivarsha Rajshekar** [1], **Omar Adame-Arana**[2,3], **Gaurav Bajpai**[2,4], **Serafin U. Colmenares**[1], **Hannah Papoi**[1], **Lucy D. Brennan**[1], **Shingo Tsukamoto**[5], **Samuel Safran** [2] **& Gary H. Karpen** [1,6] ✉

[1]Department of Molecular and Cell Biology, University of California, Berkeley, CA, USA. [2]Department of Chemical and Biological Physics, Weizmann Institute of Science, Rehovat, Israel. [3]Max Planck Institute for the Physics of Complex Systems, Dresden, Germany. [4]Department of Physics, Northeastern University, Boston, MA, USA. [5]Departments of Bioengineering and Mechanical Engineering, University of California, Berkeley, CA, USA. [6]Division of Biological Sciences and the Environment, Lawrence Berkeley National Laboratory, Berkeley, CA, USA. ✉e-mail: gkarpen@berkeley.edu

## Methods

### *Drosophila* stocks and genetics

All crosses were maintained at 25 °C. To visualize the dynamics of HP1a and nucleolar proteins, live embryos were imaged from the following transgenic stocks: *RFP–HP1a; GFP–Fib, RFP–Fib; GFP–HP1a, RFP–HP1a; GFP–Mod, RFP–HP1a; Pit–GFP, RFP–HP1a; GFP–Nopp140* and *RFP–HP1a; GFP–Ns1*. Embryos lacking rDNA were obtained as previously described[40] by crossing C(1)DX/Y; *RFP–HP1a; GFP–Fib* or C(1)DX/Y; *RFP–HP1a; Pit–GFP* virgin females to C(1;Y)6,y[1]w[1]f[1]/0 males. One-quarter of the resulting embryos from this cross lacked rDNA, and the −rDNA embryos were selected based on the presence of Fib neocondensates in live and fixed embryos, and DAPI morphology in fixed embryos. For maternal depletion of Pit, mat-α-GAL4 drivers were crossed to Pit RNAi lines (VAL20 and VAL22) and ovaries were dissected from F1 females. To knockdown Pit in eye discs, eyeless-GAL4 virgin females were crossed with Pit RNAi VAL20 males, and eye discs were dissected from F1 third-instar larvae. All genetic crosses were set up with at least ten females and ten males per cage or vial.

The different fly genoptypes were obtained from the following sources: RFP–HP1a (chromosome 2)[74] and GFP–HP1a (chromosome 3)[22], Karpen laboratory; eGFP–Fib (chromosome 3)[47], RFP–Fib (chromosome 2)[40], eGFP–Mod[47], Pit-eGFP[47], eGFP–Nopp140[47] and eGFP–Ns1[47], Weischaus laboratory; and FM6/C(1)DX,y[*]f[1], (stock number 784), C(1)RM/C(1;Y)6,y[1]w[1]f[1]/0, (stock number 9460), mat-α-GAL4, (stock number 7063), eyeless-GAL4, (stock number 5534) Pit RNAi VAL20 (stock number 80368) and Pit RNAi VAL22 (stock number 43984) from the Bloomington Drosophila Stock Center.

### Preparation of *Drosophila* embryos for live imaging

Adult flies of the desired genotype were placed in a plastic cage with apple juice agar plates and maintained for at least 3 days at 25 °C. On the day of imaging, a fresh plate was added and embryos were developed to capture the desired developmental stage/mitotic cycles based on standardized *Drosophila* embryo Bownes stages[75]. Embryos were placed on a paper towel square (5 cm × 5 cm) and dechorionated in 50% bleach for 1 min. The bleach was wicked off with a paper towel after 1 min, the square was washed with a small amount of distilled water, and excess water was wicked off. Dechorionated embryos were secured onto a semipermeable membrane (Lumox film, Sarstedt) on a membrane slide holder using heptane glue. These embryos were mounted in Halocarbon oil 27 (Sigma) between the membrane and a coverslip.

### Immunostaining

Embryos were collected on apple juice agar plates, aged until the appropriate stage, dechorionated in 50% bleach, fixed in 1:1 heptane:4% formaldehyde (Sigma) in 1×PBS for 25 min, devitellinized in a 1:1 mixture of methanol:heptane and stored at −20 °C in methanol. The embryos were rehydrated by washing in 1×PBS + 0.2% Triton X-100 (PBT). Dissected eye discs were fixed with 4% formaldehyde in PBS for 20 min, and the fixative was washed off with PBT. Following washes with PBT, the tissues were blocked with 2% BSA in PBT for 1 h and then incubated overnight with the primary antibody at 4 °C. After incubation with the appropriate secondary antibody at room temperature for 2 h, the samples were stained in DAPI and mounted onto a slide using VectaShield (Vector Laboratories) mounting medium. All primary antibodies were used at a dilution of 1:250 (unless mentioned otherwise) and secondary antibodies at 1:1,000 in blocking buffer. The following primary antibodies were used: rabbit anti-Fib (Abcam, ab5821), mouse anti-H3K9me2 (Abcam, ab1220), rabbit anti-H3K9me3 (Abcam, ab8898), mouse anti-Mod (a gift from the Mellone laboratory), mouse anti-Lamin, Dm0 (DSHB, ADL67.10) and guinea pig anti-Mxc (used at 1:1,000; a gift from the Duronio laboratory). The following secondary antibodies were used: goat anti-mouse Alexa Fluor 488 (Invitrogen, A-11001), goat anti-mouse Alexa Fluor 568 (Invitrogen, A-11004), goat anti-rabbit Alexa Fluor 488 (Invitrogen, A-11034), donkey anti-rabbit Alexa Fluor 568 (Invitrogen, A-10042) and goat anti-guinea pig Alexa Fluor 647 (Invitrogen-A-21450).

### DNA fluorescent in situ hybridization and immuno-fluorescent in situ hybridization

**Probe labelling.** A probe for ITS-1 rDNA was prepared by PCR amplifying a 704-bp fragment from genomic DNA using the primers 5′-ACGGTTGTTTCGCAAAAGTT-3′ and 5′-TGTTGCGAAATGTCTT AGTTTCA-3′, cloned into a pGEM T-Easy vector (Promega) and used as a template for probe synthesis. The ITS-1 rDNA probe was labelled with Alexa 488, Alexa 555 or Alexa 648 using the FISH tag DNA multicolor kit (Thermo Fisher) following the manufacturer's protocol. The following bridged nucleic acid oligonucleotides (Integrated DNA technologies) conjugated to Cyanine5 or 6-Carboxyfluorescein were used as probes for 359 bp, 1.686 and AAGAG satellite DNA repeats: 359 bp, 5′-CAGATATTCGTACATCTATG-3′; 1.686, 5′-CAATAGA CAATAGACAATAG-3′; and AAGAG, 5′-GAAGAGAAGAGAAGAGAAGA-3′.

**Hybridization.** Fixed embryos were washed (15 min each wash) in 2×SSC-T (2×SSC containing 0.1% Tween-20) containing increasing formamide concentrations (20, 40 and 50%). Next, 100 ng DNA probe in 40 μl hybridization solution (50% formamide, 3×SSC-T and 10% dextran sulfate) was added, denatured together with the embryos at 95 °C for 5 min and incubated overnight at 37 °C. Following hybridization, the embryos were washed twice (30 min each wash) in 2×SSC-T at 37 °C and thrice in PBT (5 min each wash) at room temperature. After completing the washes, the embryos were stained in DAPI and mounted onto a slide using VectaShield mounting medium.

**Immunofluorescence with fluorescent in situ hybridization.** Immunofluorescence was performed first on embryos for combined in situ detection of proteins and DNA sequences. Embryos were post-fixed in 4% formaldehyde for 25 min and then processed for FISH.

### Pan-protein staining using 488 NHS ester

Formaldehyde-fixed embryos were devitellinized in a 1:1 mixture of methanol:heptane and stored in methanol. The embryos were rehydrated by washing in PBT. After washing off the methanol, the embryos were stained in diluted (1:50) Atto 488 NHS ester fluorophore (Sigma) from a 10 mg ml⁻¹ stock in 0.1% PBST for 6 h at 4 °C, followed by three washes in PBT (30 min each) at room temperature. The embryos were stained in DAPI for 10 min and mounted onto a slide using VectaShield mounting medium.

### Propidium iodide staining

Formaldehyde-fixed and heptane-devitellinized embryos were rehydrated by washing in PBT. The samples were equilibrated in 2×SSC. RNase-treated controls alone were incubated in 100 μg ml⁻¹ DNase-free RNase in 2×SSC for 20 min at 37 °C. After washing away the RNase with 2×SSC, the embryos were incubated in 500 nM propidium iodide (Invitrogen) in 2×SSC for 10 min at room temperature. The samples were rinsed in 2×SSC, stained with DAPI and mounted on a slide with VectaShield mounting medium.

### Microscopy

Imaging was performed on a Zeiss LSM880 AiryScan microscope (Airy Fast mode) with a ×63 1.4 numerical aperture oil-immersion objective at room temperature. Depending on the fluorophore, 405, 488, 514 or 633 nm laser lines were used for excitation with the appropriate filter sets. Laser intensity values, detector gain, image size, zoom, *z*-stack intervals and time intervals (for time-lapse acquisitions) were adjusted to minimize bleaching and ensure uniform detection across all AiryScan detection elements. Once standardized for an experiment, the settings were kept identical across all samples in the experimental groups. Raw images were processed using the Zeiss ZEN Black software with

the AiryScan processing module for reconstruction and subsequent image analysis.

## Modelling

To better understand the association of PCH with the nucleolus, we developed a physical model that simulates the interactions between different types of molecules found in these biomolecular condensates. In our physical model, we simulated four components of the nucleus: PCH and rDNA as long polymers, and Fib and a dual-affinity protein as independent single monomers, which we hereafter refer to as beads. As the experiments focused on the PCH domain of *Drosophila*, our physical model only simulated this specific region of the genome (30% of the genome), not the entire genome. This allowed us to study the dynamics of this particular region of the genome more accurately and efficiently. PCH is modelled as a semiflexible bead-spring polymer chain, where $N$ beads are connected by $N-1$ harmonic bonds. Each bead of the chain represents a cluster of PCH containing approximately 5 kilobase (kb) pairs of DNA, with an effective diameter of $\approx 30$ nm. The semiflexibility of the chain is determined by its persistence length, which is taken to be 60 nm (two beads) in accordance with previous studies that indicate chromatin has a persistence length of 50–100 nm (ref. [76]). We represent rDNA as a self-avoiding chain that occupies approximately 20% of the middle domain of PCH. Fib and the dual-affinity protein X are modelled as individual diffusive beads. Each protein has specific interaction strengths with both chromatin and other proteins.

To simplify the model, we assume that the size of each protein bead is equal to the size of the heterochromatin bead, with both having a diameter $\sigma$. The non-bonding polymer–polymer, protein–protein and polymer–protein interactions are modelled using a standard Lennard–Jones (LJ) potential. The LJ potential is truncated at $2.5\sigma$, so interactions are considered only between beads separated by a distance smaller than this. At short distances ($r_{ij} \leq 2^{1/6}\sigma$), the LJ potential is strongly repulsive, representing excluded-volume effects. At intermediate separations ($2^{1/6}\sigma < r_{ij} < 2.5\sigma$), the potential becomes attractive, with a strength ($\epsilon$) tuned to represent different interaction strengths—such as chromatin compaction, chromatin–protein affinity or phase separation between protein species. All components are confined within a spherical boundary that represents the nucleus. This boundary mimics the effect of the nuclear envelope, which constrains the movement of these beads and affects their interactions with each other. In our study, we used the Large-scale Atomic/Molecular Massively Parallel Simulator (LAMMPS) package to simulate the behaviour of our biomolecular system[77]. LAMMPS uses Brownian dynamics, which accounts for the viscous forces acting on the beads, and a stochastic force (Langevin thermostat) to ensure that the system of beads and solvent is maintained at a constant temperature. Each simulation is performed in the canonical (NVT) ensemble by maintaining the system volume and the number of proteins constant. This allows us to model the polymer–polymer, polymer–protein and protein–protein bead interactions accurately and study the behaviour of the system over time.

### Rationale for the choice of parameters in the coarse-grained model

To analyse the experimental observations of phase separation in the nucleus, we studied a minimal model with four crucial components: PCH, rDNA, Fib and a dual-affinity protein (X) that binds both nucleolar and PCH components. The parameters in the simulations included the number of molecules of each component $\alpha$ ($N_\alpha$), the strength of the attractions between two components (denoted by indices $\beta$ and $\gamma$, $\epsilon_{\beta\gamma}$), and the size of the confinement ($R_c$). The fraction of the nucleus that is hydrated (does not contain PCH or the other proteins) is obtained from the relative difference between the confinement volume and the volumes of PCH and the other proteins.

**Bonding potential between monomers in the polymer made of PCH and rDNA.** Adjacent beads on the polymer chain are interconnected by harmonic springs using the potential function:

$$V_s = \sum_{i=1}^{N-1} k_s(r_i - \sigma)^2$$

Here $r_i$ represents the distance between the $i$-th and $(i+1)$-th beads. The spring constant and equilibrium distance between neighbouring beads are denoted as $k_s$ and $\sigma$ respectively. In our simulations, the spring constant $k_s$ is set to $\frac{100\,k_B T}{\sigma^2}$ to ensure the presence of rigid bonds between adjacent beads of the polymer chain.

**Attraction strength.** The LJ potential is used to model the attraction between any two non-bonded beads:

$$V_{\beta\gamma}(r) = \sum_{i<j} 4\epsilon_{\beta\gamma}\left[\left(\frac{\sigma}{r_{ij}}\right)^{12} - \left(\frac{\sigma}{r_{ij}}\right)^{6}\right] \text{ for } r_{ij} \leq r_c \text{ and } 0 \text{ for } r_{ij} > r_c$$

Here the symbol $\epsilon_{\beta\gamma}$ represents the attraction strength between beads of type $\beta$ and $\gamma$, where $\beta, \gamma \in \{H, rD, F, X\}$ represent PCH, rDNA, Fib and dual-affinity protein, respectively. For instance, $\varepsilon_{F-H}$ represents the attraction strength between Fib and the dual-affinity protein beads. When dealing with attractive chromatin–chromatin and chromatin–protein bead interactions, a distance cutoff of $r_c = 2.5\sigma$ is used for the LJ potential, beyond which the interaction is set to zero. To account for only excluded-volume interactions (with no attractions) using the same potential, a cutoff of $r_c = 2^{1/6}\sigma$ and $\epsilon = 1\,k_B T$ are employed. This choice is made because the potential energy is at its minimum at that point and the resulting force on a bead is zero. Note that in the interaction matrix, setting the attractive strength to zero implies that only excluded-volume interactions are present between those types of beads. As there are four components in our model, there are a total of ten combinations ($(m(m+1)/2)$, where $m$ is the number of components) of binary attraction strength parameters between the different components.

**Confinement size.** The size of the confinement is determined by defining the volume fraction of chromatin ϕ:

$$\phi = N_G \times \frac{\text{volume of one bead}}{\text{volume of confinement}} = N_G \times \frac{\left(\frac{4}{3}\pi\sigma^3\right)}{\left(\frac{4}{3}\pi R_c^3\right)}$$

Here, $N_G$ represents the total number of beads in the *Drosophila* genome, where each bead corresponds to 5 kb pairs of DNA. The diameter of a spherical bead, denoted as $\sigma$, is taken to be 30 nm (further explanation in ref. [78]). The total length of the diploid *Drosophila* genome is 360 Mbps, and $N_G$ can be calculated by dividing the *Drosophila* genome length by the amount of DNA represented by one bead (5 kb pairs). The volume fraction ϕ is commonly reported as approximately 0.1 in existing literature[79,80]. Using the equation above, the calculated radius of the confinement ($R_c$) is found to be $45\sigma$.

**Number of molecules in PCH and rDNA.** We modelled the PCH domain as a polymer chain composed of $N = 10,000$ beads, which represents approximately 30% of the beads in the entire genome. Among these 10,000 beads, 20% are designated as rDNA, resulting in a total of 2,000 rDNA beads. We do not explicitly model the rest of the genome as the experiments showed that the nucleolar components are localized near PCH.

**Concentration of Fib.** The concentration of Fib and dual-affinity protein was calculated as follows:

$$\text{concentration}\,(c_\alpha) = \frac{N_\alpha}{\text{volume of confinement}}$$

Here α represents the type of protein, where $c_\alpha = c_F$ for the Fib concentration and $c_\alpha = c_X$ for the dual-affinity protein concentration. After defining the parameters and obtaining the value for the confinement radius parameter from the assumed volume fraction of PCH, we proceed with an initial simulation, focusing on a single component, namely the Fib protein, within the confinement. During this simulation, we vary the concentration ($c_F$) and the attraction strength ($\varepsilon_{F-F}$) of Fib. Fib undergoes phase separation independently of the other components (rDNA or PCH) during cycle 14 (Fig. 2a). These initial-stage simulation results yield a phase diagram, which indicates that a minimum attraction strength of 1.3–2.0 $k_B T$ is required to condense Fib particles within the $c_F$ concentration range of 0.0013–0.0130. The reported concentration of nucleolar particles is 0.015 (ref. 79). Consequently, we maintain a fixed $c_F$ of 0.013 when simulating all the other protein components (Extended Data Fig. 4a).

**Parameter range of $\varepsilon_{H-H}$.** To understand the behaviour of each component separately (before including interactions between different components) in the next stage, we conducted simulations specifically focusing only on the PCH chain. The PCH chain represents a condensed chromatin region in the nucleus, which implies self-attractive interactions of the beads representing the polymer[81]. During these simulations, we varied the attraction strength between PCH beads ($\varepsilon_{H-H}$) from zero to 0.5 $k_B T$. Our results revealed that within the $\varepsilon_{H-H}$ range of 0.35–0.50 $k_B T$ (Extended Data Fig. 4b), the PCH chain underwent collapse, resulting in a condensed conformation that is phase-separated from the aqueous component of the system (not simulated explicitly).

**Parameter value of $\varepsilon_{rD-F}$.** We next simulated the self-organization due to the interactions between the three components (where each component so far was considered alone): PCH, rDNA and Fib. We set $\varepsilon_{H-H} = 0.35\,k_B T$ and $\varepsilon_{F-F} = 2\,k_B T$, incorporating only excluded-volume H–F interactions (hard-core, repulsive interactions), that is, there is no direct attraction between PCH and Fib, as implied by the experiments. By varying the attraction strength between rDNA and Fib ($\varepsilon_{rD-F}$), we made the following observations based on our simulation results: a weaker attraction ($\varepsilon_{rD-F} = 0.75\,k_B T$) resulted in rDNA wrapping around the surface of a Fib condensate, whereas a stronger attraction (and $\varepsilon_{rD-F} = 2\,k_B T$) led to the condensation of rDNA within the Fib condensate. This latter observation aligns with our experimental findings. Therefore, we selected $\varepsilon_{rD-F} = 2\,k_B T$ as the parameter value in the subsequent simulations (Extended Data Fig. 4b).

**Parameter ranges of $\varepsilon_{F-F}$ and $\varepsilon_{X-X}$.** Finally, we introduced the fourth component, a dual-affinity protein X, which we suggest may interact attractively with both PCH and Fib. Initially, we investigated the relative attraction strengths between Fib and protein X when considering the same concentration for both. We explored all possible combinations of $\varepsilon_{F-F}$ both greater than and less than $\varepsilon_{X-X}$. For $\varepsilon_{X-X} \geq \varepsilon_{F-F}$, we observed PCH surrounding the dual-affinity protein-rich phase but just a partial wetting between the dual-affinity protein-rich phase and the Fib-rich phase (Extended Data Fig. 4c). Only when $\varepsilon_{X-X} < \varepsilon_{F-F}$ did we observe PCH surrounding the dual-affinity protein-rich phase, which in turn surrounded the Fib condensate, consistent with the experimental results in WT embryos.

**Parameter range of $\varepsilon_{F-X}$.** We proceeded to vary the attraction strength between Fib and dual-affinity protein X within the range of $\varepsilon_{F-X} = 1.5\,k_B T$. When the attraction strength is relatively low ($\varepsilon_{F-X} \leq 0.75\,k_B T$), the Fib-rich phase and the dual-affinity protein-rich phase do not associate with each other. At moderate attraction strengths ($\varepsilon_{F-X} \geq 1\,k_B T$ and $\varepsilon_{F-X} \leq 1.25\,k_B T$), the Fib-rich and dual-affinity protein-rich phases partially wet each other. Finally, at higher attraction strengths ($\varepsilon_{F-X} \geq 1.5\,k_B T$), the dual-affinity protein completely wets Fib (Extended Data Figs. 4d,e and 8c).

**Parameter range of $c_X$ and $c_P$.** As the concentration of dual-affinity protein was not determined from the experiments, we conducted multiple simulations, systematically varying the concentration of the dual-affinity protein concentration ($c_X$) or Pit ($c_P$) within the range of 0.00013–0.05240. We set normal $c_P$ levels to be the same as $c_F = 0.0131$ for the simulations of the 'surrounding' configuration where heterochromatin completely engulfs the dual-affinity protein-rich phase, which in turn engulfs Fib. When we decreased to $c_P = 0.000131$, heterochromatin did not engulf Fib (Fig. 6a).

**Parameter range for $\varepsilon_{rD-P}$.** We determined through experiments that Pit also localizes in the nucleolar core and is not just restricted to the nucleolar edge. To model this behaviour, we included a new interaction between rDNA and Pit to develop a refined model. We varied $\varepsilon_{rD-P}$, within the range 0–2.2 $k_B T$, and observed that as long as $\varepsilon_{rD-P} > \varepsilon_{F-P}$, Pit localizes within the Fib-rich protein phase (Extended Data Fig. 8a,b).

**Radial distribution calculation.** To quantify the spatial distribution of Pit in the nucleolus, we computed its radial distribution relative to the Fib condensate, which occupies the interior of the nucleolus. First, we calculated the centre of mass of all Fib beads, representing the core of the nucleolar condensate. Using this centre of mass as the reference point, we measured the distances of all Pit beads from the centre of mass and constructed a normalized radial density profile. This analysis shows how Pit is distributed radially within the nucleolar environment (Fig. 7b and Extended Data Fig. 8b).

## Cell culture

*Drosophila* S2R+ cells (*Drosophila* Genomics Resource Center, stock number 150) were cultured in Schneider's *Drosophila* Medium (Gibco) with 10% fetal bovine serum and 1% antibiotic–antimycotic (Gibco) at 25 °C. For transfections, the cells were seeded at $5 \times 10^5$ cells ml⁻¹ in six-well plates 24 h beforehand. Plasmid DNA (1 µg) was diluted in 100 µl serum-free medium and mixed with 2 µl TransIT-2020 transfection reagent (Mirus Bio). After a 15-min incubation, the DNA-reagent complexes were added to the cells and incubated at 25 °C for 48–72 h before visualization.

## Plasmids and recombinant DNA

Codon-optimized gene blocks for *Pit* (FlyBase identifier: FBgn0266581), *Fib* (FBgn0003062), *Mod* (FBgn0002780), *Polr1E* (FBgn0038601) and *HP1a* (FBgn0003607) were synthesized by Twist Biosciences and cloned into pCOPIA vectors fused with fluorescent protein tags. Truncated constructs *Pit^N*, *Pit^Hel* and *Pit^C* were similarly synthesized and cloned with a *c-Myc* nuclear localization signal tag. Site-directed mutagenesis was performed to introduce PxExL and DQVD mutations into the full-length Pit using the Q5 Site-Directed Mutagenesis Kit (New England Biolabs) and the following primer pairs: Pit^PxExL_F, 5′-CCGGTAGAAGATCTCAAAGAAGGAGCTGCTAAGCG-3′ and Pit^PxExL_R, 5′-CGGTACCAGGAAGCCGAAAC-3′; and Pit^DQVD_F, 5′-CCAAGTCGACAGGATCCTGG-3′ and Pit^DQVD_R, 5′-TCGATGATGAGGCACTGCAA-3′.

## APEX2 proximity ligation

Stable S2R+ cell lines expressing V5-APEX2–Pit were generated using puromycin selection. The cells were incubated in 2.5 mM biotin-phenol (APExBIO) in S2 Cell medium for 30 min, followed by 1 mM $H_2O_2$ for 1 min to initiate biotinylation. The reaction was quenched using STOP buffer (0.5 mM MgCl₂, 1 mM CaCl₂, 5 mM Trolox, 10 mM sodium ascorbate and 10 mM sodium azide). The cells were then lysed and resuspended in modified RIPA buffer (RIPA without NaCl), followed by benzonase treatment for 30 min to digest nucleic acids. Nuclear proteins were then extracted by adding 1 M NaCl and incubating for 1 h at 4 °C. Supernatant containing the solubilized proteins was diluted to 0.15 M NaCl and added to Streptavidin magnetic beads overnight. The beads were washed successively with RIPA buffer (50 mM Tris pH 8.0,

150 mM NaCl, 5 mM EDTA, 0.5% sodium deoxycholate, 0.1% SDS and 1% Triton X-100), 1 M KCl, 0.1 M Na$_2$CO$_3$ and 2 M urea in 10 mM Tris–HCl pH 8. After washing, biotinylated proteins bound to the Streptavidin beads were eluted by boiling in Laemmli sample buffer for 2 min and analysed by western blot. Biotinylated proteins were detected using Streptavidin-IRDye 700 (1:5,000; LI-COR). HP1a was probed using antibody to C1A9 (1:100; DSHB), followed by LI-COR secondary antibodies (1:5,000) or IRDye 800CW Streptavidin conjugate (1:5,000) and visualized using the LI-COR Odyssey imaging system.

## Custom generation of anti-Pit

A custom monospecific antibody to a peptide corresponding to Pit (28–46): Cys–NKNAQKQEPPKQNGNKPSK (20 amino acids) was generated by Pacific Immunology. The peptide was synthesized and conjugated to a keyhole limpet haemocyanin carrier protein before immunization in rabbits. Sera were affinity purified against the peptide antigen. The antibody was validated by western blotting of nuclear extracts from *Drosophila* S2R+ cells; a band at the expected molecular weight of 76.9 kDa was detected. Antibody specificity was further confirmed via detection of recombinant Pit expressed in *Escherichia coli* at the expected size.

## Co-immunoprecipitation

S2R+ cells ($2 \times 10^8$) were harvested, pelleted by centrifugation and washed with S2 Cell Wash Buffer (10 mM HEPES pH 7.5 and 140 mM NaCl). The cells were lysed in Nuclei Isolation Buffer (15 mM HEPES pH 7.5, 10 mM KCl, 5 mM MgCl$_2$, 0.1 mM EDTA, 0.5 mM EGTA and 350 mM sucrose) and nuclei were pelleted by centrifugation. The nuclear pellet was resuspended in Extraction Buffer (20 mM HEPES pH 7.5, 10% glycerol, 1 mM MgCl$_2$, 0.1% Triton X-100, 100 μM phenylmethylsulfonyl fluoride and protease inhibitors) to prepare the nuclear extract. Benzonase was added to the samples, which were then incubated for 30 min to digest nucleic acids. Next, NaCl was added to the samples (final concentration of 350 mM) for 1 h at 4 °C with rotation to extract nuclear proteins. The lysate was centrifuged to remove insoluble material and the resulting supernatant was diluted to 150 mM NaCl, which served as the input nuclear lysate.

For immunoprecipitation, the nuclear lysate was incubated with 4.5 μg anti-Pit (custom-made by Pacific Immunology) or control IgG (Invitrogen, 02-6102) for 1 h at 4 °C. Protein G magnetic beads (Invitrogen, 10003D) were then added and the mixture was incubated overnight at 4 °C with rotation. After incubation, the flow-through was collected and the beads were washed with cold Wash Buffer (20 mM HEPES pH 7.5, 10% glycerol, 150 mM NaCl, 1 mM MgCl$_2$ and 0.1% Triton X-100) to remove non-specific interactions. Immunoprecipitated complexes were eluted by heating the beads in Laemmli sample buffer at 65 °C for 5 min. Input, flow-through and co-immunoprecipitation samples were analysed by western blotting, probing with antibodies to HP1a (1:100) and Pit (1:5,000) to assess protein interactions.

## Protein sequence alignment

Multiple sequence alignments of full-length homologues of Pit were conducted using Clustal Omega[82]. The GenBank accession numbers for homologues used for alignment are: NP_524446.3 (*D. melanogaster*), XP_016035699.1 (*Drosophila simulans*), XP_002032147.2 (*Drosophila sechellia*), XP_002098237.1 (*Drosophila yakuba*), XP_001982328.1 (*Drosophila erecta*), XP_015027080.1 (*Drosophila virilis*), XP_032585543.1 (*Drosophila mojavensis*), XP_026842155.1 (*Drosophila persimilis*), XP_002137552.2 (*Drosophila pseudoobscura*), XP_001954024.1 (*Drosophila ananassae*), XP_032596000.1 (*Drosophila grimshawi*), XP_002073690.1 (*Drosophila willistoni*), NP_001003411.1 (*Danio rerio*), NP_006764.3 (*Homo sapiens*), NP_080136.2 (*Mus musculus*), NP_492779.1 (*C. elegans*), NP_594488.1 (*Schizosaccharomyces pombe*), NP_014017.1 (*Saccharomyces cerevisiae*) and NP_200302.1 (*Arabidopsis thaliana*).

## AlphaFold-Multimer structure prediction

AlphaFold-Multimer[57] was employed to predict the structure of a 25-amino-acid peptide centred around the PxVxL motif of Pit in its C-terminal disordered domain and an HP1a–CSD dimer with C-terminal tails. Input for the *Drosophila* HP1a–CSD dimer and Pit peptide was TGFDRGLEAEKILGASDNNGRLTFLIQFKGVDQAEMVPSSVANEKIPRMVI-HFYEERLSWYSDNED (UniProt ID: P05205) and VAKSFGFLVPPVVDLKV-GAAKRERP, respectively; 'pdb100' was used for the template mode. The model type was set to 'alphafold2_mmultimer_v3'. The highest-ranked model based on pLDDT (predicted local distance difference test) scores was visualized using the VMD software[83].

## Protein expression and purification

*E. coli* codon optimized gene block for Pit was synthesized by Twist Biosciences and cloned into pBH4 expression vector (Addgene, catalogue number 243658) with an N-terminal 6×His tag. Site-directed mutagenesis was performed to introduce PxExL mutations into pBH4-Pit using a Q5 site-directed mutagenesis kit (New England Biolabs) and the following oligonucleotides: Pit_PxExL_Ecol_F, 5′-CCGGTAGAAGATCTGAAAGAAGGCGCCGCTAAACG-3′ and Pit_PxExL_Ecol_R, 5′-CGGTACCAGGAAGCCGAAAC-3′.

Chemically competent *E. coli* BL21 cells were transformed with plasmids encoding the recombinant proteins and selected using ampicillin and chloramphenicol. A starter culture was cultured overnight in Luria–Bertani medium at 37 °C and then diluted 1:100 into fresh medium the following day. The culture was incubated at 37 °C until it reached an optical density at 600 nm of 0.6 and protein expression was induced with 0.2 mM isopropyl β-D-1-thiogalactopyranoside. After induction, the culture was incubated overnight at 18 °C. The cells were harvested by centrifugation and resuspended in 30 ml lysis buffer (500 mM NaCl, 25 mM Tris–HCl pH 7.5, 10 mM imidazole, 10% glycerol, 1 mM phenylmethylsulfonyl fluoride and protease inhibitors) for every 1 l of culture. Cell lysis was performed using sonication and the lysate was clarified by high-speed centrifugation. Affinity chromatography using Ni-NTA resin (Takara Bio) was used to purify 6×His-tagged proteins from the clarified lysate. Further purification was performed via size-exclusion chromatography using a Superdex 200 column on an AKTA purifier system (GE Life Sciences) in the final storage buffer (200 mM NaCl, 25 mM Tris–HCl pH 7.5 and 2 mM dithiothreitol). Protein expression levels, fractions from affinity purification and size-exclusion chromatography eluates were analysed using SDS–PAGE and western blotting. The purified proteins were pooled and concentrated to 0.5 mg ml$^{-1}$ using Millipore Amicon centrifugal filter units. Aliquots (30 μl) were snap-frozen in liquid nitrogen and stored at −80 °C. HP1a$^{WT}$ and HP1a$^{191E}$ were purified as described by Brennan and colleagues[21].

## His-tag pull-down binding assay

Direct interactions between Pit and HP1a were assessed using a His-tag pull-down assay using purified recombinant proteins. His-tagged Pit (Pit$^{WT}$ or Pit$^{PxExL}$; 5 μM) was incubated with 5 μM purified HP1a in binding buffer (20 mM sodium phosphate, 50 mM NaCl and 10 mM imidazole) at 4 °C for 1 h with gentle rotation to allow complex formation. Ni-NTA magnetic beads (Invitrogen, 10103D) were added to the reaction to capture Pit-bound complexes and incubated for an additional hour. These incubation conditions were optimized by titrating HP1a$^{WT}$ concentrations from 50 μm to 5 μM to reveal specific binding above background levels observed with beads + HP1a-only controls. The beads were washed three times with binding buffer and eluted by boiling in 1×Laemmli SDS Buffer. Inputs and eluted fractions were analysed by SDS–PAGE, followed by western blotting using anti-Pit and anti-HP1a.

## Pit RNAi

To generate the amplicon for Pit RNAi targeting its 3′ untranslated region, genomic DNA from S2R+ cells was used as the template for

PCR amplification with the following primer pairs: T7-Pit-RNAi-F, 5′-TAATACGACTCACTATAGGgctgctttacttgagtgtgtgt-3′ and T7-Pit-RNAi-R, 5′-TAATACGACTCACTATAGGccaaggtggcccgcaattat-3′; and T7-Mock-RNAi-F, 5′-TAATACGACTCACTATAGGgaaaaactaagccaacgtcatc-3′ and T7-Mock-RNAi-R, 5′-TAATACGACTCACTATAGGgccgtggatataggcaaaaa-3′. The T7 promoter sequence is capitalized in the primers. Mock RNAi targeting the y gene was used as a negative control.

A Hi-Scribe T7 synthesis kit (New England Biolabs) was used to synthesize double-stranded RNA according to the manufacturer's protocol. Following synthesis, RNA purification was carried out utilizing a MinElute RNeasy kit (Qiagen). The purified RNA was then diluted to 1 µg µl⁻¹. For the RNAi experiment, 3–5 µg double-stranded RNA with DOTAP liposomal transfection reagent (Roche) was used per $0.5 \times 10^6$ cells. The cells were analysed 5 or 6 days after the initiation of RNAi.

Total RNA from S2R+ cells was extracted by homogenizing the cells in TRIzol reagent (Invitrogen). Next, a 0.5× volume of chloroform was added to the sample, the mixture was shaken for 15 s, incubated for 3 min and centrifuged at 12,000$g$ and 4 °C for 15 min. The aqueous phase was transferred to a new tube, mixed with 500 µl isopropanol, incubated for 10 min and centrifuged at 12,000$g$ and 4 °C for 10 min. The RNA pellet was washed with 75% ethanol, centrifuged at 7,500$g$ and 4 °C for 5 min, air-dried for 5–10 min and resuspended in RNase-free water. The total RNA was treated with a DNA-free DNA removal kit (Invitrogen), as per the manufacturer's protocol, to remove any contaminating genomic DNA. The RNA was converted to complementary DNA using a GoScript reverse transcriptase kit (Promega) and random primers, and real-time PCR was performed using PerfeCTa SYB Green FastMix (Quantabio) on a BioRad CFX96 real-time PCR detection system. Analysis was performed using the $2^{-\Delta\Delta Ct}$ method, with relative messenger RNA levels of Pit normalized to those of $\beta$-actin.

### Actinomycin D treatment to inhibit rRNA synthesis

S2R+ cells transfected with fluorescently tagged markers of the nucleolus and HP1a were treated with ActD (Gibco, catalogue number 11805017) at a final concentration of 0.08 µg ml⁻¹ for 10 min before imaging. ActD was prepared as a 10 mg ml⁻¹ stock in dimethylsulfoxide, stored at −20 °C and diluted to the working concentration in culture media immediately before use. Control cells were treated with an equal volume of dimethylsulfoxide as the vehicle. The inhibition of nascent rRNA synthesis was confirmed using a Click-iT RNA Alexa Fluor 488 imaging kit (Invitrogen, C10329) as per the manufacturer's protocol.

### Quantitative image analysis

An Arivis Vision4D (Zeiss) system was used to perform 3D measurements and ImageJ was used for all 2D measurements. The details of each analysis pipeline used in this study follow.

**Measurement of HP1a occupancy around the nucleolus.** Images were pre-processed by background subtraction and Gaussian Blur denoising. HP1a and Fib (nucleolar marker) signals were segmented in 3D using the Intensity Threshold Segmenter tool in Arivis. To define the nucleolar edge, each nucleolus was uniformly dilated (by 1–4 pixels) across experimental groups and the original object was subtracted from its dilated version to generate a thin shell representing the nucleolar edge. This nucleolar shell was intersected with HP1a segments to calculate the volume fraction of the nucleolar shell that overlaps with HP1a, defined as HP1a occupancy around the nucleolus. HP1a distribution relative to the nucleolus in S2R+ cells was also manually classified into four organizational phenotypes—surrounded, extended, clustered or fragmented—as described by representative images in the main text.

**Distance measurement.** Images were background-subtracted and denoised with Gaussian Blur. The Intensity Threshold Segmenter tool was used to autosegment 1.686, AAGAG and Fib foci, and 3D distances between each Fib or AAGAG segment and its nearest 1.686 locus in the same nucleus were measured using the Distances feature in Arivis.

**Measurement of the HP1a aspect ratio.** Individual nuclei were manually selected 15 min after the start of cycle 15. Pre-processing steps included background subtraction and denoising using Gaussian Blur. HP1a was autosegmented using Otsu thresholding. The aspect ratio of the segment was determined using the Analyze Particles feature in Fiji.

**Measurement of Pit neocondensate formation.** To measure the dynamics of the formation of Pit in −rDNA embryos, maximum intensity projections of Amnioserosa were first pre-processed in Fiji using Subtract Background and a Gaussian Blur filter. Auto thresholding for each time point was performed using the Yen method. Using Analyze Particles, the area, circularity and mean intensity of each segment of Pit was extracted. The mean intensity over time was normalized to its value at $t = 0$.

**Line scans.** A plane through the centre of each nucleolus was used for segmentation in Fiji using Otsu's method. The Feret's diameter was calculated for each nucleolus and intensity values were measured along the Feret's diameter. Intensities for each profile were normalized to its average value. The Feret's diameter was normalized by setting its range from zero to one.

**Actinomycin D treatment.** The intensity of ethynyl uridine was quantified from maximum intensity $z$ projections of imaged nuclei. After denoising with Gaussian Blur in Fiji, ethynyl uridine-rich regions were segmented using the Otsu method in vehicle- and ActD-treated nuclei. Segment intensities were measured using the Analyze Particles tool. The Fib volume was measured after autosegmentation in 3D using the Intensity Threshold Segmenter tool in Arivis. The Mod intensities were measured by drawing a circle of the same radius in the nucleolus and nucleoplasm. Intensities in the Mod channel were quantified using the 'Measure' tool in Fiji and the ratio of intensities (nucleolus:nucleoplasm) was plotted.

### Statistics and reproducibility

Data were plotted and statistical analyses were performed using GraphPad Prism (version 10). $P$ values were calculated using an unpaired two-tailed Student's $t$-test. For microscopy, 3–8 embryos/tissues were imaged per experimental group, with each $z$-stack acquisition containing at least 10–15 nuclei. For S2R+ cells, a minimum of ten cells were imaged for each experimental group. Individual nuclei were subjected to automated segmentation and only nuclei that failed segmentation due to weak fluorophore expression were excluded. Once a dataset was determined, no data points were excluded. This approach was sufficient to obtain adequate sample sizes for all experiments in this study. No statistical methods were used to pre-determine sample sizes. All replicates reported in this study are biological replicates. Data distribution was assumed to be normal but this was not formally tested. Samples and animals were assigned to experimental groups based on their defined genotypes or labelled perturbations, as described in the paper. Samples were randomly selected from each experimental group for image acquisition. Data collection and analysis were not performed blind to experimental conditions. All experiments were independently performed at least twice with consistent results, and complementary experimental techniques and multiple molecular markers were used to corroborate the key conclusions of this study. Representative microscopy images reflect organizational patterns consistently observed across multiple embryos and cells.

### Reporting summary

Further information on research design is available in the Nature Portfolio Reporting Summary linked to this article.

## Data availability

Accession codes (FlyBase FBgn IDs, GenBank accession numbers and UniProt IDs) for the genes and proteins analysed in this study are reported throughout the paper. Microscopy source data are available from *Dryad*: https://doi.org/10.5061/dryad.vx0k6dk59 (ref. 84). Source data are provided with this paper. All other data supporting the findings of this study are available from the corresponding author on reasonable request.

## Code availability

For the simulation component of this study, all simulations, analyses and visualizations were conducted using publicly available software packages and custom-developed codes. Langevin Dynamics simulations were performed using LAMMPS (version 23 June 2022) and visualizations were generated using Ovito (version 3.7.11). The complete set of input files, custom codes and analysis scripts necessary to reproduce the simulation results are openly available on *Zenodo*: https://doi.org/10.5281/zenodo.17253191 (ref. 85).

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

## Acknowledgements

We thank E. Wieschaus (Princeton University) for generously sharing fluorescently tagged nucleolar fly lines, B. Mellone (University of Connecticut) for the antibody to Mod, and R. Duronio (UNC, Chapel Hill) for the antibody to Mxc. We also thank the Dernburg, Welch and Brohawn laboratories at UC Berkeley for sharing equipment and resources. We thank K. Lin for assistance with *Drosophila* embryo collections. The Karpen laboratory acknowledges critical support from the National Institutes of Health (grant number R35GM139653), and S.S. and G.H.K. are grateful for the support of the Volkswagen Stiftung (grant number 98196).

## Author contributions

S.R., O.A.-A., G.B., S.S. and G.H.K. conceptualized the study. S.R. designed, performed and analysed most of the experiments. O.A.-A. and G.B. developed the coarse-grained model for PCH– nucleolar organization. S.U.C. identified the putative PxVxL motif in Pit, generated plasmids for S2R+ cell transfections and performed microscopy of nucleolar subcompartments in S2R+ cells. H.P. generated the Pit–APEX2 stable cell line and performed proximity ligation assays. L.D.B. purified recombinant HP1a and contributed to the purification of recombinant Pit. S.T. performed the AlphaFold structural analysis of Pit and HP1a. S.S. and G.H.K. supervised the study and provided funding. S.R., O.A.-A. and G.B. prepared the original and revised drafts of this paper, and S.S. and G.H.K. reviewed and edited the paper.

## Competing interests

The authors declare no competing interests.

## Additional information

**Extended data** is available for this paper at https://doi.org/10.1038/s41556-025-01806-7.

**Correspondence and requests for materials** should be addressed to Gary H. Karpen.

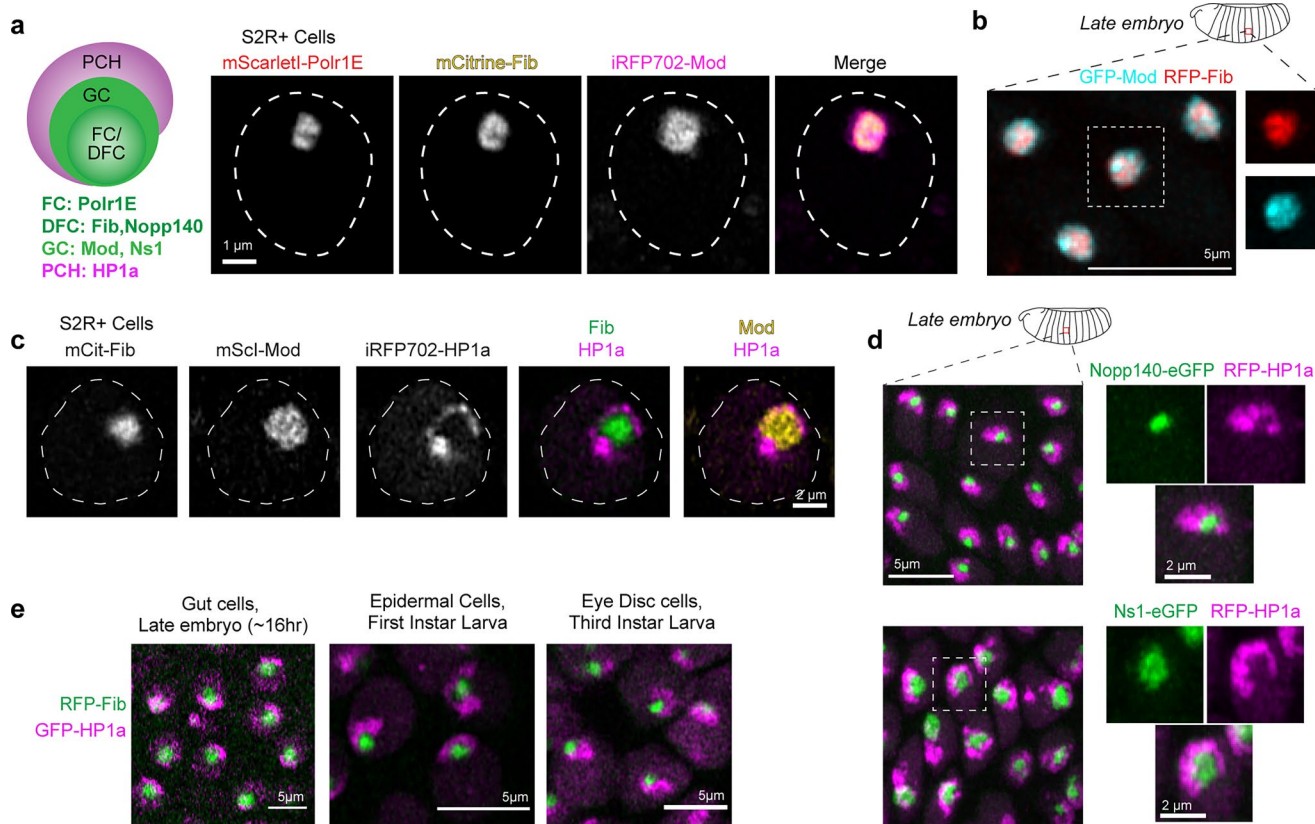

**Extended Data Fig. 1 | PCH–nucleolar organization in *Drosophila* embryos, cultured cell lines and larval tissues. a**, Left: Schematic showing the localization of nucleolar proteins analysed in this study to their respective nucleolar subcompartment in *Drosophila* cells. FC: RNA-Pol1 subunit (Polr1E), DFC: Fib and Nopp140, GC: Mod and Ns1. Right: Representative stills in live S2R+ cells transfected with mScarlet-I-Polr1E, mCitrine-Fib, iRFP702-Mod. **b**, Distribution of GFP–Mod (cyan) and RFP–Fib (red) in nucleoli of a live late-stage embryo.

**c**, Representative stills in live S2R+ cells transfected with mCitrine–Fib, mScarlet-I–Mod, iRFP702–HP1a. HP1a is organized around the GC. **d**, Distribution of RFP–HP1a (magenta) with Nopp140-eGFP (green, top) or Ns1-eGFP (green, bottom) in nucleoli of a live late embryo. **e**, Stills of nuclei expressing GFP–HP1a (magenta) and RFP–Fib (green) in gut cells from a late embryo (-16 h), first-instar larval epidermal cells, and third-instar larval eye disc.

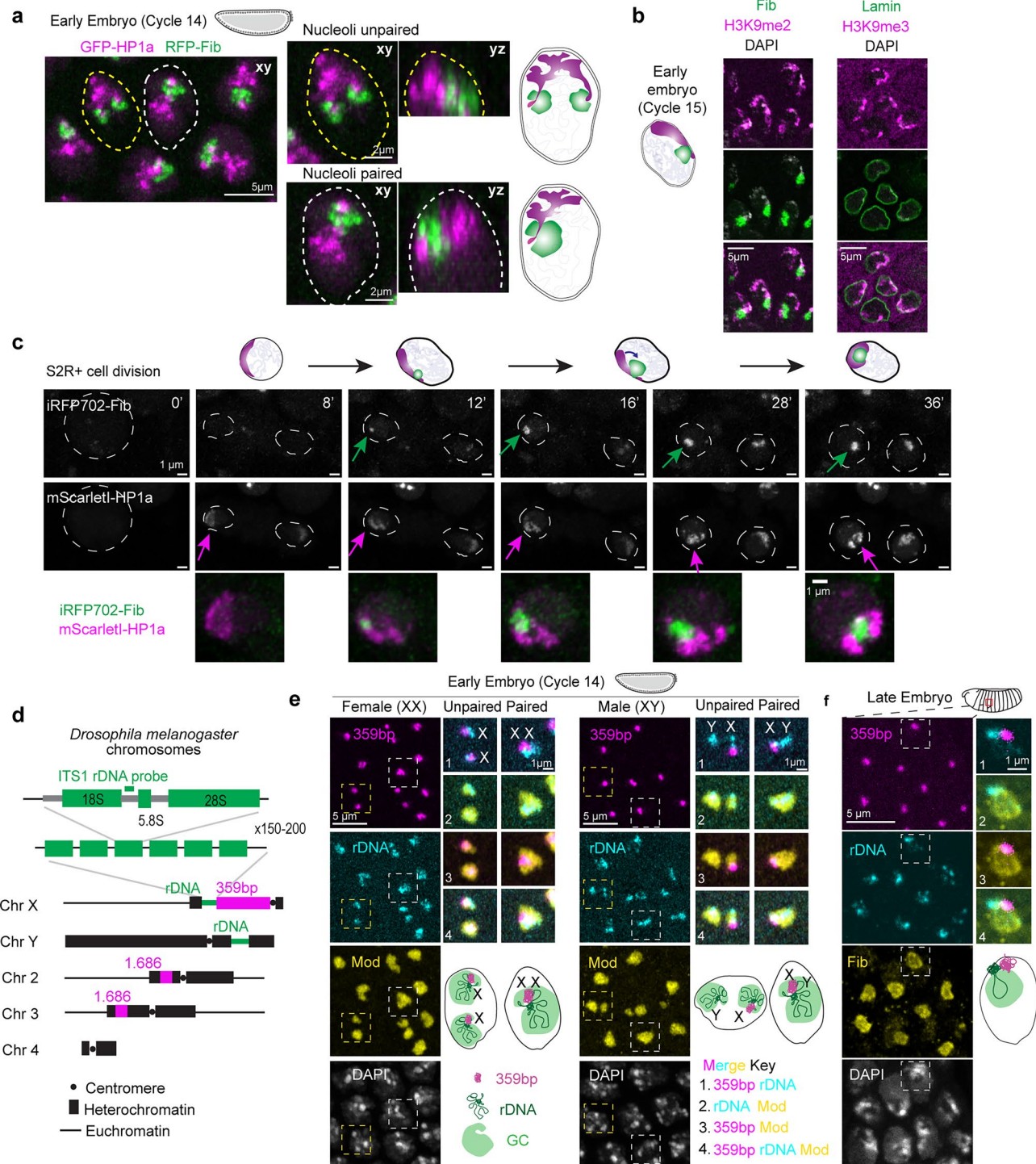

**Extended Data Fig. 2 | Dynamics of PCH reorganization relative to the nucleolus during *Drosophila* embryonic development. a**, Nuclei from Cycle 14 embryos expressing GFP–HP1a (magenta) and RFP–Fib (green), showing two unpaired nucleoli (yellow dashed line) and one paired nucleolus (white dashed line). Both nuclei have been enlarged to show the 'extended' conformation of HP1a relative to the nucleolus in the xy and yz views. **b**, Immunofluorescence staining of nuclei from early (Cycle 15) embryos. Left: H3K9me2 (magenta) and Fib (green). Right: H3K9me3 (magenta) and Lamin (green). DAPI (grey) is included in all panels. **c**, Time-lapse stills of two daughter S2 cells exiting mitosis, transfected with iRFP702–Fib (green) and mScarlet-I–HP1a (magenta). Numbers on the top right corner indicate time in minutes from the end of mitosis. Green arrows track a nucleolus over time, while magenta arrows follow the reassembly

of an HP1a condensate. The nucleus marked with the arrows is enlarged below. **d**, Schematic representation of pericentromeric satellite repeats (359 bp and 1.686) and rDNA repeats in *Drosophila melanogaster* chromosomes. The schematic of ribosomal DNA (rDNA) arrays indicates the position of the ITS-1 rDNA probe used for FISH. **e**, Combined immuno–FISH shows the organization of 359 bp satellite DNA (magenta), ITS-1 rDNA (cyan), Mod (yellow) and DAPI (grey) in female (XX) and male (XY) early (Cycle 14) embryos. A white dashed box marks a nucleus with a single paired nucleolus, while a yellow dashed box marks a nucleus with two unpaired nucleoli. Enlarged views of these nuclei are shown on the right with different fluorescent channel combinations. **f**, Combined immuno-FISH stained for 359bp (magenta), ITS-1 rDNA (cyan), Fib (yellow), and DAPI (grey) in nuclei from the epidermis (external surface) of late embryos.

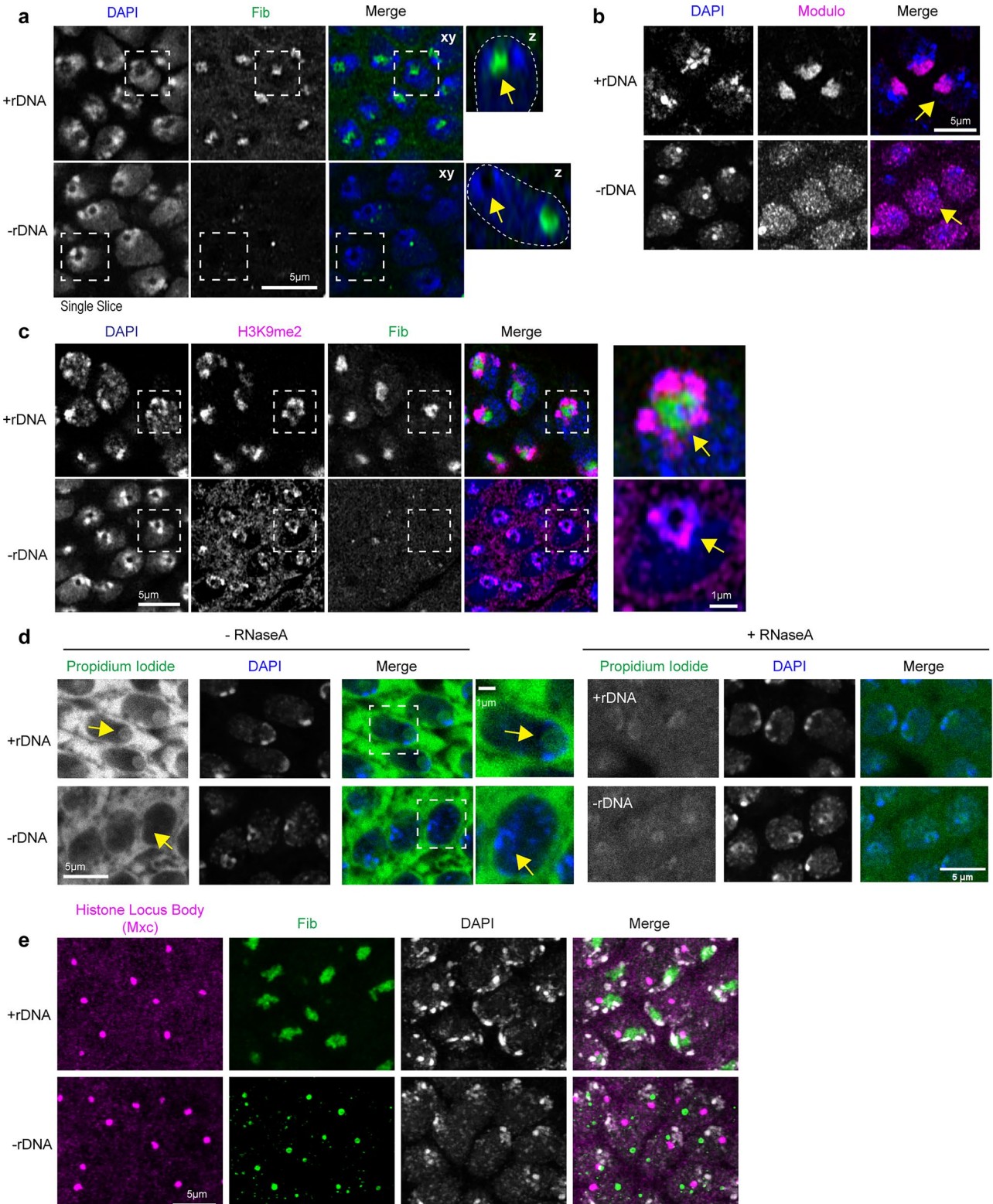

**Extended Data Fig. 3 | The PCH void in −rDNA embryos does not stain for DAPI, Fib, Mod, H3K9me2 or Propidium Iodide (RNA). a**, Representative images of fixed nuclei from late-stage +rDNA and −rDNA embryos stained for Fib (green) and DAPI (blue). Nuclei marked with white dashed box are enlarged and shown in the z plane. The yellow arrow points to the lack of Fib in the 'PCH void' in −rDNA nuclei. **b**, Representative images of fixed nuclei from +rDNA and -rDNA embryos stained for Mod (magenta) and DAPI (blue). The yellow arrow points to Mod localization in a +rDNA nucleus and dispersion in −rDNA. **c**, H3K9me2 (magenta), Fib (green) immunofluorescence and DAPI (blue) staining in +rDNA and −rDNA

nuclei in late-stage *Drosophila* embryos. The yellow arrow points to the lack of H3K9me2 in the 'PCH void' in −rDNA nuclei. **d**, Representative images of fixed nuclei from late-stage +rDNA and −rDNA embryos stained with propidium iodide (green) and DAPI (blue) without (left) and with (right) RNase A. The yellow arrow points to the RNA staining in the nucleolus in +rDNA and lack of propidium iodide staining in the PCH void of −rDNA nuclei. **e**, Immunofluorescence images staining for Mxc, a marker for the Histone Locus body (magenta), Fib (green), and DAPI (grey) in +rDNA and −rDNA nuclei in early embryos (~cycle 15).

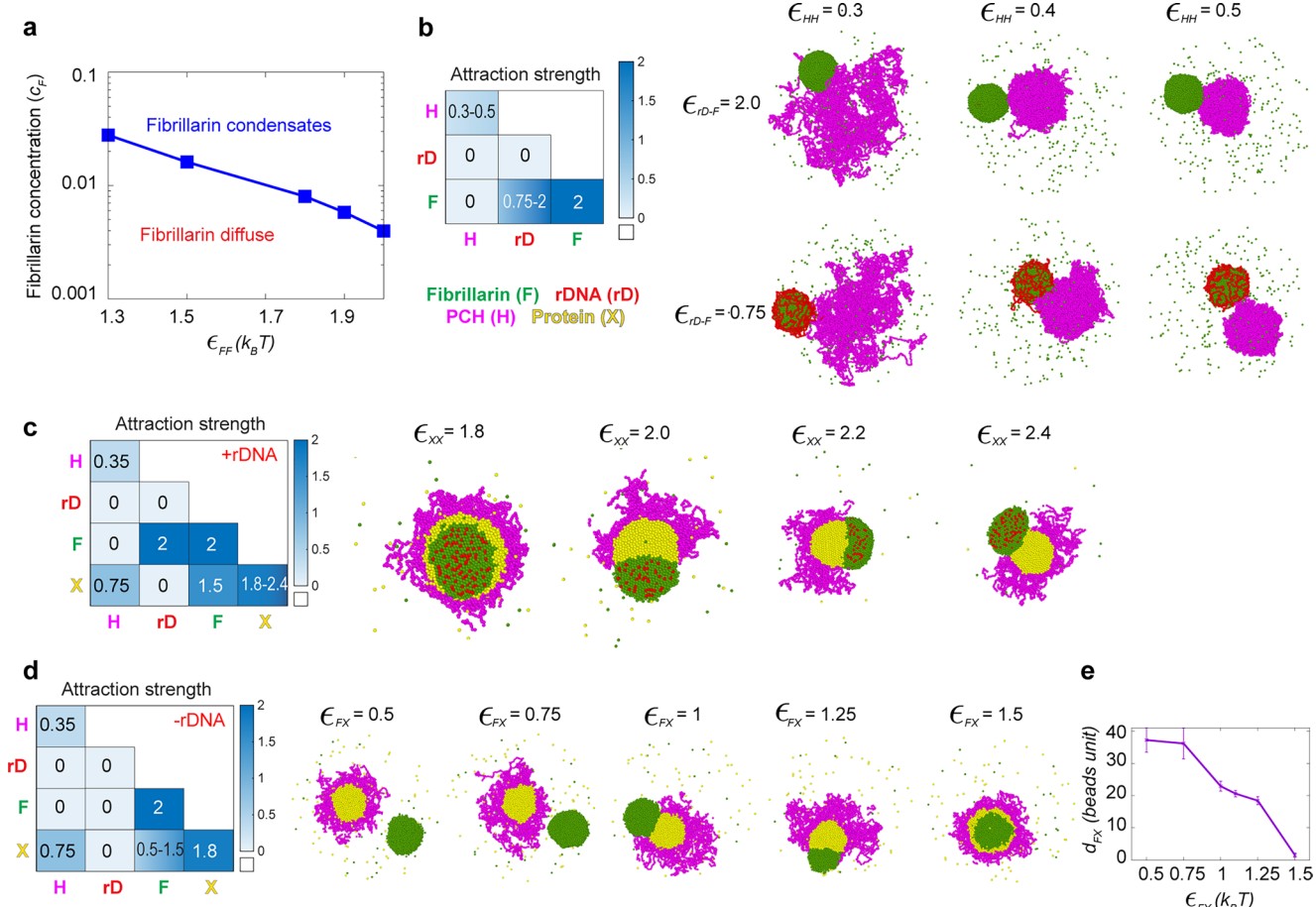

**Extended Data Fig. 4 | Coarse-grained model for the assembly of the nucleolus and PCH. a**, Simulation results yield a phase diagram which indicates that a minimum attraction strength of $\epsilon_{FF} = 1.3 - 2.0\,k_B\text{T}$ is required to condense Fib particles within the concentration range of $c_F = 0.0013 - 0.013$. **b**, Simulation endpoint snapshots depict the outcomes of varying the attraction strengths between beads of the PCH (H) polymer chain ($\epsilon_{HH}$). Within the range of $\epsilon_{HH} = 0.3 - 0.5\,k_B\text{T}$, the PCH chain underwent collapse, resulting in a condensed conformation that is phase-separated from the aqueous component of the system. Based on these results, $\epsilon_{HH} = 0.35\,k_B\text{T}$ was set for subsequent simulations. The attraction strength between rDNA and Fib ($\epsilon_{rD-F}$) was varied to show that weaker attraction (and $\epsilon_{rD-F} = 0.75\,k_B\text{T}$) resulted in rDNA wrapping around the surface of a Fib condensate, while a stronger attraction ($\epsilon_{rD-F} = 2\,k_B\text{T}$) led to the condensation of rDNA within the Fib condensate. The latter observation aligns with experimental observations; thus $\epsilon_{rD-F} = 2\,k_B\text{T}$ was chosen in the

subsequent simulations. **c**, Simulation endpoint snapshots depict the outcomes of varying X-X attraction strengths ($\epsilon_{XX}$). If $\epsilon_{XX} < \epsilon_{FF}$, PCH surrounds the Protein X-rich phase. For $\epsilon_{XX} \geq \epsilon_{FF}$, we observe PCH surrounding the dual-affinity protein-rich phase, but just a partial wetting between the dual-affinity protein-rich phase and the Fib-rich phase. **d**, Simulation endpoint snapshots depict the outcomes of varying attraction strengths between Fib and protein X ($\epsilon_{FX}$) in the −rDNA condition. If $\epsilon_{FX} \leq \epsilon_{XH}$, the Fib-rich phase and the dual-affinity protein-rich phase do not associate with each other. Increasing $\epsilon_{FX} \geq \epsilon_{XH}$ causes the protein X phase to partially or completely wet Fib. **e**, The average distance between the centre of masses of Fib and protein X condensates ($d_{FX}$) is measured for different attraction strengths between Fib and protein X ($\epsilon_{FX}$) in the −rDNA condition. Each simulation produced 2000 timeframes. Statistical analyses were performed on the last $n = 1,000$ frames per simulation, corresponding to the equilibrium portion of the trajectory. Error Bars represent s.d.

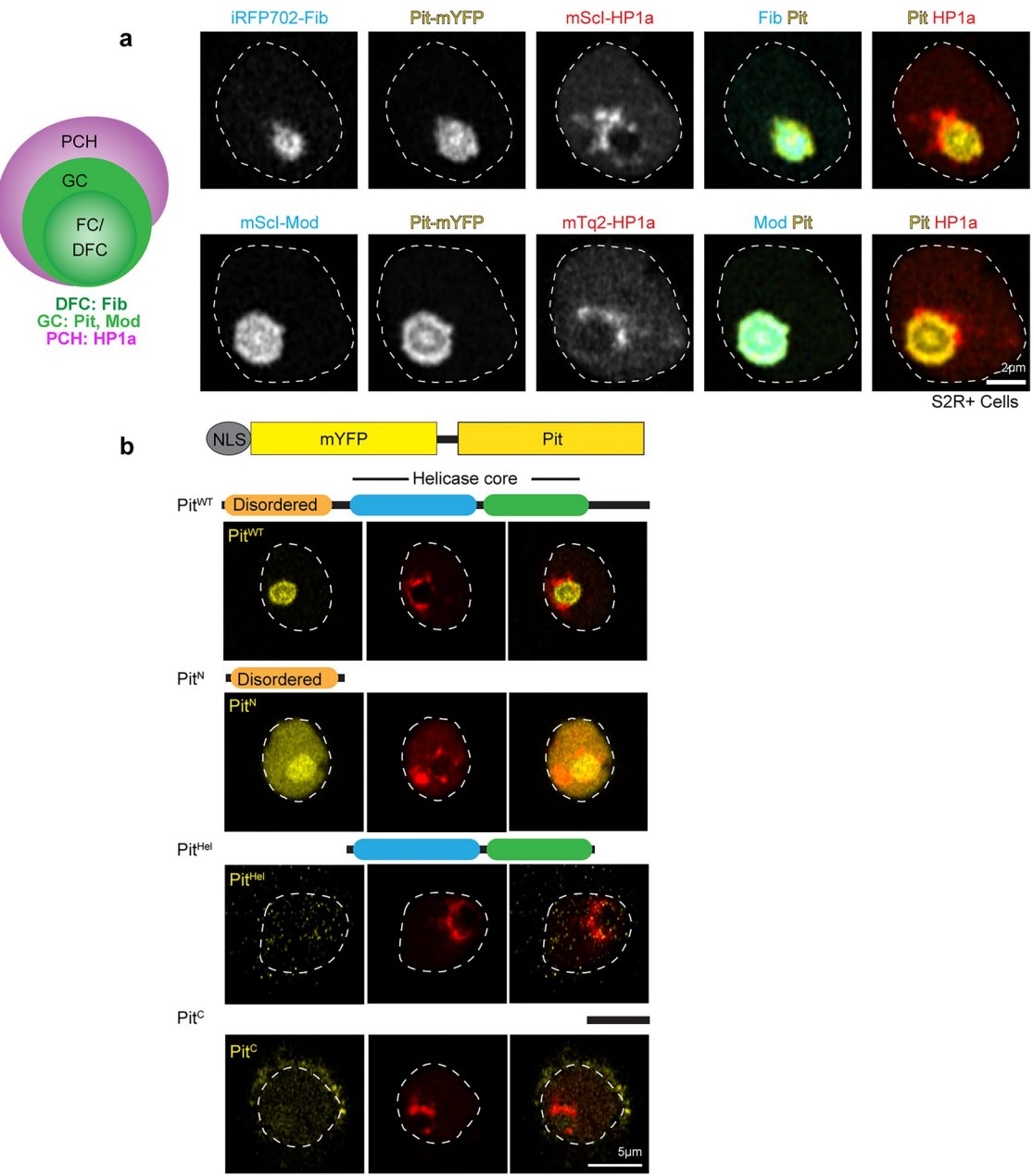

**Extended Data Fig. 5 | Pit is a GC protein. a**, Representative stills in live S2R+ cells transfected with (top) Pit–mYFP, mScarlet-I–Mod, iRFP702–HP1a and (bottom) Pit–mYFP, iRFP702–Fib, and mTurquoise2–HP1a. **b**, Representative stills from live S2R+ cells expressing mYFP-tagged versions of Full-Length Pit (Pit[WT]), the disordered N-terminal domain (Pit[N]), the central helicase core (Pit[Hel]), or the disordered C-terminal domain (Pit[C]), each containing the c-myc nuclear localization signal (NLS). Each of these constructs were co-transfected with Scarlet-I–HP1a.

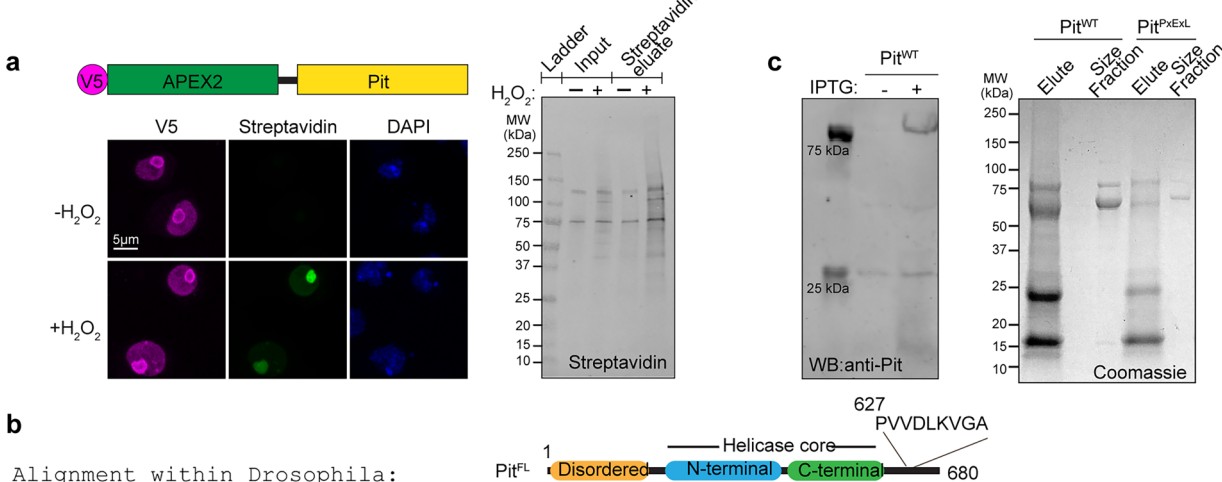

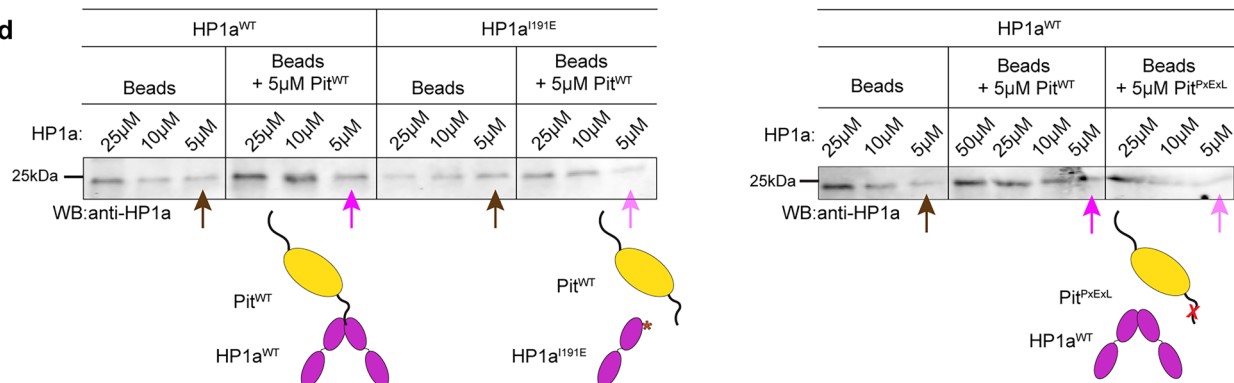

**Extended Data Fig. 6 | See next page for caption.**

**Extended Data Fig. 6 | Pit interacts with HP1a via a conserved PxVxL HP1a-interacting motif. a**, Top: Schematic of the V5-APEX2–Pit construct used to generate a stable S2R+ cell line for biotin-based proximity ligation. Bottom left: Validation of APEX2-tagged Pit expression using anti-V5 immunostaining. After treating cells with $H_2O_2$, biotinylation was confirmed using fluorescent streptavidin staining. Bottom right: Western Blots of inputs and streptavidin-captured (biotinylated) proteins following APEX2–Pit proximity labelling, probed with anti-Streptavidin antibody. **b**, Sequence alignment showing conservation of a PxVxL motif in Pit within *Drosophila* species (top) and across non-*Drosophila* eukaryotic model organisms (bottom). **c**, Left: Western Blot probed with anti-Pit to confirm expression of His-tagged recombinant Pit in *E. coli* after IPTG induction. Right: SDS–PAGE analysis of proteins eluted from Ni-NTA resin after purification of recombinant Pit in Pit$^{WT}$ and Pit$^{PxExL}$ forms. Eluted proteins (left lane) were further purified using size-exclusion chromatography (right lane). **d**, His-tagged binding assay was performed using 5 μM His-tagged Pit protein (WT or PxExL mutant form) incubated with decreasing concentrations of HP1a (WT or I191E dimer-mutant form) at 50, 25, 10, and 5 μM. As a negative control, HP1a alone (no Pit protein) was incubated with beads, revealing non-specific binding of HP1a to the Ni-NTA magnetic beads (brown arrows). Thus, a dilution series of Beads + HP1a alone was included to assess background signal. Dilution series reveals an enrichment for HP1a signal (solid magenta arrow) over background (brown arrow) when Pit$^{WT}$ and HP1a$^{WT}$ are incubated at equimolar concentrations (5 μM). This enrichment is reduced when Pit$^{WT}$ and HP1a$^{II91E}$ or Pit$^{PxExL}$ and HP1a$^{WT}$ were incubated (faded magenta arrow).

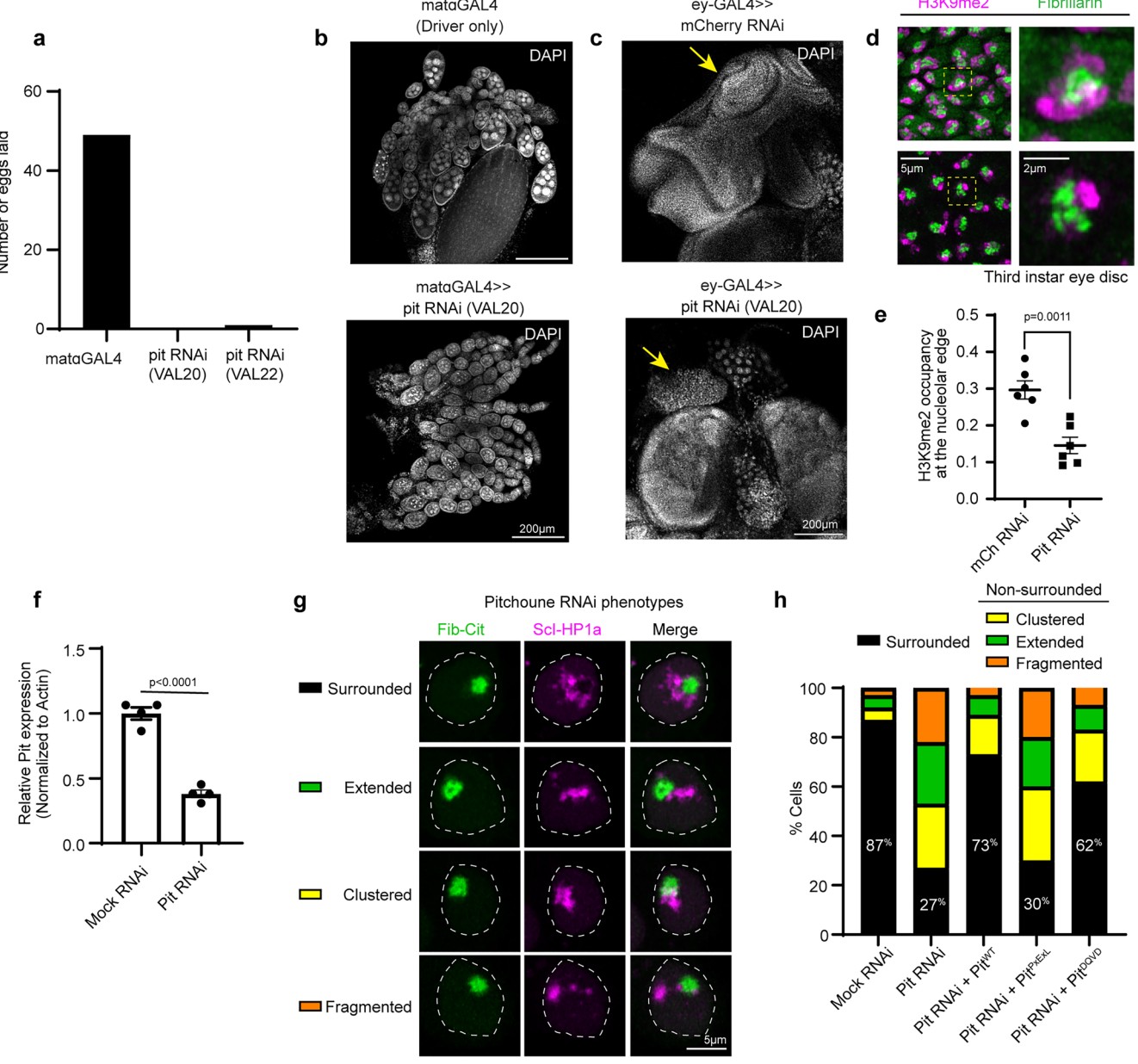

**Extended Data Fig. 7 | Developmental defects and HP1a disorganization phenotypes due to Pit knockdown in *Drosophila* tissues and cultured cells. a**, Number of eggs laid after a 3-h collection in matαGAL4 (driver only) and matαGAL4 driving UAS-Pit RNAi using two independent RNAi lines (VAL20 and VAL22). **b**, Representative images of dissected ovaries in control (matαGAL4, driver only) and after Pit RNAi (VAL20) knockdown stained with DAPI. **c**, Representative images of dissected eye antennal discs (yellow arrow) in control (ey-GAL4>mCherry RNAi) and after Pit knockdown (ey-GAL4>pit RNAi, VAL20) stained with DAPI. **d**, Immunofluorescence of H3K9me2 (magenta), Fib (green) and DAPI in nuclei from dissected eye discs in third-instar larvae from UAS−pit RNAi and UAS−mCherry RNAi (control) driven by the eyeless-GAL4 driver. **e**, H3K9me2 occupancy at the nucleolar edge was quantified in eye-antennal discs from $n = 6$ control animals (intact discs) and $n = 6$ Pit RNAi

animals (developmentally deformed discs). On average, 90 nuclei (with one nucleolus per nucleus) were analysed per animal. Each data point represents the mean H3K9me2 occupancy around the nucleolus per animal. Error bars: s.e.m. **f**, Quantitation of Pit transcripts using qPCR to confirm the knockdown of Pit, normalized to actin. Bar graphs depict mean ± s.e.m. $n = 4$ biological replicates. **g**, Representative nuclei transfected with Fib-Citrine and Scarlet-I-HP1a, showing four categories of HP1a distribution phenotypes observed after Pit knockdown. **h**, Quantification of the % nuclei with Surrounded, Extended, Clustered, and Fragmented HP1a phenotype after Pit RNAi and its rescue with Pit$^{WT}$, Pit$^{PxExL}$, and Pit$^{DQVD}$. Phenotypes were counted in $n = 39$ (Mock RNAi), $n = 73$ (Pit RNAi), $n = 51$ (Pit RNAi + Pit$^{WT}$), $n = 60$ (Pit RNAi + Pit$^{PxExL}$), $n = 52$ nuclei (Pit RNAi + Pit$^{DQVD}$) pooled from two independent experiments.

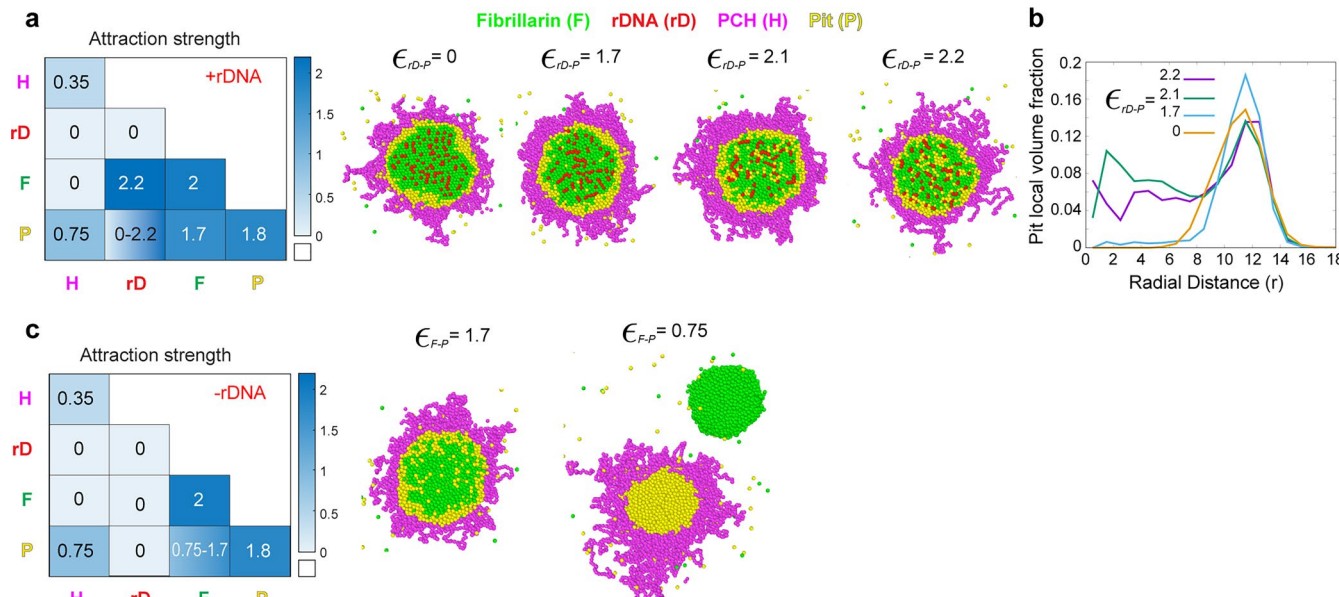

**Extended Data Fig. 8 | Refining the coarse-grained model for PCH–nucleolar organization to capture domain-specific interactions of Pit with PCH and nucleolar components. a**, Simulation endpoint snapshots depict the outcomes of varying rD–P attraction strengths ($\epsilon_{rD-P}$). If $\epsilon_{rD-P} > \epsilon_{F-P}$, Pit localizes within the Fib-rich protein phase aligning with experimental observations of Pit localization in the nucleolar core. **b**, Radial distribution of Pit calculated from the centre of mass of the Fib beads under varying rD–P attraction strengths ($\epsilon_{rD-P}$). **c**, Simulation endpoint snapshots depict the outcomes of varying F–P attraction strengths ($\epsilon_{F-P}$) using parameters from the refined Pit interaction model, except

all rDNA interactions are set to zero to simulate the -rDNA condition. At $\epsilon_{F-P} = 1.7$, Pit completely wets the Fib phase. Only when $\epsilon_{F-P} = 0.75$ do the Fib and the Pit-rich phases become separate. This indicates that removing rDNA–Fib interactions is not sufficient to detach Fib from PCH. A reduction in Fib-Pit interactions is also necessary. In cells, this implies that nucleolar proteins recruited in +rDNA nuclei likely strengthen Fib-Pit interactions, and their absence in -rDNA conditions may weaken Fib-Pit affinity to generate the observed -rDNA nuclear phenotypes.

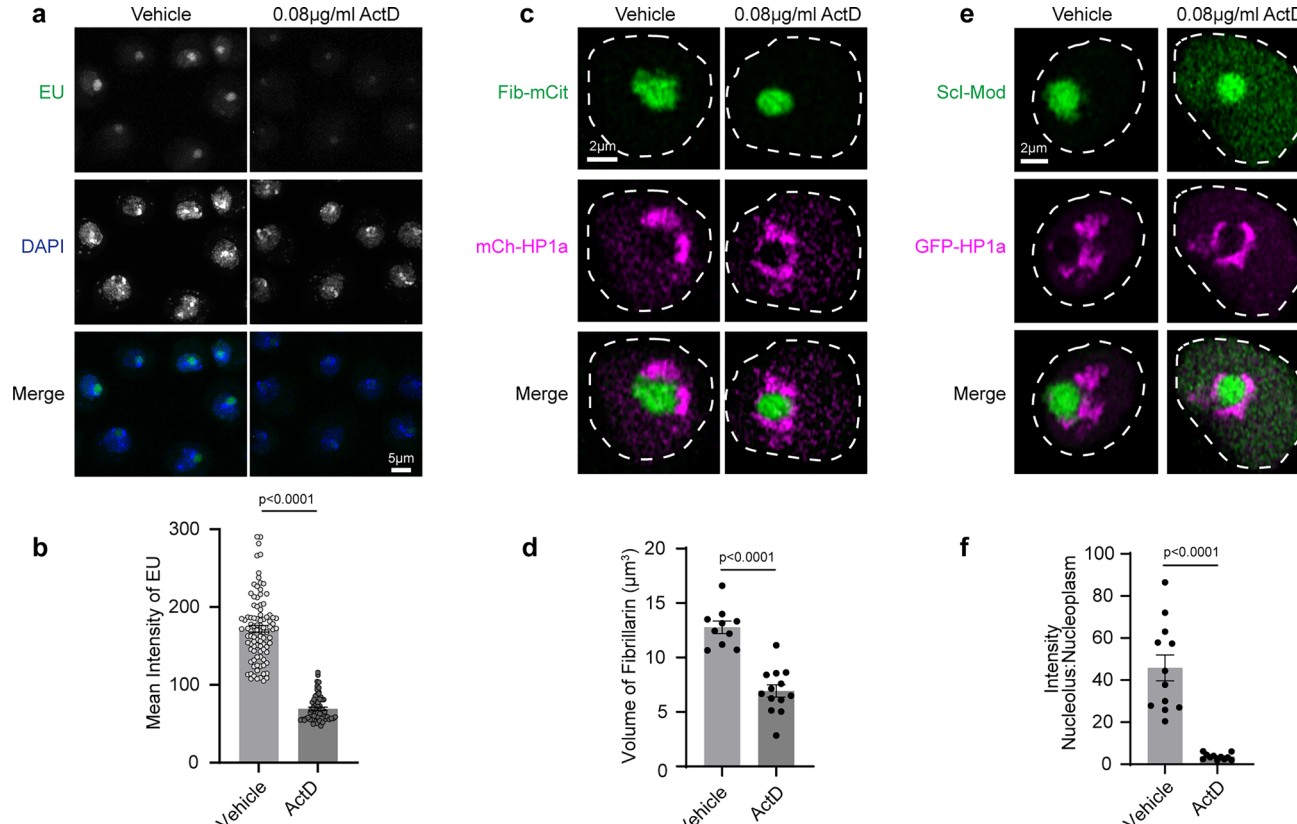

**Extended Data Fig. 9 | Pol-I inhibition using a low dose of ActD in S2R+ cells changes nucleolar and PCH organization. a**, Representative images of S2R+ cells stained with 5-Ethynyl Uridine (EU) (green) and DAPI (blue) to visualize nascent rRNA transcription following 0.08 µg ml⁻¹ ActD treatment after 10 min. **b**, Quantification of EU fluorescence intensity in Vehicle ($n = 96$) and ActD-treated ($n = 64$) nuclei, showing a significant decrease in nascent rRNA synthesis. Bar graphs depict mean ± s.e.m. **c**, Representative images of live S2R+ cells transfected with Fib–mCitrine (green) and mCh–HP1a (magenta), treated with vehicle or 0.08–µg–ml⁻¹ ActD, and imaged between 10–60 min after drug treatment. **d**, Volume of Fib measured in S2R+ cells treated with vehicle

($n = 10$ cells) or ActD ($n = 13$ cells). Bar graphs depict mean ± s.e.m. **e**, Representative images of live S2R+ cells transfected with Scarlet-I–Mod (green) and GFP–HP1a (magenta) and treated with vehicle or ActD. Display brightness and contrast in the Scl-Mod channel were linearly adjusted in ActD-treated cells to aid visualization of Mod's distribution within and outside the nucleolus. **f**, The ratio of fluorescence intensity of Modulo in the nucleolus over nucleoplasm was measured in representative nuclei in vehicle ($n = 12$) or ActD-treated ($n = 11$) cells. Bar graphs depict mean ± s.e.m. All ActD treatment quantification panels represent data collected from one independent seeding each, and results were qualitatively verified in another independent seeding.

# Reporting Summary

## Statistics

For all statistical analyses, confirm that the following items are present in the figure legend, table legend, main text, or Methods section.

| n/a | Confirmed | |
|---|---|---|
| ☐ | ☒ | The exact sample size (*n*) for each experimental group/condition, given as a discrete number and unit of measurement |
| ☐ | ☒ | A statement on whether measurements were taken from distinct samples or whether the same sample was measured repeatedly |
| ☐ | ☒ | The statistical test(s) used AND whether they are one- or two-sided<br>*Only common tests should be described solely by name; describe more complex techniques in the Methods section.* |
| ☒ | ☐ | A description of all covariates tested |
| ☒ | ☐ | A description of any assumptions or corrections, such as tests of normality and adjustment for multiple comparisons |
| ☐ | ☒ | A full description of the statistical parameters including central tendency (e.g. means) or other basic estimates (e.g. regression coefficient) AND variation (e.g. standard deviation) or associated estimates of uncertainty (e.g. confidence intervals) |
| ☐ | ☒ | For null hypothesis testing, the test statistic (e.g. *F*, *t*, *r*) with confidence intervals, effect sizes, degrees of freedom and *P* value noted<br>*Give P values as exact values whenever suitable.* |
| ☒ | ☐ | For Bayesian analysis, information on the choice of priors and Markov chain Monte Carlo settings |
| ☒ | ☐ | For hierarchical and complex designs, identification of the appropriate level for tests and full reporting of outcomes |
| ☒ | ☐ | Estimates of effect sizes (e.g. Cohen's *d*, Pearson's *r*), indicating how they were calculated |

*Our web collection on statistics for biologists contains articles on many of the points above.*

## Software and code

Policy information about availability of computer code

| Data collection | ZEN Zeiss |
|---|---|
| Data analysis | ImageJ (Fiji), Arivis Vision 4D, GraphPad Prism, LAMMPS, Ovito, Github link: https://github.com/gauravbajpaimaths/Coarse-grained_model_of_nucleolar_heterochromatin_condensates |

For manuscripts utilizing custom algorithms or software that are central to the research but not yet described in published literature, software must be made available to editors and reviewers. We strongly encourage code deposition in a community repository (e.g. GitHub). See the Nature Portfolio guidelines for submitting code & software for further information.

## Data

Policy information about availability of data

All manuscripts must include a data availability statement. This statement should provide the following information, where applicable:
- Accession codes, unique identifiers, or web links for publicly available datasets
- A description of any restrictions on data availability
- For clinical datasets or third party data, please ensure that the statement adheres to our policy

All data supporting the findings of this study are available within the paper and its Supporting Information.

# Research involving human participants, their data, or biological material

Policy information about studies with human participants or human data. See also policy information about sex, gender (identity/presentation), and sexual orientation and race, ethnicity and racism.

| | |
|---|---|
| Reporting on sex and gender | n/a |
| Reporting on race, ethnicity, or other socially relevant groupings | n/a |
| Population characteristics | n/a |
| Recruitment | n/a |
| Ethics oversight | n/a |

Note that full information on the approval of the study protocol must also be provided in the manuscript.

# Field-specific reporting

Please select the one below that is the best fit for your research. If you are not sure, read the appropriate sections before making your selection.

☒ Life sciences ☐ Behavioural & social sciences ☐ Ecological, evolutionary & environmental sciences

For a reference copy of the document with all sections, see nature.com/documents/nr-reporting-summary-flat.pdf

# Life sciences study design

All studies must disclose on these points even when the disclosure is negative.

| | |
|---|---|
| Sample size | >30 nuclei were analyzed from at least 3 animals for each experimental group. In cell culture experiments, At least 10 cells were analyzed for each condition. While formal sample size calculations were not performed, 'n' was determined to be sufficient based on the reproducibility and consistency of the observed phenotypes among biological replicates. Precise n values for each experiment are reported in the manuscript. |
| Data exclusions | Microscopy data were only excluded from analysis when low fluorescence intensities caused inaccurate segmentation of the region of interest. |
| Replication | Experimental findings reported in the study were replicated at least once. |
| Randomization | Samples were allocated into experimental groups based on their genotype. Within each experimental group, samples were randomly selected for image acquisition and downstream analyses. |
| Blinding | Investigators were not blinded to group allocation during data collection or analysis. Experiments on control and test groups were performed concurrently, but they were processed in separate labeled tubes. To mitigate potential bias, all downstream measurements comparing different experimental groups were conducted using identical acquisition parameters. |

# Reporting for specific materials, systems and methods

We require information from authors about some types of materials, experimental systems and methods used in many studies. Here, indicate whether each material, system or method listed is relevant to your study. If you are not sure if a list item applies to your research, read the appropriate section before selecting a response.

## Materials & experimental systems

| n/a | Involved in the study |
|---|---|
| ☐ | ☒ Antibodies |
| ☐ | ☒ Eukaryotic cell lines |
| ☒ | ☐ Palaeontology and archaeology |
| ☐ | ☒ Animals and other organisms |
| ☒ | ☐ Clinical data |
| ☒ | ☐ Dual use research of concern |
| ☒ | ☐ Plants |

## Methods

| n/a | Involved in the study |
|---|---|
| ☒ | ☐ ChIP-seq |
| ☒ | ☐ Flow cytometry |
| ☒ | ☐ MRI-based neuroimaging |

# Antibodies

| | |
|---|---|
| Antibodies used | Antibodies used in this study: Rabbit anti-Fibrillarin (Abcam ab5821), Mouse anti-H3K9me2 (Abcam ab1220), Rabbit anti-H3K9me3 (Abcam ab8898), Mouse anti-Modulo (Gift from Mellone Lab), Mouse anti-Lamin, Dm0 (DSHB ADL67.10), Guinea Pig anti-Mxc (Gift from Duronio Lab), Rabbit anti-Pitchoune (Pit) was generated in this study by Pacific Immunology (CA), Goat-anti-Mouse Alexa Fluor 488 (Invitrogen A-11001), Goat-anti-Mouse Alexa Fluor 568 (Invitrogen A-11004), Goat-anti-Rabbit Alexa Fluor 488 (Invitrogen A-11034), Donkey-anti-Rabbit Alexa Fluor 568 (Invitrogen A-10042), Goat anti-Guinea Pig Alexa Fluor 647 (Invitrogen-A-21450), IgG (Invitrogen, 02-6102), Mouse anti-HP1a (DSHB, CA19) |
| Validation | Commercially purchased antibodies were validated by the suppliers. The antibody against Pit was generated for this study and validated by Western blotting in nuclear lysates, which detected a band at the predicted size of Pit. Additionally, when recombinant Pit was expressed in E. coli, the antibody detected it at the expected size. |

# Eukaryotic cell lines

Policy information about cell lines and Sex and Gender in Research

| | |
|---|---|
| Cell line source(s) | S2R+ Drosophila cell lines were used in this study and procured from DGRC (Stock 150 ; https://dgrc.bio.indiana.edu// stock/150 ; RRID:CVCL_Z831). S2R+ are male Drosophila embryonic cells. |
| Authentication | The cell lines were not authenticated. |
| Mycoplasma contamination | Cells were not tested for mycoplasma contamination. |
| Commonly misidentified lines (See ICLAC register) | No commonly misidentified lines reported in the ICLAC register were used in this study. |

# Animals and other research organisms

Policy information about studies involving animals; ARRIVE guidelines recommended for reporting animal research, and Sex and Gender in Research

| | |
|---|---|
| Laboratory animals | RFP-HP1a, GFP-HP1a (Karpen Laboratory); eGFP-Fibrillarin, RFP-Fibrillarin, eGFP-Mod, Pit-eGFP, eGFP-Nopp140, eGFP-Ns1 (shared by the Wieschaus laboratory), FM6/C(1)DX, y[*] f[1] (BDSC # 784), C(1)RM/C(1;Y)6,y[1]w[1]f[1]/0 (BDSC # 9460), Mat-alpha GAL4 (BDSC # 7063), Eyeless GAL4 (BDSC # 5534), Pit RNAi VAL20 (BDSC # 80368), Pit RNAi VAL22 (BDSC # 43984) |
| Wild animals | n/a |
| Reporting on sex | n/a |
| Field-collected samples | n/a |
| Ethics oversight | n/a |

Note that full information on the approval of the study protocol must also be provided in the manuscript.

# Plants

| | |
|---|---|
| Seed stocks | n/a |
| Novel plant genotypes | n/a |
| Authentication | n/a |

