## [Peer Review File · Nature Cell Biology]

Hierarchical interactions between nucleolar and heterochromatin condensates are mediated by a dual-affinity protein

Corresponding Author: Professor Gary Karpen

Version 0:

Decision Letter:

*Please delete the link to your author homepage if you wish to forward this email to co-authors.

Dear Dr Karpen,

I apologize once again for the delay; due in part to one reviewer who was overdue in their report.

Your manuscript, "Affinity hierarchies and amphiphilic proteins underlie the co-assembly of nucleolar and heterochromatin condensates", has now been seen by 3 referees, who are experts in biomolecular condensation and nucleoli (referee 1); *Drosophila* genome organization (referee 2); and nucleoli and rRNA (referee 3). As you will see from their comments (attached below) they find this work of potential interest, but have raised substantial concerns, which in our view would need to be addressed with considerable revisions before we can consider publication in Nature Cell Biology.

Nature Cell Biology editors discuss the referee reports in detail within the editorial team, including the chief editor, to identify key referee points that should be addressed with priority, and requests that are overruled as being beyond the scope of the current study. To guide the scope of the revisions, I have listed these points below. We are committed to providing a fair and constructive peer-review process, so please feel free to contact me if you would like to discuss any of the referee comments further.

I should stress that the referees' concerns point to potential pleiotropic effects of the genetic perturbations and methodologies including those of -rDNA, as well as unclear mechanistic links including those that include effects on subnucleolar organization, which would need to be addressed with experiments and data. Reconsideration of the study for this journal and re-engagement of referees would depend on strength of these new data and revisions.

In particular, it would be essential to:

A) Assess potential direct interactions of PCH with other components which would include new experiments in vitro (Reviewer #1) and in cell culture (Reviewers #2 and #3)

B) Provide further experimental evidence to assess the effects, off-target or otherwise, of -rDNA perturbations (Reviewers #2 and #3)

C) Justify the parameters and predictions from the simulations (Reviewers #1 and #3)

D) Experimentally assess localization and organization of other nucleolar components (Reviewer #1)

D) All other referee concerns pertaining to strengthening existing data, providing controls, methodological details, clarifications and textual changes, should also be addressed.

E) Finally please pay close attention to our guidelines on statistical and methodological reporting (listed below) as failure to do so may delay the reconsideration of the revised manuscript. In particular please provide:

We would be happy to consider a revised manuscript that would satisfactorily address these points, unless a similar paper is published elsewhere, or is accepted for publication in Nature Cell Biology in the meantime.

- ensure that it conforms to our format instructions and publication policies (see below and www.nature.com/nature/authors/).

- provide a point-by-point rebuttal to the full referee reports verbatim, as provided at the end of this letter.

- provide the completed Editorial Policy Checklist (found here <https://www.nature.com/authors/policies/Policy.pdf>), and Reporting Summary (found here <https://www.nature.com/authors/policies/ReportingSummary.pdf>). This is essential for reconsideration of the manuscript and these documents will be available to editors and referees in the event of peer review. For more information see <http://www.nature.com/authors/policies/availability.html> or contact me.

Nature Cell Biology is committed to improving transparency in authorship. As part of our efforts in this direction, we are now requesting that all authors identified as 'corresponding author' on published papers create and link their Open Researcher and Contributor Identifier (ORCID) with their account on the Manuscript Tracking System (MTS), prior to acceptance. ORCID helps the scientific community achieve unambiguous attribution of all scholarly contributions. You can create and link your ORCID from the home page of the MTS by clicking on 'Modify my Springer Nature account'. For more information please visit <http://www.springernature.com/orcid>.

Link Redacted

We would like to receive a revised submission within six months. We would be happy to consider a revision even after this timeframe, however if the resubmission deadline is missed and the paper is eventually published, the submission date will be the date when the revised manuscript was received.

We hope that you will find our referees' comments, and editorial guidance helpful. Please do not hesitate to contact me if there is anything you would like to discuss.

Best wishes,

Daryl

Daryl Jason Verzosa David, PhD

Senior Editor, Nature Cell Biology
Advisory Editor, npj Biological Physics and Mechanics
Nature Portfolio

Heidelberger Platz 3, 14197 Berlin, Germany
Email: daryl.david@nature.com
ORCID: <https://orcid.org/0000-0002-9253-4805>

Reviewers' Comments:

Reviewer #1 (Remarks to the Author):

In this work, the authors present their findings of the spatial organization of the PCH and the nucleolus with respect to one another. These results come from studies in *Drosophila* at different stages of embryogenesis. The key finding is that the PCH organizes around the nucleolus, and that this organization is disrupted when the nucleolus is removed. The authors argue that Pitchoune is an amphiphilic protein that facilitates the organization of the PCH around the nucleolus. Overall, this is an interesting and timely study that will animate a lot of healthy and vibrant discussion. It certainly merits publication in Nature Cell Biology. However, there are several issues that need to be addressed on the technical, conceptual, and scholarship side. These are listed below.

1) Modulo is a paralog of nucleolin. However, nucleolin, at least in *Xenopus* oocytes and in mammalian cells, is a marker of the DFC and not the GC. NPM1 is the most widely used marker of the GC, and in *Drosophila* this would be nucleoplasmin like protein (NLP). Please see <https://doi.org/10.1093/nar/gky988>. Experts in the nucleolus field will likely be interested in whether nucleolin behaves very differently in flies or if a different marker such as NLP should be used. Either the biology should provide a sound case for not using NLP or it should be used to make a definitive case for the organization that the authors are putting forth. Most of the images only show HP1a and Fibrillarin markers, and so it is unclear if the PCH is layered atop the GC or the DFC.

2) The use of 488 NHS ester treatment is fraught with concerns. This is being used to assert that the apparent void in the toroidal structures is filled with proteins. Typically, efficient and reliable NHS ester labeling of proteins requires buffers of pH 8.3 or thereabouts. How can one rule out that the labeling is not of free amines in the apparent void? The naive expectation is of a lot of off-target quenching once these NHS esters enter cells. At a minimum, it would be helpful to ask if an orthogonal approach yields corroborating results. For example, something like biocompatible condensation reactions as shown here <https://onlinelibrary.wiley.com/doi/10.1002/anie.200903627> or here <https://www.nature.com/articles/nchem.480>.

3) In what sense is Pitchoune an amphiphile? Formally, this refers to either a surfactant-like structure or a block copolymeric organization. It would help to justify this nomenclature and the underlying concepts. Also, the simulations do not call for an amphiphile. What they

introduce as a spreading coefficient was discussed in Ref. 8. The simulations themselves are not especially novel. Similar designs have been implemented in other studies including Ref. 8. And these simulations do not arrive at the description of an amphiphile. Essentially, what the simulations show is that there is a specific interaction scheme that generates the architecture they desire. For what it is worth, the work in Ref. 8 shows that there are numerous interaction schemes that are compatible with a core-shell structure. To arrive at the most reasonable interaction matrix, one needs additional constraints such as measurements of the pairwise interfacial tensions or pairwise excluded volumes. It is difficult to see how the simulations predict the existence of a so-called amphiphile as claimed. In the simulations, the amphiphile is generated by differential binding to different regions (see Fig. 4b). Does Pitchoune show similar effects?

4) Overall, despite the narrative, it does not follow that the simulations predict the existence of Pitchoune nor do they identify this as a candidate. It is important to a) highlight other simulations that have arrived at very similar results, b) emphasize that the simulations provide a plausible explanation for the phenomenology and c) explain why, as asserted in the MS, they believe that Pit is an amphiphile. In general, this verbiage is distracting and misleading. Essentially, differential interaction preferences, without imposing the structural connotations of an amphiphile are sufficient to arrive at the architecture they observe. Differential affinities do not make for an amphiphile in the conventional sense. This should be cleaned up.

5) Finally, the main concern is that a very specific mechanistic picture emerges from the observations made via live cell imaging. Can these observations be recapitulated in vitro? If not, then a purely equilibrium picture, which may well be valid, needs a more cautious presentation. Of course, if in vitro reconstitutions are feasible using HP1a, and 3-4 nucleolar components, that would be clinching evidence. This is likely to be a serious challenge, and hence appropriate caution might be needed.

Overall, this is an interesting paper. The scholarship is weak in terms of the citations and apportioning credit to previous observations. Whether this is remedied or not is the prerogative of the authors. The title, with the term amphiphile is misleading. What is being observed is more nuanced. Whether one can make a definitive case for the observed organization sans a bona fide GC marker is unclear. The simulations provide a phenomenological rationalization. They do not yield predictions. This should be made clear. With appropriate revisions, this paper deserves to be published in NCB and will almost certainly animate a lot of interesting discussions.

Reviewer #2 (Remarks to the Author):

Summary:

Nucleoli and pericentric heterochromatin (PCH) are known to colocalize at the apical side of embryonic nuclei in *Drosophila*. However, the nature of this interaction has not been characterized. In this study, the authors perform time lapse imaging to characterize the dynamic nature of these interactions and their dependence on ribosomal DNA for nucleation. The authors present a coarse-grained model to explain nucleolar-heterochromatin interactions. Finally, the authors show that Pitchoune, a nucleolar component, has the characteristics of an amphiphilic protein that fits one of the requirements of this model.

Strengths:

1. The authors characterize the changing interactions between the nucleolus and the pericentric heterochromatin (PCH) at different stages of *Drosophila* development.
2. The authors propose a simple intuitive model that can predict the organization of the nucleolus and PCH and recapitulates certain phenotypes.
3. The authors show that Pitchoune, an essential nucleolar protein, mediates the nucleolus-PCH interaction, probably via HP1.

Major points to address:

1. The model proposed by the authors assumes the presence of an amphiphilic molecule to explain steady states. The identification of such a protein is not sufficient proof, and the authors should perform more extensive perturbations (such as partial and double knockdowns in S2 cells) to test their model.

Minor points to address:

1. The authors characterize the appearance of PCH loci in the absence of rDNA as one of increased compaction (Figure 2). Although the change in the shape of the PCH loci is clear (Figure 2c,d), it remains uncertain whether the decreased distance between different repeats (Figure 2e,f) is simply due to changes in organization of the PCH loci around the neocondensate. If the authors believe that the PCH is indeed functionally more compact, it would be useful to test whether the expression of genes/repeats in the PCH is reduced due to the absence of rDNA. If not, it might be misleading to characterize this change in organization of PCH as overcompaction.
2. Although the simple intuitive model predicts the final organization of the nucleolus and PCH, it is unclear from the text if the authors believe that the earlier stage phenotypes can also be understood by this model, although they appear to be inconsistent with it. The authors also do not discuss any possible PCH-nuclear lamina interactions, although it might appear that such interactions would help explain the PCH phenotypes in earlier developmental stages.
3. The authors identify the interaction domain between Pitchoune and heterochromatin, however, the affinities for other PCH components are unclear. The authors hypothesize that interaction between Pitchoune and Fibrillarin might depend on ribosomal RNA, but no evidence for this is shown. This could be tested using Pol I inhibitors.

Conclusion:

The manuscript does a great job of characterizing the dynamic nature of the interactions between the nucleolus and heterochromatin. The authors also find a nucleolar protein with affinity for heterochromatin and demonstrate its importance in maintaining the nucleolar-PCH steady state. However, the proposed model for nucleolar-PCH interactions requires considerably more support to justify publication in a general journal such as NCB.

Reviewer #3 (Remarks to the Author):

Using both *Drosophila* embryos and the S2R+ *Drosophila* cell line with imaging techniques, the authors aim to dissect the interactions

between the nucleolus and the Pericentromeric Heterochromatin (PCH), two large nuclear condensates that are located within close proximity of each other, with the PCH enveloping the nucleolus. Initially this report defines the organisation of both condensates around each other during sequential stages of embryonic development, noting three distinct conformations where the PCH firstly is extended from, then surrounds and finally wraps around the nucleolus (RFP-HP1a and GFP-Fib for PCH and nucleolus respectively). Following removal of the rDNA, the nucleolus is lost and fibrillarin forms independent neo-condensates which are mostly localised separately from the PCH, with the PCH forming a tight toroidal structure around a non-fibrillarin containing protein rich centre in late development. Following these observations, they developed a physical model which orders nucleolar-PCH-amphiphilic protein interactions into a hierarchy of interaction strengths to predict the PCH-nucleolar conformation. To validate this model, the Dead box RNA helicase, Pitchoune (human DDX18/yeast Has1) was identified as a potential amphiphilic protein, facilitating interactions between the nucleolus and PCH. Indeed, Pitchoune localises to the protein rich PCH centres and when Pitchoune is depleted or mutated, the PCH envelope structure was lost, indicating a role for Pitchoune in the formation or stability of the PCH wrapped nucleolus conformation. Overall, this study defines early developmental interactions and conformations between the PCH and nucleoli and the importance of both the rDNA and Pitchoune for normal PCH organisation.

Main Critiques:

Towards the end, this paper really hammers home the interaction between Pitchoune and the PCH protein HP1a due to Pitchoune containing two PxVxL domains, which were previously defined as HP1a-interacting motifs. Whilst the data indicates some level of interaction, at no point are these two proteins shown to directly interact, i.e. through a co-IP, so I would reword these sections or show a co-IP with both WT and PxVxL mutant Pitchoune. Some examples where I think they over state their findings:

- Line 356 - I would say that the PxVxL region of Pitchoune influences the conformation of the HP1a containing PCH.
- Line 389 - Unless I missed it in the paper, they did not show differences in Pitchoune expression or localisation during the different cycles, so this conclusion/discussion seems like a reach and I would remove it or reword this section, again, a direct interaction between the two proteins was not defined.

Whilst removal of the rDNA does disrupt nucleolar formation, this technique likely disrupts the nucleus in many ways in addition to the loss of the nucleoli which could also disrupt PCH conformation. The authors mention that the PCH starts forming two cycles before the nucleolus – what is the conformation at this time? Are there pre-nucleolar condensates that form a structure? It would be an interesting way to further determine how the nucleolus impacts PCH assembly and assembly dynamics in physiological conditions instead of just minus rDNA. The minus rDNA could be affecting the condensates in multiple unknown ways to disrupt the PCH, especially since there is a lot of overlap in the box plots in Figure 2. Maybe use a pol1 inhibitor like BMH-21?

Figure 3C and Supplemental video 7:

- The 'donut' conformation looks similar to the normal PCH conformation to my untrained eye. It would be useful to include the xy, yz and xz views at t=0 and t=70 for with rDNA with and without Fib overlay for comparison.
- A number of the cells in this video (minus rDNA) have similar PCH conformation around the neo-condensates to what was previously shown with WT embryos contradicting Figure 3C. Please reconcile.

Minor Changes/Notes:

1. There is no reference to extended figure 3B anywhere in the text.
2. Line 127 – Without another nuclear marker, it is not confirmed if the PCH is 'lining the nuclear edge' -> I would move the part about figure 1D here as this backs this statement instead of having it at the end of the next paragraph.
1. Figure 3d/line 194 - They have confirmed there is protein in the PCH void; however, have not confirmed that 'no nucleoli proteins are present? They have just confirmed that fibrillarin may not be present (again, in the movie there were a few cells where fibrillarin WAS present). Indeed, they go on to show that Pit is in this void. Please reconcile.
2. Figure 2E – hard to see the DAPI stain here, could just be the printout though.
3. Figure 2A - -rDNA and +rDNA HP1a look quite similar to me, But I agree that there is an obvious difference in conformation in 2C.
4. I would just say in the absence of rDNA through section describing Figure 2, as mentioned above, the absence of rDNA may be affecting the nucleus and DNA/chromosome structure in multiple ways so any changes observed may not be purely due to loss of a nucleoli.
5. Line - 389 – Unless I missed it in the paper, the authors did not show differences in Pitchoune expression or localisation during the different cycles, so this conclusion/discussion seems like a reach for me and I would remove it or reword this section, again, a direct interaction between the two proteins is not characterised.

Paragraph from line 399:

6. I don't know where the data are that shows that Pitchoune and HP1a don't stably mix unless I have completely overlooked a figure.
7. Line 413 – Suggest using a Pol1 inhibitor here. Again, they stated that the rDNA being in close proximity to the 359bp region may be required for the proximity of the PCH and nucleoli, so maybe just deleting this region is enough to disrupt the PCH conformation.

Reviewer #3 (Remarks on code availability):

There isn't any code that I could see--only equations.

Methods should be written concisely, but should contain all elements necessary to allow interpretation and replication of the results. As a guideline, Methods sections typically do not exceed 3,000 words. The Methods should be divided into subsections listing reagents and techniques. When citing previous methods, accurate references should be provided and any alterations should be noted. Information must be provided about: antibody dilutions, company names, catalogue numbers and clone numbers for monoclonal antibodies; sequences of RNAi and cDNA probes/primers or company names and catalogue numbers if reagents are commercial; cell line names, sources and information on cell line identity and authentication. Animal studies and experiments involving human subjects must be reported in detail, identifying the committees approving the protocols. For studies involving human subjects/samples, a statement must be included confirming that informed consent was obtained. Statistical analyses and information on the reproducibility of experimental results should be provided in a section titled "Statistics and Reproducibility".

All Nature Cell Biology manuscripts submitted on or after March 21 2016 must include a Data availability statement at the end of the Methods section. For Springer Nature policies on data availability see <http://www.nature.com/authors/policies/availability.html>; for more information on this particular policy see <http://www.nature.com/authors/policies/data/data-availability-statements-data-citations.pdf>. The Data availability statement should include:

- Accession codes for primary datasets (generated during the study under consideration and designated as "primary accessions") and secondary datasets (published datasets reanalysed during the study under consideration, designated as "referenced accessions"). For primary accessions data should be made public to coincide with publication of the manuscript. A list of data types for which submission to community-endorsed public repositories is mandated (including sequence, structure, microarray, deep sequencing data) can be found here <http://www.nature.com/authors/policies/availability.html#data>.
- Unique identifiers (accession codes, DOIs or other unique persistent identifier) and hyperlinks for datasets deposited in an approved repository, but for which data deposition is not mandated (see here for details <http://www.nature.com/sdata/data-policies/repositories>).
- At a minimum, please include a statement confirming that all relevant data are available from the authors, and/or are included with the

manuscript (e.g. as source data or supplementary information), listing which data are included (e.g. by figure panels and data types) and mentioning any restrictions on availability.

- If a dataset has a Digital Object Identifier (DOI) as its unique identifier, we strongly encourage including this in the Reference list and citing the dataset in the Methods.

We recommend that you upload the step-by-step protocols used in this manuscript to protocols.io. More details can found at <https://www.protocols.io/help/publish-articles>.

All imaging data should be accompanied by scale bars, which should be defined in the legend.

Cropped images of gels/blots are acceptable, but need to be accompanied by size markers, and to retain visible background signal within the linear range (i.e. should not be saturated). The boundaries of panels with low background have to be demarked with black lines. Splicing of panels should only be considered if unavoidable, and must be clearly marked on the figure, and noted in the legend with a statement on whether the samples were obtained and processed simultaneously. Quantitative comparisons between samples on different gels/blots are discouraged; if this is unavoidable, it should only be performed for samples derived from the same experiment with gels/blots were processed in parallel, which needs to be stated in the legend.

The total number of Supplementary Figures (not including the "unprocessed scans" Supplementary Figure) should not exceed the number of main display items (figures and/or tables (see our Guide to Authors and March 2012 editorial <http://www.nature.com/ncb/authors/submit/index.html#supinfo>; <http://www.nature.com/ncb/journal/v14/n3/index.html#ed>). No restrictions apply to Supplementary Tables or Videos, but we advise authors to be selective in including supplemental data.

GUIDELINES FOR EXPERIMENTAL AND STATISTICAL REPORTING

REPORTING REQUIREMENTS – To improve the quality of methods and statistics reporting in our papers we have recently revised the reporting checklist we introduced in 2013. We are now asking all life sciences authors to complete two items: an Editorial Policy Checklist (found here <https://www.nature.com/authors/policies/Policy.pdf>) that verifies compliance with all required editorial policies and a reporting summary (found here <https://www.nature.com/authors/policies/ReportingSummary.pdf>) that collects information on experimental design and reagents. These documents are available to referees to aid the evaluation of the manuscript. Please note that these forms are dynamic 'smart pdfs' and must therefore be downloaded and completed in Adobe Reader. We will then flatten them for ease of use by the reviewers. If you would like to reference the guidance text as you complete the template, please access these flattened versions at <http://www.nature.com/authors/policies/availability.html>.

Version 1:

Decision Letter:

Our ref: NCB-A55184A

20th August 2025

Dear Dr. Karpen,

Thank you for submitting your revised manuscript "Hierarchical interactions between nucleolar and heterochromatin condensates are mediated by a dual-affinity protein" (NCB-A55184A).

Please accept our sincerest apologies for the length of time your manuscript has been under consideration at our journal. This is because we have been persistently chasing Reviewer #2 for the report but unfortunately we have not yet received it. We will forward the report if we receive it in the future for your information. Nevertheless, based on the reports that we have currently and after discussion within the editorial team, we are now prepared to deliver our editorial decision.

It has now been seen by the original referees #1 and #3 and their comments are below. The reviewers find that the paper has improved in

revision, and therefore we'll be happy in principle to publish it in Nature Cell Biology, pending minor revisions to comply with our editorial and formatting guidelines.

The current version of your manuscript is in a PDF format. Please email us a copy of the file in an editable format (Microsoft Word or LaTeX)-- we can not proceed with PDFs at this stage.

Thank you again for your interest in Nature Cell Biology. Please do not hesitate to contact me if you have any questions.

Sincerely,
Daryl

Daryl Jason Verzosa David, PhD

Senior Editor, Nature Cell Biology
Advisory Editor, npj Biological Physics and Mechanics
Nature Portfolio

Heidelberger Platz 3, 14197 Berlin, Germany
Email: daryl.david@nature.com
ORCID: <https://orcid.org/0000-0002-9253-4805>

Reviewer #1 (Remarks to the Author):

The authors have revised their manuscript extensively. Their revisions are fully responsive and I have no further revisions to recommend. This will be an important contribution and should be published post haste in NCB.

Reviewer #3 (Remarks to the Author):

This is a re review of a paper reviewed previously from Rajshekar et al and the Karpen laboratory. They have done an excellent job of responding to everyone's comments leading to a much improved manuscript. This is OK to publish.

Reviewer #3 (Remarks on code availability):

We do not have this expertise.

Version 2:

Decision Letter:

Dear Dr Karpen,

I am pleased to inform you that your manuscript, "Hierarchical interactions between nucleolar and heterochromatin condensates are mediated by a dual-affinity protein", has now been accepted for publication in Nature Cell Biology.

Please note that *Nature Cell Biology* is a Transformative Journal (TJ). Authors may publish their research with us through the traditional subscription access route or make their paper immediately open access through payment of an article-processing charge (APC). Authors will not be required to make a final decision about access to their article until it has been accepted. [Find out more about Transformative Journals](https://www.springernature.com/gp/open-research/transformative-journals)

Authors may need to take specific actions to achieve compliance with funder and institutional open access mandates. If your research is supported by a funder that requires immediate open access (e.g. according to [Plan S principles](https://www.springernature.com/gp/open-science/plan-s-compliance) or the [NIH public access policy](https://www.springernature.com/gp/open-science/us-federal-agency-compliance)) then you should select the gold OA route, and we will direct you to the compliant route where possible. Because authors warrant under our subscription licensing terms that they haven't committed to licensing any version of their article under a licence inconsistent with the terms of our agreement – including the applicable embargo period – publication under the subscription model isn't suitable for authors whose funders require no embargo.

If you have not already done so, we strongly recommend that you upload the step-by-step protocols used in this manuscript to protocols.io (<https://protocols.io>), an open online resource that allows researchers to share their detailed experimental know-how. All uploaded protocols are made freely available and are assigned DOIs for ease of citation. Protocols and Nature Portfolio journal papers in which they are used can be linked to one another, and this link is clearly and prominently visible in the online versions of both. Authors who performed the specific experiments can act as primary authors for the Protocol as they will be best placed to share the methodology details, but the Corresponding Author of the present research paper should be included as one of the authors. By uploading your Protocols onto protocols.io, you are enabling researchers to more readily reproduce or adapt the methodology you use, as well as increasing the visibility of your protocols and papers. You can also establish a dedicated workspace to collect your lab Protocols. Further information can be found at <https://www.protocols.io/help/publish-articles>.

Nature Cell Biology encourages authors presenting evidence for cell, biological, molecular, and genetic interactions to consider communicating these findings using Biofactoid (<https://biofactoid.org/>). This tool helps users share a searchable representation of interactions (e.g. binding, gene expression, post-translational modification) between genes, gene products, or chemicals. Information added to Biofactoid, with author attribution, is shared on social media and public databases, such as Pathway Commons, where it can be discovered and analyzed in the context of a large and growing corpus of knowledge.

With kind regards,

Daryl

Daryl Jason Verzosa David, PhD

Senior Editor, Nature Cell Biology
Advisory Editor, npj Biological Physics and Mechanics
Nature Portfolio

Heidelberger Platz 3, 14197 Berlin, Germany
Email: daryl.david@nature.com
ORCID: <https://orcid.org/0000-0002-9253-4805>

** Visit the Springer Nature Editorial and Publishing website at http://editorial-jobs.springernature.com?utm_source=ejP_NCB_email&utm_medium=ejP_NCB_email&utm_campaign=ejp_NCB for more information about our career opportunities. If you have any questions please click [here](mailto:editorial.publishing.jobs@springernature.com).

Manuscript Title (Edited): Hierarchical interactions between nucleolar and heterochromatin condensates are mediated by a dual-affinity protein

Manuscript ID: NCB-A55184

Dear Editor and Reviewers,

Thank you for the detailed comments on our manuscript. The additional experimental and simulation results motivated by this constructive feedback, we believe, have strengthened and extended the main findings of this study and improved the clarity of the manuscript. We hope these revisions sufficiently address the reviewers' concerns and that the revised manuscript will be recommended for publication.

In this revised version, we describe the *de novo* assembly dynamics of pericentromeric heterochromatin (PCH) around the outermost layer of the multiphasic nucleolus during *Drosophila* embryonic development. We confirm that the DEAD-Box RNA Helicase Pitchoune (Pit) is a 'dual-affinity' protein that mediates PCH organization around the nucleolus by interacting with both nucleolar components and HP1a. By adding a combination of live imaging, proximity labeling, co-immunoprecipitation, *in vitro* binding assays, additional simulations, and Pol-I inhibitor drug treatments, we now demonstrate that Pit directly interacts with HP1a via a conserved PxVxL motif. Pit also exhibits domain-specific interactions with nucleolar components that determine its sub-nucleolar localization. Importantly, modulating Pit affinity hierarchies with the nucleolus or HP1a altered PCH-nucleolar 3D organization. Our results provide a molecular mechanism by which two compositionally distinct, immiscible condensates interact and influence each other's 3D organization *in vivo*.

Below, we provide a point-by-point response addressing the editor's overview and each reviewer's comment.

Response to the Editor

A) Assess potential direct interactions of PCH with other components which would include new experiments in vitro (Reviewer #1) and in cell culture (Reviewers #2 and #3)

An important conclusion from our original submission was that Pit functions as a dual-affinity protein interacting with both nucleolar and PCH components, based on microscopy and modeling phenotypes of *in vivo* genetic perturbations. A key concern raised by the reviewers was the lack of direct evidence for Pit's interactions with PCH. To address this criticism, we have included a new main figure and associated extended Data that establishes the molecular basis of Pit's interactions with heterochromatin (**Fig. 5 and Extended Data Fig. 6**). We assessed interactions between HP1a and Pit in cell culture (via proximity ligation assays, **Fig. 5a** and co-immunoprecipitation, **Fig. 5b**), *in silico* using AlphaFold multimer (**Fig. 5c**), and *in vitro* using pull down binding assays (**Fig. 5d**). These experiments demonstrate a direct, weak interaction between Pit and HP1a, which is eliminated when a conserved HP1a interaction motif in Pit, or HP1a dimerization are

mutated. Together, these results provide strong molecular evidence for Pit's direct interactions with PCH via HP1a, explaining its essential role in mediating PCH-nucleolar interactions.

B) Provide further experimental evidence to assess the effects, off-target or otherwise, of -rDNA perturbations (Reviewers #2 and #3)

We now cite additional relevant results from Falahati & Wieschaus 2017 on the localization of different DFC and GC nucleolar proteins in -rDNA embryos at Cycle 14. We have revised **Extended Data Fig. 3** to clearly show the changes in organization of DAPI and chromatin in -rDNA embryos, the main phenotype being increased PCH compaction and subsequent reorganization to a toroid-like structure. To assess potential off-target effects, we also examined the localization of another nuclear condensate, the Histone Locus Body, which appeared to be unaffected in -rDNA embryos (**Extended Data Fig. 3e**).

As a complementary approach to rDNA deletions, we used low concentrations of Actinomycin D in S2R+ *Drosophila* cells to assess the impact of blocking RNA polymerase I activity and nascent rRNA synthesis on nucleolar and PCH organization. This perturbation does not fully phenocopy the -rDNA deletion phenotypes reported in Fig. 2 and Fig. 4 (i.e., Fib displacement to a neocondensate and Pit filling the PCH void), due to the persistence of rRNA produced before treatment. Importantly, it does cause phenotypes consistent with the dual-affinity model proposed in this study. Specifically, PolI inhibition reduces Pit's nucleolar associations while preserving Pit-PCH interaction. This causes Pit to redistribute and become enriched at the nucleolar periphery next to HP1a but diminished from the nucleolar interior. These results are presented in **Fig. 6h-j**, **Extended Data Fig. 9** and associated main text.

C) Justify the parameters and predictions from the simulations (Reviewers #1 and #3)

In the revised manuscript, we have edited the main text to clarify that the central idea motivating our coarse-grained simulations is that a hierarchy of interfacial tensions governs the layered organization of the multi-phasic nucleolus and PCH. Specifically, phases with higher interfacial tension localize more internally in a layered condensate architecture. The parameter choices in our coarse-grained model for PCH-nucleolar organization are constrained by previous work characterizing the multiphasic nature of the nucleolus (Falahati & Wieschaus 2017; Feric et al. 2016), and our experimental observations in the -rDNA embryos, importantly, the formation of a protein-rich domain within PCH and the formation of Fibrillarin-rich neocondensates. These features limit the possible range of interaction strengths in our model and a detailed explanation of how absolute parameter values were chosen is described in the Methods section under "Rationale for the choice of parameters in the coarse-grained model", with appropriate language edits to the main text. Simulation outcomes that deviate from the chosen parameters are also presented in **Extended Data Fig. 4**. To improve transparency and readability, we now include interaction matrices alongside every simulation panel in the

figures for easier interpretation of the modeled interactions. We have also revised all modeling-associated Figure legends to make the parameter choices and simulation outcomes clearer.

In response to reviewer feedback, we have also edited our terminology and no longer use “amphiphilic” to describe Pit as it caused more confusion. Instead, we refer to it as a “dual-affinity” protein, due to its experimentally backed property of interacting with both nucleolus components and HP1a. Moreover, motivated by experimental data showing the changes in Pit’s sub-nucleolar localization in different mutant conditions, we have expanded our model to include additional simulations that test how modulating the attraction of Pit with PCH or with rRNA impact Pit localization profiles within the nucleolus, and nucleolar organization relative to PCH in nuclei (**Fig. 6f-g and Extended Data Fig. 8a-b**). Together, these results show how a differential tension model can explain the joint organization of the nucleolus and PCH in normal and mutant conditions.

We hope our revised text, detailed description in the Methods, updated Figures and Figure legends, and expanded simulations effectively address reviewer concerns on justifying parameter choices and predictions that align with experimental outcomes.

D) Experimentally assess localization and organization of other nucleolar components (Reviewer #1)

Reviewer #1 raised concerns about using Modulo (Mod) as a bona fide GC marker. We now provide citations, as well as localization data, proving its localization as a GC protein (**Extended Data Fig. 1a-c**). Additionally, we expand our analysis of Pit’s localization patterns as a GC protein and include additional data of its localization relative to other nucleolar sub-compartments (**Extended Data Fig. 5a**). We have also included new data on the localization of different nucleolar components relative to HP1a in *Drosophila* embryos and S2R+ *Drosophila* cell culture, including an RNA-Pol-1 subunit as an FC marker, Nopp140 and Fib as DFC markers, and an additional GC marker Ns1. Consistent with a bipartite nucleolar organization reported in *Drosophila* (Knibiehler et al. 1982), Polr1E (FC) and Fib (DFC) colocalized and were nested within Mod (GC). These new data are included in **Extended Data Fig. 1**. Although separate FC and DFC compartments may exist below the resolution of our imaging, we refer to this layer as FC/DFC and visualize it using Fib through the rest of the manuscript.

E) All other referee concerns have been addressed in our point-by-point response below.

F) Additional data

- a. A Supplementary Figure including unprocessed Data for all gels and blots is included (**RajshekarS_2025_Supplementary Figure.pdf**)
- b. All numerical source data is provided in an Excel File (**RajshekarS_2025_Supplementary Table.xlsx**)

- c. Complete Editorial Checklist has been provided (**RajshekarS_2025_nr-editorial-policy-checklist.pdf** and **RajshekarS_2025_nr-reporting-summary.pdf**)

Response to Reviewers

Reviewer #1

*In this work, the authors present their findings of the spatial organization of the PCH and the nucleolus with respect to one another. These results come from studies in *Drosophila* at different stages of embryogenesis. The key finding is that the PCH organizes around the nucleolus, and that this organization is disrupted when the nucleolus is removed. The authors argue that Pitchoune is an amphiphilic protein that facilitates the organization of the PCH around the nucleolus. Overall, this is an interesting and timely study that will animate a lot of healthy and vibrant discussion. It certainly merits publication in *Nature Cell Biology*. However, there are several issues that need to be addressed on the technical, conceptual, and scholarship side. These are listed below.*

- 1) Modulo is a paralog of nucleolin. However, nucleolin, at least in *Xenopus* oocytes and in mammalian cells, is a marker of the DFC and not the GC. NPM1 is the most widely used marker of the GC, and in *Drosophila* this would be nucleoplasmin like protein (NLP). Please see <https://doi.org/10.1093/nar/gky988>. Experts in the nucleolus field will likely be interested in whether nucleolin behaves very differently in flies or if a different marker such as NLP should be used. Either the biology should provide a sound case for not using NLP or it should be used to make a definitive case for the organization that the authors are putting forth. Most of the images only show HP1a and Fibrillarin markers, and so it is unclear if the PCH is layered atop the GC or the DFC.*

We provide supporting citations and additional data demonstrating Modulo as a GC marker in *Drosophila* (**Extended Data Fig. 1a-c**). While the reviewer suggests the use of NLP, Nucleoplasmin-like protein, in fly cells it appears to colocalize with centromeres (Fig 1B in Padeken et al., 2013) and not the nucleolus, thereby making it unsuitable as a GC marker. We acknowledge that most Figures from our initial submission primarily showed Fibrillarin as a marker for the nucleolus, making HP1a localization patterns relative to the nucleolus less clear. We have therefore included additional data and changed the relevant sections in the main text to clarify HP1a localization relative to the different nucleolar sub compartments, including additional FC (Polr1E), DFC (Nopp140) and GC (Ns1) markers (**Fig. 1a and Extended Data Fig. 1**, also please refer to Response to Editor, Point D).

- 2) The use of 488 NHS ester treatment is fraught with concerns. This is being used to assert that the apparent void in the toroidal structures is filled with proteins. Typically, efficient and reliable NHS ester labeling of proteins requires buffers of pH 8.3 or thereabouts. How can one rule out that the labeling is not of free amines in the apparent void? The naive expectation is of a lot of off-target quenching once these*

NHS esters enter cells. At a minimum, it would be helpful to ask if an orthogonal approach yields corroborating results. For example, something like biocompatible condensation reactions as shown here <https://onlinelibrary.wiley.com/doi/10.1002/anie.200903627> or here <https://www.nature.com/articles/nchem.480>.

The 488 NHS Ester has been validated as a pan-protein label for use in tissues (M'Saad & Bewersdorf 2020). In our study, we used it to assess whether the PCH void is protein-rich, with the nucleolus serving as a positive control. While this does not rule out the possibility that the void accumulates free amines, or other components, we used live microscopy as an orthogonal approach to confirm the accumulation of full-length Pit protein in the PCH void (**Fig. 4**). Our goal in using the 488 NHS Ester was to broadly determine which type of macromolecule could be accumulating in the void. Also, implementing the biocompatible condensation reactions suggested by the reviewer for use inside nuclei in *Drosophila* embryos was anticipated to be technically challenging within the timeline of revisions. Most importantly, although we cannot rule out the presence of other proteins/molecules, these results (and the modeling) were only used to lead us to identify the putative Protein X, and subsequent experimental results clearly demonstrate Pit's enrichment in the void in -rDNA embryos and its role in mediating PCH-nucleolar associations.

3) In what sense is Pit an amphiphile? Formally, this refers to either a surfactant-like structure or a block copolymeric organization. It would help to justify this nomenclature and the underlying concepts. Also, the simulations do not call for an amphiphile. What they introduce as a spreading coefficient was discussed in Ref. 8. The simulations themselves are not especially novel. Similar designs have been implemented in other studies including Ref. 8. And these simulations do not arrive at the description of an amphiphile. Essentially, what the simulations show is that there is a specific interaction scheme that generates the architecture they desire. For what it is worth, the work in Ref. 8 shows that there are numerous interaction schemes that are compatible with a core-shell structure. To arrive at the most reasonable interaction matrix, one needs additional constraints such as measurements of the pairwise interfacial tensions or pairwise excluded volumes. It is difficult to see how the simulations predict the existence of a so-called amphiphile as claimed. In the simulations, the amphiphile is generated by differential binding to different regions (see Fig. 4b). Does Pit show similar effects?

In our model, we showed that the core-shell structure of the Nucleolus (core) and PCH (shell) can only be achieved if 'Protein X' has attractive interactions with both the nucleolus and PCH. This motivated our initial choice of the term 'amphiphile' (etymology: likes both) to describe Pit. However, we acknowledge the referee's point that the term amphiphile can attribute surfactant-like properties to Pit, which we don't have evidence for. To avoid confusion, we now use the term 'dual-affinity' protein to describe Pit, based

on experimental evidence that shows its differential interactions with both nucleolar components and HP1a (**Figs. 4-6**). We use a 'differential interfacial tensions' framework to describe the system and begin the main text on the modeling results by situating our work in the context of results from Feric et al. 2016 and others. We used spreading coefficients to describe the relative interfacial tensions between phases, because this gives us a generic choice of interaction matrices, not a unique one. We find that as long as the chosen parameters satisfy the hierarchy of interfacial tensions, we will obtain the multilayered organization of the nucleolus and PCH. We also show simulation snapshots for different parameter choices that deviate from the proposed interaction hierarchy and do not form the experimentally observed 'surrounded' organization (**Extended Data Fig. 4**).

While we agree with the reviewer that multiple interaction schemes can lead to core-shell structures, our parameter choices were guided and strongly constrained by the following experimental observations emerging from this study and previous work. First, Fibrillarin undergoes phase separation in the absence of rDNA (rRNA), and in the presence of rDNA is always localized to the inner nucleolar layers, suggesting a high interfacial tension between the fibrillarin-rich phase and the nucleoplasm (Falahati et al. 2016; Feric et al. 2016). Second, we observed a dense compartment of proteins emerging within the PCH "void" in -rDNA embryos, suggesting the existence of additional components that interact with PCH and can form a dense phase. Third, PCH forms via phase separation and associates with the nucleolus, suggesting attractive interactions between their constituents. Fourth, neither Fibrillarin nor Modulo interact directly with PCH in -rDNA embryos and are absent from the "PCH void". These constraints led us to hypothesize that the protein within the PCH void could be a GC protein that can phase separate (self-associate) and has affinity for PCH, which allowed us to identify Pit as a candidate.

In this revised version, we provide evidence for the molecular basis for Pit's dual affinity for the PCH and nucleolus and show that these interactions mediate nucleolar-PCH associations (**Fig. 5-6**). Additionally, after characterizing the localization patterns and affinities of the different protein domains of Pit, we incorporated these features explicitly into a refined model by adding an interaction between Pit and rDNA (to account for the helicase domain's rRNA interactions) (**Fig. 6f-g and Extended Data Fig. 8**). Together these results support a model where interfacial tension hierarchies and the dual-affinity of Pit mediate the associations between the nucleolus and PCH.

4) Overall, despite the narrative, it does not follow that the simulations predict the existence of Pit nor do they identify this as a candidate. It is important to a) highlight other simulations that have arrived at very similar results, b) emphasize that the simulations provide a plausible explanation for the phenomenology and c) explain why, as asserted in the MS, they believe that Pit is an amphiphile. In general, this verbiage is distracting and misleading. Essentially, differential interaction preferences, without imposing the structural connotations of an amphiphile are sufficient to arrive

at the architecture they observe. Differential affinities do not make for an amphiphile in the conventional sense. This should be cleaned up.

We agree that the simulations alone do not predict the existence of Pit, nor can they identify it as a candidate protein. Rather, the simulations rooted in the theory of the coexistence of three phases (Torza & Mason 1969), helped us define the properties for a molecule that would be enriched within PCH in the -rDNA case, and also be a nucleolar component in the +rDNA condition. Specifically, our modeling results suggested that the PCH-nucleolar organizational phenotypes that we observed in +rDNA and -rDNA nuclei could be explained if we included a protein exhibiting differential affinities for both compartments.

These constraints led us to identify Pit as a candidate, based on imaging results reported in Falahati & Wieschus 2017, where the authors showed that in -rDNA cycle 14 nuclei, Pit is enriched in a region which we suspected to coincide with PCH, suggesting a weak interaction for Pit with PCH. This observation converging with the predictions of our model led us to experimentally test the role of Pit in mediating PCH-nucleolar associations. In the revised version of this manuscript, we have tried to clarify this reasoning and highlighted prior modeling work that invokes the differential tension model to explain the multiphasic nature of the nucleolus (Feric et al. 2016).

We also understand the reviewer's concern that differential affinities do not make for an amphiphile. Rooted in experimental evidence included in the revisions, we now refer to Pit as a dual-affinity protein and identify the molecular basis of its differential interactions with the nucleolus and PCH. We show that the N-term disordered domain of Pit is sufficient for nucleolar localization, helicase activity drives rRNA mediates localization to the nucleolar center, and a conserved PxVxL motif is required for HP1a associations (**Fig. 5-6**). To further strengthen the model, we have added simulations that account for the specific interactions of Pit's sub-domains and show that modifying Pit interactions with either PCH or nucleoli recapitulate organizational patterns observed in cells (**Fig 6**).

5) Finally, the main concern is that a very specific mechanistic picture emerges from the observations made via live cell imaging. Can these observations be recapitulated in vitro? If not, then a purely equilibrium picture, which may well be valid, needs a more cautious presentation. Of course, if in vitro reconstitutions are feasible using HP1a, and 3-4 nucleolar components, that would be clinching evidence. This is likely to be a serious challenge, and hence appropriate caution might be needed.

In vitro reconstitutions using minimal components is a long-term goal but is likely to require extensive troubleshooting that goes beyond the scope of this manuscript. Instead, we addressed concerns about whether Pit interacts with HP1a that are more direct than live imaging observations, specifically *in vitro* binding assays, co-immunoprecipitation, and proximity ligation, as well as the impact of eliminating HP1a interactions *in vivo* and *in vitro*. A new figure (**Fig. 5**) presents these results, showing that HP1a and Pit interact

and the PxVxL mutation in Pit or the dimerization mutant in HP1a disrupts the interaction (Fig. 5d).

Overall, this is an interesting paper. The scholarship is weak in terms of the citations and apportioning credit to previous observations. Whether this is remedied or not is the prerogative of the authors. The title, with the term amphiphile is misleading. What is being observed is more nuanced. Whether one can make a definitive case for the observed organization sans a bona fide GC marker is unclear. The simulations provide a phenomenological rationalization. They do not yield predictions. This should be made clear. With appropriate revisions, this paper deserves to be published in NCB and will almost certainly animate a lot of interesting discussions.

We thank the reviewer for their enthusiasm for our work and hope that the revisions described above satisfy their recommendation for publication. We were unclear about which specific aspects related to scholarship required improvement but agree this is important and have improved on citations of prior knowledge throughout the revised manuscript.

Reviewer #2

Summary:

Nucleoli and pericentric heterochromatin (PCH) are known to colocalize at the apical side of embryonic nuclei in Drosophila. However, the nature of this interaction has not been characterized. In this study, the authors perform time lapse imaging to characterize the dynamic nature of these interactions and their dependence on ribosomal DNA for nucleation. The authors present a coarse-grained model to explain nucleolar-heterochromatin interactions. Finally, the authors show that Pitchoune, a nucleolar component, has the characteristics of an amphiphilic protein that fits one of the requirements of this model.

Strengths:

- 1. The authors characterize the changing interactions between the nucleolus and the pericentric heterochromatin (PCH) at different stages of Drosophila development.*
- 2. The authors propose a simple intuitive model that can predict the organization of the nucleolus and PCH and recapitulates certain phenotypes.*
- 3. The authors show that Pitchoune, an essential nucleolar protein, mediates the nucleolus-PCH interaction, probably via HP1.*

Major points to address:

The model proposed by the authors assumes the presence of an amphiphilic molecule to explain steady states. The identification of such a protein is not sufficient proof, and the authors should perform more extensive perturbations (such as partial and double knockdowns in S2 cells) to test their model.

We agree with this criticism and in response we now present additional lines of experimentation to strengthen our conclusions:

- 1) We have added a new main figure that characterizes the molecular basis of the dual-affinity properties of Pit (**Fig. 5**)
 - a. We performed transfections of constructs containing Pit subdomains to determine their respective contributions to Pit's intrinsic affinities for the nucleolus or PCH (**Extended Data Fig. 5b**).
 - b. We added new experimental evidence that demonstrates that Pit interacts with HP1a both in cells (via proximity labeling: **Fig 5a** and co-IP: **Fig 5b**) and *in vitro* (via a pull-down binding assay with recombinant proteins: **Fig 5d**).
 - c. The above findings complement the data in our original submission that Pit knockdown causes a decrease in PCH organization around the nucleolus, which is rescued by wild-type Pit but not when the HP1a-binding PxVxL motif is mutated (**Fig. 6a-e**). These perturbations revealed the role of Pit-HP1a interactions in PCH-nucleolar organization.
- 2) As an additional perturbation, we treated S2R+ *Drosophila* cells with a low dose of ActD, a Pol-I inhibitor, to block the transcription of nascent rRNA. This treatment led to reorganization of PCH and nucleolar proteins Fib, Pit and Modulo. Importantly, Pit localization was reduced within the nucleolus and accumulated at the PCH-nucleolar interface. These results are consistent with the predictions of the dual-affinity model, where ActD treatment reduced Pit-rRNA interactions, allowing increased Pit-HP1a interactions. These data and their implications are presented in **Fig. 6h-j**, **Extended Data Fig. 9** and relevant main text.

Minor Points to Address:

- 1) *The authors characterize the appearance of PCH loci in the absence of rDNA as one of increased compaction (Figure 2). Although the change in the shape of the PCH loci is clear (Figure 2c,d), it remains uncertain whether the decreased distance between different repeats (Figure 2e,f) is simply due to changes in organization of the PCH loci around the neocondensate. If the authors believe that the PCH is indeed functionally more compact, it would be useful to test whether the expression of genes/repeats in the PCH is reduced due to the absence of rDNA. If not, it might be misleading to characterize this change in organization of PCH as overcompaction.*

In the absence of rDNA, we observed that satellite repeats are located significantly closer to each other compared to +rDNA embryo nuclei, as well as decreased aspect ratio of the entire domain (**Fig. 2**). We believe these results are sufficient to warrant the conclusion that loss of rDNA/nucleoli results in increased compaction of PCH DNA/chromatin, and that normal structural organization of PCH requires nucleoli. The reviewer suggested assessing whether PCH compaction has functional consequences by investigating changes in heterochromatin transcription in the -rDNA condition. We agree this is an exciting question and is in fact the focus of ongoing long-term studies of the reciprocal impact of heterochromatin-nucleolar associations on their functions. These experiments

are now possible due to the tools and insights generated in this study but go beyond the scope of this manuscript.

2) *Although the simple intuitive model predicts the final organization of the nucleolus and PCH, it is unclear from the text if the authors believe that the earlier stage phenotypes can also be understood by this model, although they appear to be inconsistent with it. The authors also do not discuss any possible PCH-nuclear lamina interactions, although it might appear that such interactions would help explain the PCH phenotypes in earlier developmental stages.*

This is a very interesting question but is difficult to resolve using our coarse-grained model. As discussed in text (Lines 495-505), accounting for the early stage extended configuration requires the inclusion of 1) lower PCH-nucleolar (Pit-HP1a) affinity and 2) higher PCH-lamina association in cycle 14 (vs 17). The initial states of PCH being extended along the nuclear edge are transient and our model can only explain equilibrium states. We believe that carefully studying these transitions will require more detailed models of the dynamics of these macromolecules compared with the simple Langevin dynamics we used, as well as more in-depth experimental analyses.

3) *The authors identify the interaction domain between Pit and heterochromatin, however, the affinities for other PCH components are unclear. The authors hypothesize that interaction between Pit and Fibrillarin might depend on ribosomal RNA, but no evidence for this is shown. This could be tested using Pol I inhibitors.*

In this revised version, we add a new main figure (**Fig. 5**) focused on the molecular basis of Pit's interactions with HP1a. Using co-IPs and in vitro pull-down binding assays we demonstrate a direct interaction between Pit and HP1a. Further, we show mutating Pit's HP1a-interaction motif, disrupted HP1a binding *in vitro* and PCH-nucleolar associations *in vivo*, suggesting that Pit-HP1a interactions are necessary for the organization of PCH around the nucleolus. Although we have not yet systematically studied Pit's interactions with other PCH components, our results establish a strong foundation for future studies to study the implications of recruiting other PCH components to the PCH-nucleolar interface mediated by Pit.

Pol-I inhibitor experiments were suggested by two reviewers and now included in the main-text (**Fig. 6h-j** and **Extended Fig. 9**) (Refer response to Rev 2, Major Point 1 and Rev 3, Major Point 4). The manuscript was also carefully revised to reflect the observed experimental outcomes.

Conclusion:

The manuscript does a great job of characterizing the dynamic nature of the interactions between the nucleolus and heterochromatin. The authors also find a nucleolar protein with affinity for heterochromatin and demonstrate its importance in maintaining the nucleolar-PCH steady state. However, the proposed model for nucleolar-PCH

interactions requires considerably more support to justify publication in a general journal such as NCB.

Reviewer # 3

Using both Drosophila embryos and the S2R+ Drosophila cell line with imaging techniques, the authors aim to dissect the interactions between the nucleolus and the Pericentromeric Heterochromatin (PCH), two large nuclear condensates that are located within close proximity of each other, with the PCH enveloping the nucleolus. Initially this report defines the organisation of both condensates around each other during sequential stages of embryonic development, noting three distinct conformations where the PCH firstly is extended from, then surrounds and finally wraps around the nucleolus (RFP-HP1a and GFP-Fib for PCH and nucleolus respectively). Following removal of the rDNA, the nucleolus is lost and fibrillarin forms independent neo-condensates which are mostly localised separately from the PCH, with the PCH forming a tight toroidal structure around a non-fibrillarin containing protein rich centre in late development. Following these observations, they developed a physical model which orders nucleolar-PCH-amphiphilic protein interactions into a hierarchy of interaction strengths to predict the PCH-nucleolar conformation. To validate this model, the Dead box RNA helicase, Pitchoune (human DDX18/yeast Has1) was identified as a potential amphiphilic protein, facilitating interactions between the nucleolus and PCH. Indeed, Pitchoune localises to the protein rich PCH centres and when Pitchoune is depleted or mutated, the PCH envelope structure was lost, indicating a role for Pitchoune in the formation or stability of the PCH wrapped nucleolus conformation. Overall, this study defines early developmental interactions and conformations between the PCH and nucleoli and the importance of both the rDNA and Pitchoune for normal PCH organisation.

Main Critiques:

- 1) Towards the end, this paper really hammers home the interaction between Pit and the PCH protein HP1a due to Pit containing two PxVxL domains, which were previously defined as HP1a-interacting motifs. Whilst the data indicates some level of interaction, at no point are these two proteins shown to directly interact, i.e. through a co-IP, so I would reword these sections or show a co-IP with both WT and PxVxL mutant Pit.*

We agree and now include three experiments (co-IP assays in cells, in vitro binding with purified proteins, and proximity labeling in cells) that confirm a direct physical interaction between Pit and HP1a, and its dependence on the PxVxL motif (new **Fig. 5**) (also see response to Rev 2, Major Point 1).

- 2) Line 356 - I would say that the PxVxL region of Pit influences the conformation of the HP1a containing PCH.*

We now include data to show a direct interaction between Pit and HP1a mediated via the PxVxL motif of Pit (**Fig 5**) and edit the main text accordingly.

3) - Line 389 - *Unless I missed it in the paper, they did not show differences in Pit expression or localisation during the different cycles, so this conclusion/discussion seems like a reach and I would remove it or reword this section, again, a direct interaction between the two proteins was not defined.*

Line 389/ New Line 495: This sentence expressed our speculation about how Pit-HP1a interactions may be impacting PCH dynamics in the early embryo. We have now edited the sentence to clearly express our interpretation and speculation based on the following experimental evidence now included in the revised version: (i) biochemical proof for direct interactions between Pit and HP1a (**Fig. 5d**), (ii) emergence of the Pit neocondensate in -rDNA embryos at the same developmental cycle as PCH stably enters the 'surrounded' conformation in +rDNA embryos (**Fig. 4b**), (iii) decrease in HP1a occupancy around the nucleolus when Pit's HP1a-interaction motif is mutated (**Fig. 6c-d**).

4) *Whilst removal of the rDNA does disrupt nucleolar formation, this technique likely disrupts the nucleus in many ways in addition to the loss of the nucleoli which could also disrupt PCH conformation. The authors mention that the PCH starts forming two cycles before the nucleolus – what is the conformation at this time? Are there pre-nucleolar condensates that form a structure? It would be an interesting way to further determine how the nucleolus impacts PCH assembly and assembly dynamics in physiological conditions instead of just minus rDNA. The minus rDNA could be affecting the condensates in multiple unknown ways to disrupt the PCH, especially since there is a lot of overlap in the box plots in Figure 2. Maybe use a pol1 inhibitor like BMH-21?*

In +rDNA cycle 11 and 12 (Data not shown but reported in Strom et al., 2017), PCH (HP1a) condensates appear and dissolve independent of nucleolar (Fib) condensates, which do not appear until cycle 13. In cycle 13 and beyond, PCH foci form first, followed by nucleoli. These data indicate that the nucleation and initial growth of PCH and Fib condensates occur independently. However, once both structures are formed, they begin to interact during interphase before dissolving in mitosis, as revealed from our live imaging in the early embryo (**Fig. 1b**) and cycling S2R+ cells (**Extended Data Fig 2c**). We did not assess PCH conformation in -rDNA embryos before cycle 14, as embryos were identified as -rDNA and selected for imaging based on the formation of Fib neocondensates, which first appear in cycle 14 (Falahati et al. 2016).

In addition to the -rDNA perturbation, we inhibit Pol-I using low concentrations of ActD to interrogate the impact of nucleolar loss by an orthogonal method, as recommended by two reviewers. Blocking nascent rRNA synthesis did not phenocopy the effects of -rDNA (i.e., Fib displacement to a neocondensate and Pit filling the PCH void), due to the persistence of rRNA produced before treatment. However, it did cause PCH-nucleolar reorganizations consistent with Pit's dual-affinity properties. Specifically, PolI inhibition reduces Pit's nucleolar associations while preserving Pit-PCH interaction causing Pit to redistribute and become enriched at the nucleolar periphery next to HP1a with increased

HP1a occupancy around the nucleolus. These results are presented in **Fig. 6h-j**, **Extended Data Fig. 9** and associated main text.

5) *Figure 3C and Supplemental video 7:*

- *The 'donut' conformation looks similar to the normal PCH conformation to my untrained eye. It would be useful to include the xy, yz and xz views at t=0 and t=70 for with rDNA with and without Fib overlay for comparison.*
- *A number of the cells in this video (minus rDNA) have similar PCH conformation around the neo-condensates to what was previously shown with WT embryos contradicting Figure 3C. Please reconcile.*

We appreciate this suggestion, and +rDNA 3D views are now included in the revised version (new **Fig 2f**) to highlight the difference in PCH organization in +rDNA and -rDNA nuclei. We omit the "Without Fib overlay" images in the main figure only to emphasize the point that the 'PCH-voids' in -rDNA nuclei are devoid of Fib. However, we include single channel and merged views of PCH and nucleoli in wild-type embryos and a *Drosophila* cell line in **Extended Data Fig. 1** and in -rDNA embryos (with +rDNA controls) in **Extended Data Fig. 3**.

Supplementary Video 7 is a Max intensity z projection, which erroneously suggests the presence of Fib in the PCH void in some nuclei, when in fact it is in a lower plane. We review our 3D data again to confirm that Fib neocondensates in -rDNA nuclei are detached from PCH and not in the PCH void.

Minor Changes: All minor changes have been corrected in the revised manuscript. We thank the reviewer for their careful reading and recommendations.

1) *There is no reference to extended figure 3B anywhere in the text.*

Extended Figure 3B: reference corrected in main text.

2) *Line 127 – Without another nuclear marker, it is not confirmed if the PCH is 'lining the nuclear edge' -> I would move the part about figure 1D here as this backs this statement instead of having it at the end of the next paragraph.*

Line 127/ New Line 146: As suggested, we moved text on Lamin and H3K9me2 (PCH lining the nuclear edge early in development) earlier however primary data related to this result has been moved to **Extended Data Fig. 2b** in the current version.

3) *Figure 3d/line 194 - They have confirmed there is protein in the PCH void; however, have not confirmed that 'no nucleoli proteins are present? They have just confirmed that fibrillarin may not be present (again, in the movie there were a few cells where fibrillarin WAS present). Indeed, they go on to show that Pit is in this void. Please reconcile.*

Line 194/ New Line 221: Edited to specify which nucleolar proteins are missing. We agree the original statement was an inaccurate generalization.

4) *Figure 2E – hard to see the DAPI stain here, could just be the printout though.*

Figure 2E/ New **Fig. 2d**: Panel Zoomed in and DAPI changed to grayscale for visual clarity.

5) *Figure 2A - -rDNA and +rDNA HP1a look quite similar to me, But I agree that there is an obvious difference in conformation in 2C.*

Fig. 2a: Panels edited to clearly reflect the change in PCH organization in cycle 14 in -rDNA. Another example of these phenotypes is shown in **Fig. 4b** (cycle 14) with HP1a and Pit.

6) *I would just say in the absence of rDNA through section describing Figure 2, as mentioned above, the absence of rDNA may be affecting the nucleus and DNA/chromosome structure in multiple ways so any changes observed may not be purely due to loss of a nucleoli.*

We edited the main text associated with Fig. 2 to keep the language precise by using “in the absence of rDNA” or ‘-rDNA’. However, we also periodically remind the reader that one of the main phenotypes of rDNA removal is the absence of nucleoli.

7) *Line - 389 – Unless I missed it in the paper, the authors did not show differences in Pit expression or localisation during the different cycles, so this conclusion/discussion seems like a reach for me and I would remove it or reword this section, again, a direct interaction between the two proteins is not characterised.*

Line 389 in the original submission was a speculation Pit’s role on reorganizing PCH around the nucleolus during embryogenesis, based on Pit’s PCH affinity. However, one of the main criticisms during the review was lacking evidence on direct interactions between Pit and HP1a. We now added a new main figure demonstrating the molecular basis of Pit’s HP1a interactions (**Fig. 5**), which includes a co-IP that suggests a weak interaction between HP1a and Pit. Lines 495-505 in the discussion of the revised manuscript presents our updated speculations on Pit-mediated PCH reorganization during *Drosophila* embryonic development.

Paragraph from line 399:

8) *I don’t know where the data are that shows that Pit and HP1a don’t stably mix unless I have completely overlooked a figure.*

The data showing the mixing of Pit and HP1a is **Fig 4b** (colocalization in cycle 14 -rDNA). **Fig 4c** (T=0’) and **Supplemental Movie 9** show how Pit and HP1a are initially mixed in amnioserosa cells, then subsequently Pit demixes from HP1a (**Fig 4c**, T=90’) to form a neocondensate and the PCH void.

9) *Line 413– Suggest using a Pol1 inhibitor here. Again, they stated that the rDNA being in close proximity to the 359bp region may be required for the proximity of the PCH and nucleoli, so maybe just deleting this region is enough to disrupt the PCH conformation.*

Original Line 413 stated “rRNA synthesis doesn’t seem to be required for Pit-HP1a associations”. As suggested, we directly tested the role of rRNA synthesis in Pit-HP1a interactions by performing a Pol-I inhibitor experiment and found that Pit’s localization from the nucleolar center is lost while becoming enriched only the nucleolar periphery at its interface with HP1a (**Fig. 6h-j**). We interpret this as rRNA synthesis is not required for Pit-HP1a associations. If anything, in the absence of Pit-rRNA interactions, availability of Pit to interact with HP1a increases resulting in nucleolar edge enrichment. This is discussed in edited Lines 515-520.

Regarding the proximity of the 359bp repeat and rDNA: while their adjacency at a chromosomal level is required for PCH to be tethered to the nucleolus, it is not sufficient to form the ‘surrounded conformation’ as described in this study. For instance, removing Pit-HP1a interactions in nuclei with intact rDNA and 359bp repeats, ‘unwraps’ HP1a from around the nucleolus while maintaining a tether. Since -rDNA nuclei clearly show increased PCH compaction phenotypes, we discuss these as a likely consequence of increased homotypic HP1a interactions and lack of nucleolar surface interactions in lines 562-565.

Reviewer #3 (Remarks on code availability):

There isn't any code that I could see--only equations.

Apologies for the bad link to the code. We have edited this to make the code for the coarse-grained simulations accessible here:

https://github.com/gauravbajpaimaths/Coarse-grained_model_of_nucleolar_heterochromatin_condensates.